# H2IL-MBOM: A Hierarchical World Model Integrating Intent and Latent Strategy for Opponent Modeling in Multi-UAV Game

## Abstract

In mixed cooperative-competitive multi-agent settings, uncertain decisions create non-stationary learning and mutual security threats. Existing opponent modeling methods typically require access to opponents' private information (observations, actions, goals, policy parameters or rewards) or rely solely on local historical observations, neglecting the intrinsic dynamics between mental states and trajectories. Inspired by human hierarchical reasoning, we propose a hierarchical world model that recursively infers opponents' intentions from their historical trajectories and reasons about their latent strategies from teammates' responses, without needing opponents' private information. Coupled with our Mutual Self-Observed Adversary Reasoning PPO (MSOAR-PPO) algorithm, it establishes a co-adaptation loop between the world model and policy. Evaluations demonstrate that our method outperforms all model-free, model-based, and opponent modeling baselines in multi-UAV games, achieving higher rewards and faster convergence while scaling robustly to 10v10 settings with improved win/survival rates. Its ability to reason about complex opponent behaviors is confirmed by cumulative error analysis and t-SNE visualizations. Superior performance generalizes to StarCraft and Google Research Football benchmarks. Videos are provided in supplemental materials.

## 1 Introduction

In multi-agent environments, agents interact and learn concurrently, leading to diverse state transitions and mental dynamics, and creating non-stationary dynamics that complicate policy learning. This challenge is particularly acute in mixed-motive games, where the fundamental tension between cooperation and competition directly amplifies the non-stationarity and strategic uncertainty. This tension requires agents to cooperate with allies while simultaneously facing opposition from adversaries. In such settings, unknown and evolving opponent policies not only hinder policy improvement but also jeopardize ally safety and curtail overall performance. Therefore, for effective decision-making in mixed-motive scenarios, it is crucial to move beyond modeling allies and instead develop a sophisticated capacity to model opponent behavior and reason about their mental states, which is essential for ensuring operational safety and achieving strategic supremacy.

Opponent modeling and intent reasoning are central to Theory of Mind (ToM), enabling agents to infer opponents' preferences, goals, beliefs, and strategies. The cognitive foundations of this capability are well-established: developmental psychology shows that even infants distinguish between enduring goals and situational actions, recognizing that intentions remain stable while strategies adapt contextually Gergely & Csibra (1997). Neuroscientific evidence further supports this dissociation, revealing distinct encodings for high-level goals in prefrontal regions and action execution in inferior frontoparietal circuits De C. Hamilton & Grafton (2008). Computationally, humans engage in hierarchical causal reasoning, first inferring others' goals and then deriving the specific action plans employed to achieve them Baker et al. (2017).

Existing computational approaches to opponent modeling fall into two categories. Some methods reconstruct policy beliefs from known behaviors, while others extrapolate strategies directly from local observations. However, the former relies on unrealistic assumptions about opponent transparency, while the latter often fails to capture the causal interactions among intentions, strategies, and actions.

Critically, both are ill-suited for the stochastic and dynamic interest alignments that characterize mixed-motive games. Specifically, they do not explain how intentions shape strategies, how agents should react to these inferences, or how mental states co-evolve and influence future trajectories. This lack of continuous reasoning about evolving intentions and strategies is a primary bottleneck for robust performance in mixed-motive environments.

Developing a human-like opponent model or intent reasoning model inevitably presents challenges. Maintaining a multi-hypothesis intention and strategy for opponents with advanced cognitive abilities in dynamic and complex competitive-cooperative scenarios, adapting to a variable number of adversaries with changing intentions, and dealing with the resulting uncertainty in strategy estimation are necessary.

**Motivations, Core idea and Contributions:** To address these challenges and bridge the gap in opponent modeling without relying on private information from the opponents, drawing inspiration from the brain's hierarchical information processing and recursive reasoning mechanisms, we introduce a Hierarchical Interactive Intent-Latent-Strategy-Aware World Model-Based Opponent Model (H2IL-MBOM), in which a Hypernetwork-based Hierarchical Dynamic Dependence Transformer State Space Model (HyperHD2TSSM), along with a Mutual Self-Observed Adversary Reasoning PPO (MSOAR-PPO) for real-time reasoning about opponents' multi-intentions and latent strategies, all without accessing any private information.

In the HyperHD2TSSM, we introduce a hierarchical mental model and action-conditioned transition models that formalize the interactions between the opponent's intentions and the team's actions in the opponent's trajectory transitions, as well as the interactions between the opponent's latent strategies and the team's actions in the allied agents' trajectory transitions. Specifically, we propose a hierarchical opponent modeling framework, HyperHD2TSSM, comprising three components: 1) High-level Dynamic Intent-aware Representation Fusion (HDIRF): High-level History Transformer Encoder (H2TE) + Multi-Intention Transformer Decoder (MITD) employs cross-attention and fusion to aggregate a consensus from teammate inferences, inferring multi-intention queries directly from opponents' past trajectories. 2) Low-level Dynamic Latent-Strategy-aware Representation Fusion (LDLRF): Inspired by our team's reactions serving as a mirror to opponent strategies, Low-level History Transformer Encoder (LHTE) + Multi-Latent-Strategy Transformer Decoder (MLTD) utilizes the same mechanism to predict latent strategy queries based on estimated intentions and historical responses. 3) Interactive Hypernetwork-based Joint Latent Gated Transformer (HJLGT): This transition model interactively infers the future mental states of opponents and reconstructs the trajectories of both opponents and cooperative agents. This design embodies the core philosophy of "inferring intentions from opponents' historical trajectories while understanding latent strategies from teammate responses," implementing a brain-inspired hierarchical recursive architecture that enables interactive modeling and interactive reasoning of co-evolving mental states and trajectories.

Our contributions are six-fold: 1) We propose a hierarchical world model that interactively infers multi-intentions, latent strategies, and trajectories of all agents without using opponents' private information. 2) We design HyperHD2TSSM, which compresses history into latent weights via a hypernetwork and supports interactive prediction of future mental states for all agents without increasing parameters 3) Our method enables any-time-step updates, facilitating parallel training, reducing computational overhead and cumulative error, and offering flexible temporal modeling. 4) We build a hierarchical architecture to model intent-strategy interactions without predefined candidates, and incorporate a hyper-network for individualized reasoning along with the cross-attention consensus mechanism for collaborative and adaptive inference. 5) By integrating MSOAR-PPO with H2IL-MBOM, our agents perform real-time adversarial reasoning from self-observation and adapt rapidly to opponent changes. 6) To the best of my knowledge, this is the first work to build world models for opponent modeling in intense adversarial environments, advancing the development of world models, opponent modeling techniques, and multi-agent adversarial decision-making.

## 2 RELATED WORK

**Opponent modeling.** Opponent modeling aims to infer an opponent's mental states, such as goals, actions, and intentions, to address non-stationarity and gain an advantage in dynamic environments. Existing methods like DPN-BPR+ Zheng et al. (2018) and ToMoP Yang et al. (2018) struggled with continuously evolving opponents. Approaches like RFM Tacchetti et al. (2018), P-BIT Tian

et al. (2020), ROMMEO Tian et al. (2019), TOM Rabinowitz et al. (2018), GSCU Fu et al. (2022), CSP Wu et al. (2023a), OMIS Jing et al. (2024) and Yu et al. (2022b); Zhang et al. (2021) utilized opponents' private information, such as their actions, policy parameters or rewards, as labels to learn and infer their goals, beliefs or strategy representation. PR2 Wen et al. (2019) and GR2 Wen et al. (2021) focused on probabilistic inference but don't simultaneously learn agent policies. GrAMMI Ye et al. (2023) applied multi-hypothesis beliefs and mutual information theory to predict opponent behaviors but misses time-varying dynamics. Although Busch Busch et al. (2022), Wu et al. (2023b) and Shi et al. (2022) used gaussian model, and graph attention or transformer based VAE to predict adversaries' incentive, intents or trajectories, they neglected underlying environmental dynamics and mutual influence between them. In contrast, our method infers multi-evolving opponents' intentions and latent strategies from historical and current observations, without requiring private information, and accounts for the opponent's same reasoning ability.

**World Model.** Current single-agent world models include like MBPO Kaiser et al. (2019), DreamerV1-V3 Hafner et al. (2019a; 2020; 2023) based on RSSM Hafner et al. (2019b), TSS-M developed by Chen et al. (2022), and graphical state space model (GSSM) developed by Wang & Van Hoof (2022). Some extend single-agent models to multi-agent models, categorized as centralized Willemsen et al. (2021) or decentralized Xu et al. (2022); Hu et al. (2021). Recently, Egorov & Shpilman (2022) and Liu et al. (2024) proposed new world models based MARL (MBMARL), MAM-BA and MAZero, and validated them in StarCraft Multi-Agent Challenge (SMAC). However, these models struggle with scalability, often making independent latent state predictions. Xie et al. (2021) used the world model to only infer latent strategy. Our approach builds an interactive multi-agent world model with hierarchical latent states to infer intent and latent strategy for mixed cooperative-competitive environments. By dynamically adjusting latent weights based on neighboring agents' states, our model enables spatiotemporal forecasting and interactive predictions without increasing parameters, offering greater scalability and adaptability compared to centralized and decentralized models.

## 3 METHODOLOGY

**Problem Statement.** We consider mixed cooperative-competitive scenarios involving $N >= 2$ agents. Each agent infers opponents' intentions and strategies and makes decisions based on local observations while interacting with others without accessing private information of competitive agents, such as opponents' learning algorithms, actions, rewards, goals, and incentives. These private details of opponents, including adversaries and missiles, remain diverse, changeable, and unknown to cooperative agents. In this study, we aim to understand opponents' mental states by constructing H2IL-MBOM models from their perspectives, and using these predictions along with observations to inform decision-makings. Therefore, we have two objectives. The Markov decision process comprises a tuple $\langle N, n, M, m, S, A, O, Z, H, R, \gamma \rangle$ where $N$ and $n$ are numbers of cooperative agents and observable cooperative neighbors, respectively; $M$ and $m$ are numbers of opponents and observable opponents, respectively; $S$ is the state sets, $A = \{A_i\}_{i=1}^N, O = \{O_{opp}, O_c\} = \{O_i\}_{i=1}^N = \{O_{opp,i}, O_{c,i}\}_{i=1}^N$ are the action sets and observation sets relative to opponents $O_{opp}$ and cooperative neighbors $O_c$. $z = \{z_I, z_L\} = \{z_i\}_{i=1}^N = \{z_{I,i}, z_{L,i}\}_{i=1}^N$ are incentive representations, which consist of intentions $z_I$ and latent strategies $z_L$. $H = \{H_{opp,t}, H_{c,t}\} = \{\{O_{opp,i,t}\}_{t=t_0,...,t-1}^{i=1,...,N}, \{O_{c,i,t}\}_{t=t_0,...,t-1}^{i=1,...,N}\}$ signifies the agents' historical observations relative to opponents and teammates; and $R, \gamma$ are rewards and discount factor, respectively. The first objective is to maximize the expected return $E_\pi \left[ \sum_{t=0}^{\infty} \gamma^t R_t(s_t, \{a_{i,t} \sim \pi(\cdot|o_{i,t}, z_{I,i,t}, z_{L,i,t})\}_{i=1}^N, s_{t+1}) \right]$, and the second objective involves updating reasoned intentions and latent strategies based on future ground-truth incentive representations.

### 3.1 COGNITIVE INTUITION ABOUT HIERARCHICAL WORLD MODEL

**Intention**: The opponent's high-level tactical objectives, answering What does the opponent want to achieve?" (e.g.,"Attacking" a specific unit, "Retreating").

**Latent Strategy**: Contextualized execution methods for implementing intentions, answering "How does the opponent achieve its intention?" (e.g.,Leveraging angular advantage" for an attack intention).

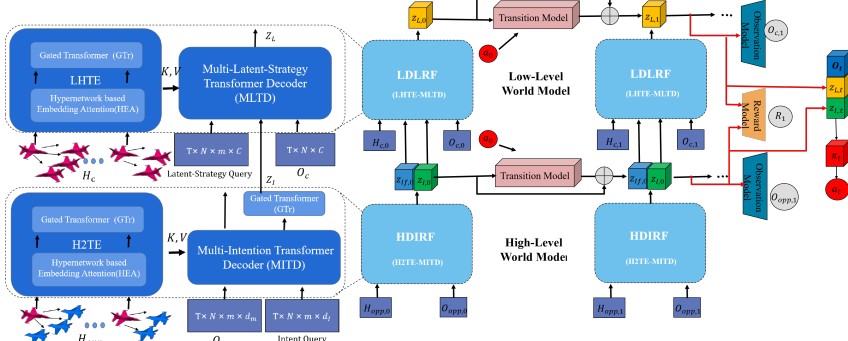

Figure 1: Overview of the H2IL-MBOM, which comprises high-level world model and low-level model. The high-level world model is utilized for reasoning about opponents' intentions and changes in their trajectories, whereas the low-level world model focuses on inferring opponents' latent strategies and their impact on allies' trajectories by taking these intentions into account. By taking as input the estimated mental states of the opponents and local observations, the policy learns to encode opponent behavior in an implicit manner. The inference phase can be found in Figure5.

**Human cognition employs multi-level recursive reasoning in adversarial settings**: **Stage one** infers opponent intentions from historical interactions; **Stage two** involves agents acting based on inferred intentions/strategies while opponents dynamically adjust theirs; **Stage three** updates policies through observed trajectories, forming a closed-loop cycle that drives long-term return maximization. This reveals that opponent modeling requires both hierarchical mental state decomposition (Stage One) and dynamic temporal evolution modeling (Stages Two/Three). However, three challenges persist: opponent mental state unobservability prevents supervised learning; existing methods like VAEs Qi & Zhu (2018); Shi et al. (2022); Wu et al. (2023b) cannot capture mental state co-evolution; and current world models Xie et al. (2021) neglect causal hierarchies.

Guided by human hierarchical reasoning, we propose a hierarchical Transformer architecture: H2TE-MITD infers opponent goals from past observations, extracting macro-behavioral trends ("what they want to do"). LHTE-MLTD employs a cognitive logic that shifts from analyzing "what the opponent has done" to examining "what outcomes their behavior caused us." The rationale is that an opponent's intention determines its strategy choice, which in turn elicits distinctive team responses. These collective responses serve as a behavioral mirror, allowing inverse deduction of latent strategies by correlating reaction patterns with inferred intentions, thereby identifying which strategies the opponent employed to produce observed team reactions.

To overcome static model limitations and capture mental state-behavior co-evolution, we introduce transition models that convert mutual reactions into trajectory observation sequences: opponent intention-team action interactions become opponent-relative trajectory transitions, while strategy-action interactions become teammate-relative trajectory transitions. This enables the model to capture how mental states and behavior co-evolve over time. To capture these evolving dynamics, we approximate higher-level and lower-level transition models using $p_{\psi_I}$ and $p_{\psi_L}$, respectively. The H2TE-MITD module estimates the high-level posterior distribution $q_{\phi_I}(z_{I,i,t}|H_{opp,t}, O_{opp,i,t})$ to infer the opponent's intention $z_{I,i,t}$ based on historical and current observations relative to opponents $H_{opp,t}$, $O_{opp,i,t}$. The LHTE-MLTD module approximates the low-level posterior $q_{\phi_L}(z_{L,i,t}|H_{c,t}, O_{c,i,t}, z_{I,i,t})$ to estimate multi-latent strategy queries $z_{L,i,t}$ based on historical and current observations relative to teammates $H_{c,i,t}$, $O_{c,i,t}$. The hierarchical evidence lower bound (HELBO) is derived via Jensen's inequality in Appendix A.4. Comparisons with RSSM, TSSM, and HyperHD2TSSM are given in Appendix A.5. Also, HJLGT and any-time-step update are detailed in Appendix A.6 and A.7.

### 3.2 DYNAMIC FUSION MECHANISMS OF INTENTIONS AND LATENT STRATEGIES IN OPPONENT MODELING

Intention inference layer estimates $q_{\phi_I}(z_{I,i,t}|H_{opp,t}, O_{opp,i,t})$ by three core mechanisms: First, the temporal consistency modeling mechanism, where H2TE analyzes observations relative to

opponents over 512 time steps, extracting macro-behavioral trends that characterize persistent intentions. Second, the observation-based encoding mechanism, where intention self-attention module uses our observations for opponents to construct feature representations, thoroughly avoiding interference from teammate response patterns and ensuring the purity of intention features. Third, the threat-centric consensus mechanism, where MITD uses team's collective threat consensus to refines intention queries around "which ally faces the greatest threat." This integrated computational process is mathematically formalized through the Bayesian framework: $P(\text{Intent} \mid H_{opp}, O_{opp}) \propto \underbrace{P(O_{opp} \mid \text{Intent})}_{\text{Observation Likelihood}} \cdot \underbrace{P(H_{opp} \mid O_{opp}, \text{Intent})}_{\text{Historical Consistency}} \cdot \underbrace{P(\text{Intent})}_{\text{Intent Prior}}$.

The strategy inference layer estimates $q_{\phi_L}(z_{L,i,t} \mid H_{c,t}, O_{c,i,t}, z_{I,i,t})$ by building an inverse reasoning framework based on the behavioral mirror principle: LHTE forms a "behavioral mirror" by encoding historical team states, comprehensively recording the characteristic response patterns of the team under various strategic pressures. On this foundation, MLTD implements Bayesian inverse reasoning to establish a complete causal chain from observed effects back to potential strategies. The core of this process lies in the concrete computation of the probability formula $P(\text{Strategy} \mid \text{Response}, \text{Intent}, O_c) \propto \underbrace{P(O_c \mid \text{Intent})}_{\text{(Observation Conditioning)}} \cdot \underbrace{P(\text{Response} \mid \text{Strategy}, O_c, \text{Intent})}_{\text{(Response Likelihood)}} \cdot \underbrace{P(\text{Strategy} \mid \text{Intent})}_{\text{Prior}}$: The latent strategy prior is embedded through query initialization, incorporating assumptions about strategy distributions given specific intentions. The intention self-cross attention module computes the observation conditioning term $P(O_c \mid \text{Intent})$, evaluating how current situational evidence aligns with inferred intentions. The likelihood term $P(\text{Response} \mid \text{Strategy}, O_c, \text{Intent})$ is calculated through the latent strategy cross-attention module, assessing how well latent strategies explain current team reactions under the given intent and situational context.

This dual-layer architecture preserves the advantages of direct observation in intention recognition while ensuring the causal rationality of strategy inference, ultimately achieving precise threat assessment and multi-agent cooperative decision-making through the team consensus mechanism.

### 3.2.1 High-level Dynamic Intent-aware Representation Fusion (HDIRF)

During each learning stage, historical states in the most recent steps undergo dynamic change. The intention queries within each MITD layer are derived from the outputs of the previous layer, adapting as the dynamics evolve. Each agent enhances its intent prediction for a given opponent through the team's collective threat consensus, specifically identifying which ally faces the greatest threat from that intent. This approach is grounded in the principle that intentions manifest as consistent patterns in how opponents present themselves to our observational systems.

**High-level History Transformer Encoder (H2TE)** constructs a team-shared representation of opponent behavior patterns by processing historical observations $H_{opp,t} \in \mathbb{R}^{N \times 512 \times D}$ relative to opponents from the perspective of our $N$ agents, where 512 denotes temporal steps, and $D = m \times d_m$ indicates observation dimensionality relative to $m$ opponents. The H2TE captures the spatiotemporal evolution of opponent behavior patterns by analyzing their historical trajectories, extracting macro-level behavioral trends and consistent patterns. Spatial consensus is achieved through hypernetwork-based embedded attention (HEA):

$$w_{H,i,j,t} = \text{Hyper}(H_{i,j,t}), \ e^{i,j,t} = \text{Tanh}(H_{i,j,t} @ w_{H,i,j,t}),$$
$$\alpha^{i,j,t} = \text{softmax}(\text{MLP}([\text{repeat}(e^{i,t}), e^{i,j,t}])), \ \text{Att}H_{i,t} = \frac{1}{m}\sum_{j=1}^{m}\alpha^{i,j,t}\varphi_h(e^{i,j,t}) \quad (1)$$

where $Hyper()$ operator is defined in A.8.1, $H_{i,j,t}$ are historical observations relative to the $j-th$ opponent. The hypernetwork generates distinct parameters $w_{H,i,j,t}$ for each agent, enabling non-shared, individualized reasoning while capturing per-agent spatial dependencies across opponents. Temporal consensus employs Transformer architecture with multi-head attention:

$$q = k = \text{Att}\text{H}opp', \ v = \text{MLP}(k), \ AHopp = \text{Att}\text{H}opp' + \text{MHA}(q, k, v),$$
$$\text{Att}\text{H}opp' = \text{LayerNorm}(AH_{opp} + \text{MLP}(\text{LayerNorm}(AH_{opp}))) \quad (2)$$

where $AttH_{opp}' = reshape(AttH_{opp}) \in \mathbb{R}^{512, N \times C}$ creates a single computational graph capturing both temporal dependency and instantaneous agent interactions through dot-product operations.

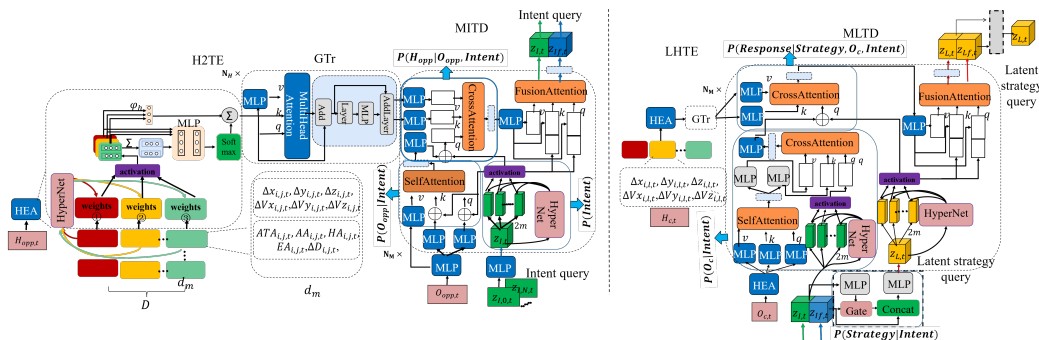

Figure 2: The structure of HDIRF that comprises H2TE and MITD incorporates given observations regarding opponents and multi-learnable intention queries generated by a hypernetwork for interactive intention feature predictions. The structure of LDLRF that comprises LHTE and MLTD incorporates observations of cooperative neighbors and latent strategy queries initialized by intention queries to capture the dynamic impact of multiple intentions on strategy decisions.

The GTr mechanism constructs a more macroscopic perspective on time series similarity and the development of opponent agent behavior.

**Multi Intention Transformer Decoder (MITD)** decodes intentions from the entire team's historical perception through three specialized components. Given the historical feature of opponents $AttH_{opp}$ and current observation $O_{opp} \in \mathbb{R}^{T \times N \times 2m \times d_m}$ regarding opponents, the MITD employs dynamic intention queries $z_I \in \mathbb{R}^{T \times N \times 2m \times d_I}$.

Hypernetwork-based Intention Self-attention Module estimate $P(O_{opp} \mid \text{Intent})$ by fusing embedding $O_{opp,e} = MLP(O_{opp})$ of $O_{opp}$ and $z_I$ to propagate information among $2m$ dynamic intentions. The self-attention mechanism integrates real-time observations with intent queries, emphasizing spatiotemporal features aligned with tactical hypotheses:

$$w_{I,i,j,t} = \text{Hyper}(z_{I,i,j,t}), \ q_{Ih,i,j,t} = \text{Tanh}(z_{I,i,j,t}@w_{I,i,j,t}),$$
$$q_{I,s} = k_{I,s} = \text{MLP}(O_{opp,e}) + q_{Ih}, \ v_{I,s} = \text{MLP}(O_{opp,e}) \tag{3}$$

Here, the hypernetwork adaptively generates different opponent intent query weights for each agent. $q_{I,s}$ serves as active reconnaissance signals combining current situational awareness with tactical intent hypotheses, guiding the attention mechanism to purposefully focus on the spatiotemporal regions and behavioral features most relevant to the current hypothesis. The intention cross-attention module estimating $P(H_{opp} \mid _{opp}, \text{Intent})$ uses its current intent hypothesis to query a "global memory" from historical opponent behavior patterns. This module enables collaborative validation through global memory access:

$$q_{I,c} = \text{MLP}(s_{I,s}) + q_{Ih}, \ k_{I,c} = \text{MLP}(AttH_{opp}), \ v_{I,c} = \text{MLP}(AttH_{opp}) \tag{4}$$

The global memory bank $\mathbb{R}^{1 \times (512N) \times C}$ contains encoded historical observations from all $N$ agents, enabling each agent to query: "Given my current intent hypothesis, which past opponent trajectories observed by any teammate are most relevant?" The intention fusion module establishes team-level threat consensus:

$$q_{I,f} = k_{I,f} = [\text{MLP}(q_{I,c}), q_{Ih}], \ v_{I,f} = [\text{MLP}(q_{I,c}), q_{Ih}] \tag{5}$$

The concatenated features $q_{I,f}, k_{I,f} \in \mathbb{R}^{T \times 2m \times N \times 2C}$ enable cross-agent attention to determine "which ally is most likely to be targeted?" for each opponent intent hypothesis, in which each agent refines its intent prediction based on team's collaborative threat assessment, guiding subsequent cooperative decisions. Finally, $z_I$ are the updated in each layer of MITD.

### 3.2.2 Low-level Dynamic Latent-strategy-aware Representation Fusion (LDLRF)

Based on the inferred intention information, the next step is to further deduce latent strategies and understand how strategies respond to intent prediction. The LDLRF module constructs latent

strategy queries by integrating cooperative agents' historical observations $H_{c,t}$, current neighbor observations $O_c$, and intention features $z_I$ generated by MITD. This process encodes the strategy prior probability $P(\text{Strategy}|\text{Intent})$ by $z_L = MLP([Gate(z_{If}, MLP(z_I)), z_{If}])$ during query initialization, embodying the prior knowledge of "conditional probabilities of strategies given specific intentions." Meanwhile, the current observation embedding $E_c = HEA(O_c)$ provides real-time context for strategy inference. The core mechanism of LDLRF lies in the fact that different latent strategies under the same opponent intention elicit distinct response patterns from teammates. By analyzing the correlation between these specific response patterns and inferred intentions, the module achieves a cognitive process of inversely reasoning about the opponent's latent strategies from team reaction effects. The specific definition of the $Gate$ operator can be found in Appendix A.8.2.

**Low-level History Transformer Encoder (LHTE)** constructs a behavioral mirror by encoding historical team states $H_{c,t} \in \mathbb{R}^{N \times 512 \times D}$ using the same HEA and GTr operations as H2TE. This generates $AttH_c$ representing the team's coordinated reactions under adversarial pressure.

**Multi-Latent Strategy Transformer Decoder (MLTD)** implements the Bayesian formulation by three-component reasoning chain, and performs true inverse reasoning: it takes the effect (team response) and context (intent) as inputs, and infers the most likely cause (opponent strategy) by team consensus on the same opponent. The intention self-cross attention module estimating $P(O_c \mid \text{Intent})$ first links opponent intent to team context:

$$q_{Ls} = \text{MLP}(E_c), \ k_{Ls} = k_{Ls} = \text{MLP}(E_c), \ k_{LI} = v_{LI} = \text{MLP}(\text{LayerNorm}(E_c + \text{MHA}(q_{Ls}, k_{Ls}, v_{Ls}))),$$
$$q_{LI} = \text{Tanh}(z_I @ \text{Hyper}(z_I)) \tag{6}$$

This constructs intent-driven queries $q_{LI}$ that attend to team context $k_{LI}, v_{LI}$, assessing "how threatening this intent seems" given current team reactions.

The latent strategy cross-attention module is used to estimate $P(\text{Response} \mid \text{Strategy}, O_c, \text{Intent})$ through attention mechanism between the hybrid query and historical response patterns:

$$q_{Lc} = \text{MLP}(s_{LI}), \ q_{Lh} = \text{Tanh}(z_L @ \text{Hyper}(z_L)), \ q_{L,c} = q_{Lc} + q_{Lh} \tag{7}$$

where $s_{LI}$ is the output of the previous module, which has integrated the information from $O_c$. The hybrid query $q_{L,c} = q_{Lc} + q_{Lh}$ fuses current team context $s_{LI}$ with strategy hypotheses, while $k_{L,c}, v_{L,c} \in \mathbb{R}^{1 \times (512N) \times C}$ represent historical team response patterns. The attention scores $q_{L,c} \cdot k_{L,c}^\top$ compute the similarity between current context (augmented with strategy hypotheses) and historical team responses, implementing the abductive reasoning: "Given $O_c$, what strategies employed by opponents best explain both the teammates' characteristic responses and opponent's current intent?" In other words, it identifies the opponent's primary target by detecting which teammate most consistently triggers reactive behaviors, thereby modeling latent causal relationships between opponent behavior patterns and team responses.

Finally, the latent strategy fusion module establishes team consensus on strategy-threat relationships:

$$q_{L,f} = [\text{MLP}(q_{L,c}), q_{Lh}], \ k_{L,f} = [\text{MLP}(q_{L,c}), q_{Lh}], \ v_{L,f} = \text{MLP}(q_{L,c}) \tag{8}$$

where $q_{I,f}, k_{I,f} \in \mathbb{R}^{T \times 2m \times N \times 2C}$. It performs team-level threat assessment through cross-agent attention, enabling each agent to continuously refine its reasoning about strategic focus through collective intelligence integration, ultimately determining which allied teammate receives the highest strategy-threat attention weight. This identification facilitates coordinated and precise team support by determining which allied teammate receives the highest strategy-threat attention weight.

## 3.3 CO-ADAPTIVE LOOP & MSOAR-PPO

Two teams engage in independent policy learning, value learning, and world model learning based on their local observations due to the limitation of imperfect game. During the execution phase, the policy of each agent relies solely on: (1) local observations relative to cooperative adjacent agents ($O_c$) and opponents ($O_{opp}$), and (2) mental states inferred via the hierarchical model. Furthermore, the world model operates through iterative observation-action-reflection cycles and is updated via HELBO loss (Eq. 20) using collected interaction data. The world model, in conjunction with the policy Cheng et al. (2024), generates multi-step imagined trajectory observations. These synthesized trajectory observations are subsequently combined with real interaction data to compute MARL policy and value objectives, thereby facilitating training and enhancing policy generalization. The any-time-step update and pseudocode are provided in Sections A.7 and A.9, respectively.

## 4 EXPERIMENTS

We evaluate our method in mixed cooperative-competitive environments: Gym-JSBSim, SMAC, and Google Research Football (GRF). Gym-JSBSim that serves as a benchmark provides high-fidelity 6-DOF dynamics for fixed-wing UAV control. In a 4v4 multi-UAV task, we compare against baselines to assess learning performance, reasoning capabilities, scalability test, and module ablation. Cumulative errors and t-SNE visualizations of inferred intentions and latent strategies further validate reasoning accuracy in Appendice A.11. Additional results on environment settings, hyperparameters, scalability test, and visualizations are in Appendices A.3, A.14, A.12, and A.16, respectively. To enhance robustness, both teams are trained as independent learners, avoiding built-in AI or self-play and making ELO inapplicable. In the testing phase, win rate is used as the evaluation criterion in equal-team scenarios, while survival rate is adopted in unequal-team settings. Also, the agents compete against opponents equipped with other MARL methods that were never encountered during training. The visual results, particularly in Appendix A.16, further investigate the impact of height-related reward components. For SMAC and GRF, we compare with baseline methods, using built-in AI for the opponent team.

### 4.1 COMPARISON WITH VARIOUS BASELINE METHODS

For each algorithm, we use the same network architecture as described in corresponding literature. To ensure fair comparison, we train these baseline algorithms with 5 random seeds under the same conditions such as initial conditions, number of simulation steps, observation space, action space, and reward functions.

**Comparison with model-free MARL.** We compare our method with CTDE MARL (MAPPO Yu et al. (2021), MADDPG Lowe et al. (2017)), decentralized MARL (HAPPO, HADDPG Zhong et al. (2024)), and a model-free RL baseline where MAPPO/HAPPO act randomly and MADDPG/HADDPG act deterministically. All algorithms share the same network architecture and hyperparameters from their original papers. As shown in Figure. 3a, other MARLs yield negative rewards. MADDPG and HADDPG perform poorly because deterministic actions cannot cope with dynamic, evolving opponents, leaving both sides vulnerable to missile attacks. MAPPO and HAPPO improve stability but still fluctuate under environmental non-stationarity, keeping rewards below zero. Our method (Figure. 3b) achieves near-100 rewards due to three factors: (1) hierarchical decomposition (H2TE-MITD, LHTE-MLTD) enables structured reasoning over intentions and latent strategies; (2) mental simulation (HJLGT) supports multi-step prediction of adversarial and team trajectories for proactive decisions; and (3) MSOAR-PPO dynamically couples inference and policy updates, refining mental states through real interactions. This integration allows real-time adaptation to unseen tactics, mitigates non-stationarity, and ensures safe, coordinated responses in complex adversarial settings.

**Comparison with other opponent modeling methods.** We compared recent opponent-modeling approaches—ROMMEO Tian et al. (2019), PR2 Wen et al. (2019), TDOM-AC Tian et al. (2023), and AORPO Zhang et al. (2021). As shown in Figure. 3c, all exhibit negative, fluctuating rewards: AORPO oscillates around –100, while PR2 fluctuates more sharply around –203.2. These results indicate that none accurately capture opponent behavior, leading to erroneous predictions and disadvantaged decisions. AORPO further struggles to model environmental dynamics using MBPO's world model. A core limitation of prior methods is their reliance on opponent actions as labeled data—an unrealistic assumption in real adversarial settings. Intentions and strategies are latent and evolve over time, making action-level modeling inadequate. As discussed in Section 3.1, future trajectories depend on the temporal evolution of mental states. Yet most methods, including VAE-based ones, learn static latent representations and directly reconstruct trajectories, failing to capture the dynamics of evolving intentions and strategies, and thus cannot anticipate future behaviors.

**Comparison with other MBMARL.** In recent MB-MARL studies, comparisons mainly focus on MAMBA Egorov & Shpilman (2022). MAZero Liu et al. (2024) is excluded because its Monte Carlo Tree Search (MCTS) is designed for discrete action spaces, whereas our UAV game features a five-dimensional continuous space (aileron, elevator, rudder, thrust, missile launch), which MAZero cannot effectively handle. Qualitatively, our method converges rapidly to positive rewards and maintains stable performance with only minor fluctuations. In contrast, MAMBA converges slowly and shows higher variability, especially early in training. Quantitatively, our approach surpasses zero reward within 5 million ($M$) steps and reaches about 100 by 10 $M$ steps, sustaining 100–150

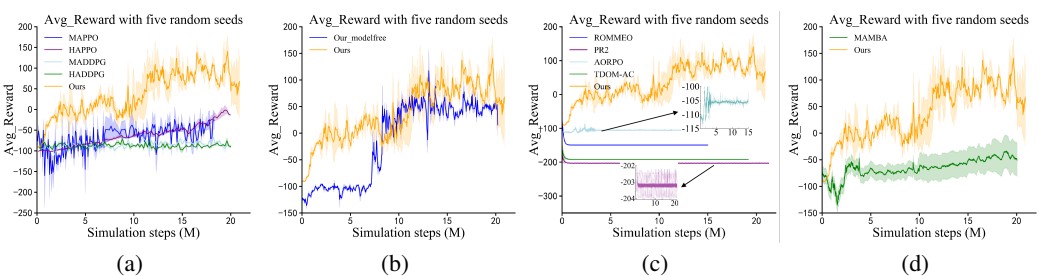

Figure 3: Performance comparison of various methods.

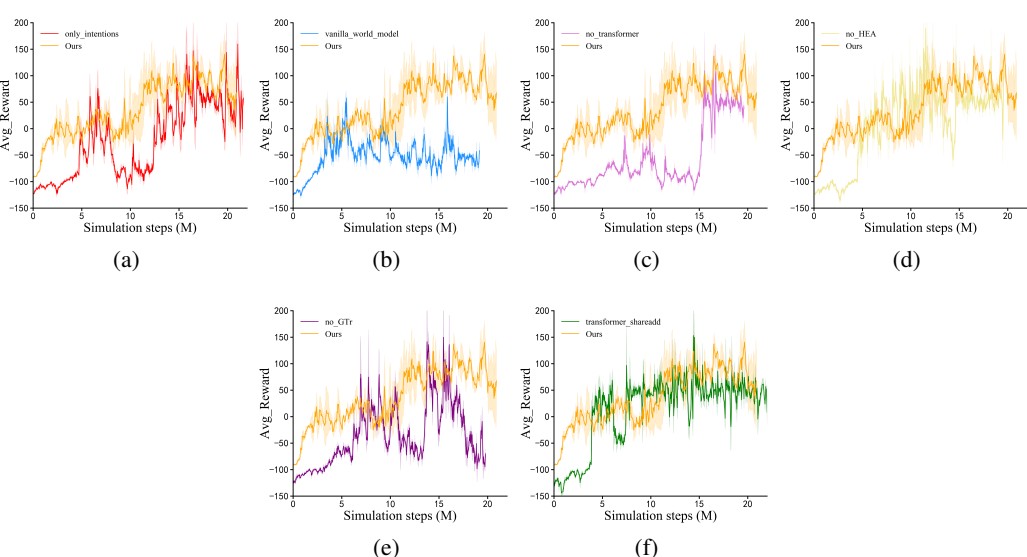

Figure 4: Results for ablation study. (1) Low-level strategy modeling is crucial, as intention-only inference causes mid-game performance degradation due to inability to discern strategy-specific responses; (2) Historical encoding prevents local optima by capturing temporal opponent dynamics; (3) Transformer/HEA components ensure stable convergence through structured reasoning; (4) Hypernetworks enable adaptive agent-specific inference without homogenization, accelerating learning.

between 10 $M$ and 20 $M$ steps. MAMBA stays below zero until roughly 10 $M$ steps and continues to fluctuate around –50 even after 20 $M$ steps, with only marginal late-stage improvement. These results highlight the superiority of our approach in terms of both efficiency and effectiveness.

## 4.2 ABLATION STUDY

In this ablation study, we study the importance of each module in H2IL-MBOM by removing the low-level world model related to latent strategies (only intentions inference version), all history encoders (vanilla world model), Transformer and HEA, transition model, replacing GTr with local time Transformer, and replacing hypernetwork-add operator with share network-add operator in the Transformer. Figure 4a shows that modeling only intentions degrades mid-game performance, confirming the need for low-level strategy inference and the low-level world model. Figure 4b reveals that using only current observations leads to local optima, as short-sighted inference fails to capture evolving opponent dynamics. This highlights the importance of historical context for long-term reasoning. Figures 4c–4f show that removing Transformer, HEA, GTr, or the hypernetwork slows convergence and destabilizes rewards, validating their role in H2TE-MITD and LHTE-MLTD.

Overall, the ablation confirms the critical role of hierarchical modeling, historical encoding, structured attention, and dynamic transitions.

### 4.3 GENERALIZATION TESTING IN SMAC AND GRF

The table 1 presents a comprehensive comparison of test win rates achieved by various state-of-the-art MARL algorithms, including our proposed method, MAPPO Yu et al. (2022a), QMIX Rashid et al. (2020), QPLEX Wang et al. (2020), RODE Wang et al. (2021), MAMBA Egorov & Shpilman (2022), and MAZero Liu et al. (2024). across SMAC scenarios. It's noting that the total interactive steps are aligned with the settings used in MAZero to ensure a fair and valid comparison.

Table 1: Comparison of test Win Rate with state-of-the-art MARL in the SMAC scenarios: **the total interactive steps are aligned with the settings used in MAZero** to ensure a fair.

| Map | Ours | MAPPO | QMIX | QPLEX | RODE | MAMBA | MAZero | Steps |
|---|---|---|---|---|---|---|---|---|
| 2s_vs_1sc | 100 | 100 | 0 | 50.62 | 0 | 100 | 100 | $1e5$ |
| 2m_vs_1z | 100 | 20.75 | 2.9 | 45.50 | 0 | 90 | 100 | $5e4$ |
| 3m | 93.75 | 60.12 | 42.75 | 55.37 | 0 | 93 | 100 | $5e4$ |
| 3s_vs_5z | 97.14 | 22.37 | 85 | 96.4 | 78.9 | 20 | / | $5e4$ |
| 5m_vs_6m | 71.87 | 40.14 | 63.37 | 65.60 | 90 | 40.50 | 90.12 | $1e6$ |
| 10m_vs_11m | 93.75 | 75.12 | 85.57 | 90.87 | 60.37 | 60.12 | 89.30 | $1e6$ |
| So_many_baneling | 96.87 | 30.87 | 6.75 | 30.62 | 0 | 95 | 99.87 | $5e4$ |
| 2c_vs_64zg | 78.12 | 35.27 | 2.6 | 0 | 66.87 | 35 | 90 | $4e5$ |

Our method achieves perfect 100% win rates in both "2s_vs_1sc" (matching top-performing MAP-PO) and asymmetric "2m_vs_1z" scenarios, significantly outperforming QMIX (2.9%), QPLEX (45.50%), and MAPPO (20.75%). In "3m" environments, it maintains 93.75% win rates, substantially exceeding MAPPO (60.12%) and QMIX (42.75%). The algorithm demonstrates particular strength in complex scenarios: achieving 97.14% in "3s_vs_5z" (surpassing QPLEX's 96.4% and far exceeding MAPPO's 22.37%) and 93.75% in large-scale "10m_vs_11m" (outperforming MAZero's 89.30% and QPLEX's 90.87%). These results validate our method's superior coordination in heterogeneous settings and excellent scalability in high-dimensional multi-agent environments.

Additionally, we have conducted GRF experiments in Table 2, our method demonstrates significant advantages in dynamic adversarial scenarios. In the rPS scenario with randomized initial positions, our approach achieves a win rate of 89.94%, substantially outperforming HAPPO (77.30%) and MAPPO (76.83%). In the CA scenario requiring precise coordination, our method attains a 93.09% win rate, also exceeding HAPPO (92.00%) and MAPPO (87.76%). These two scenarios share the common characteristic of requiring real-time inference of opponent intentions and dynamic strategy adjustment. Our hierarchical intention-strategy representation system plays a crucial role in such tasks, achieving superior tactical response capabilities compared to traditional methods through online learning of intention evolution and hypernetwork-based coordination mechanisms.

Table 2: Win Rates (%) of Different Methods in Various Scenarios

| Scenario | Ours | HAPPO | MAPPO | QMIX |
|---|---|---|---|---|
| PS | 92.27 | 96.93 | 94.42 | 8.05 |
| rPS | 89.94 | 77.30 | 76.83 | 8.08 |
| 3v1 | 92.54 | 94.74 | 88.03 | 8.12 |
| CA | 93.09 | 92.00 | 87.76 | 15.98 |

## 5 CONCLUSIONS

This paper introduces a novel opponent modeling method that integrates multi-intention and latent strategy inference into the world model. Using a hierarchical architecture, we study the impact of opponents' intentions on their strategies and predict both teammates' and opponents' trajectories. We also propose MSORA-PPO, enabling teams to independently learn their own H2IL-MBOM, infer adversarial strategies and intentions from historical observations, and integrate these inferred mental states with local observations to make decisions.

ETHICS STATEMENT

We acknowledge the broader societal implications of our work on opponent modeling in multi-agent adversarial environments.

On the positive side, our method advances the capability of AI agents to understand and adapt to complex, dynamic opponents through hierarchical inference of intentions and latent strategies. This could benefit applications such as autonomous systems requiring safe interaction with unpredictable agents, where anticipating adversarial behavior can improve safety and coordination.

Our approach does not use real human data or sensitive attributes (e.g., race, gender), and all experiments are conducted in simulated environments (e.g., SMAC, GRF, Gym-jsbsim simulator). Therefore, no personal data is involved, and there is no direct risk of demographic bias in training. Nevertheless, we caution that any system capable of inferring private mental states should be subject to strict regulatory oversight before deployment.

REPRODUCIBILITY STATEMENT

We have provided detailed designs of transition model, HDIRF, and LDLRF in the Appendix A.6,3.2.1, and 3.2.2, respectively. The training details including environmental settings, hyperparameters are shown in the AppendixA.3, 1, and A.14.

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

# A  APPENDIX

## A.1  LIST OF ABBREVIATIONS USED IN THE PAPER

Table 3: List of Abbreviations and Explanations

| Abbreviation | Explanation |
|---|---|
| H2IL-MBOM | Hierarchical Interactive Intent-Latent-Strategy-Aware World Model based Opponent Model |
| MSOAR-PPO | Mutual Self-Observation Adversarial Reasoning with PPO |
| HyperHD2TSSM | Hypernetwork-Based Hierarchical Dynamic Dependency Transformer State Space Model |
| HDIRF | High-Level Dynamic Intent-aware Representation Fusion |
| LDLRF | Low-Level Dynamic Latent-Strategy-aware Representation Fusion |
| H2TE-MITD | High-Level History Transformer Encoder - Multi-Intent Transformer Decoder in HDIRF |
| LHTE-MLTD | Low-Level History Transformer Encoder - Multi-Latent Policy Transformer Decoder in LDLRF |
| HILGT | Interactive Hypernetwork Joint Latent Gating Transformer |
| HEA | Hypernetwork-Based Embedding Attention Mechanism |
| HELBO | Hierarchical Evidence Lower Bound |
| CTDE | Centralized Training with Decentralized Execution (a paradigm in MARL) |
| SMAC | StarCraft Multi-Agent Challenge (benchmark environment) |
| GRF | Google Research Football (benchmark environment) |
| RSSM | Recurrent State Space Model |
| TSSM | Transformer State Space Model |
| VAE | Variational Autoencoder |
| MHA | Multi-Head Attention |
| KL | Kullback-Leibler Divergence |

## A.2  LIMITATIONS

This study still has some limitations. First, we did not integrate multi-source information, which is important in practice and requires more representation learning. Second, although we analyzed the t-SNE distribution of opponent intentions and strategies, we have not yet studied the driving factors behind these distributions, necessitating further techniques to analyze opponents in detail. Last but not least, the method has not been validated on physical devices in the real world.

## A.3  EXPERIMENT DETAILS

The environmental setup presents significant challenges: In the gym-jsbsim environment, each aircraft is equipped with four missiles that can autonomously lock onto and pursue targets upon launch, with an engagement duration of approximately 20 seconds. The aircraft's maximum speed is limited to 2 Mach (twice the speed of sound), while the missiles travel at 4 Mach(four times the speed of sound). The objective is to achieve dominant positioning and optimal missile launch timing in this highly dynamic setting, maximizing combat effectiveness while minimizing resource expenditure. This environment differs substantially from conventional benchmarks due to its high dimensionality, extreme velocities, the substantial speed differential between missiles and aircraft, complex spatial relationships, and the critical importance of launch timing to prevent premature depletion of limited munitions.

Two opposing teams, designated as red and blue, initialize their UAV swarms randomly at starting positions separated by a distance of 16 km, both operating at an altitude of 6 km above sea level. The central coordinate for the engagement zone is set at $120°$ longitude and $60°$ latitude, with an elevation of $0\ m$. The flight altitude for all UAVs is constrained between $2.5\ km$ and $20\ km$. Each UAV is limited to a maximum acceleration of $10\ m/s2$, a maximum attack angle of $4°$, and an engagement range of up to $14\ km$. The kinematic states used in our observation space, specifically relative distance, velocity, and attitude, are grounded in well-established aerospace sensing technologies, making their acquisition realistic rather than hypothetical: 1) Relative distance is obtained through active electromagnetic ranging, where onboard radar systems (including primary, secondary, millimeter-wave, and lidar) measure the round-trip time $\Delta t$ of transmitted pulses and calculate distance using the fundamental equation $\frac{c \cdot \Delta t}{2}$, where $c$ represents the speed of light and denotes the round-trip time of the electromagnetic pulse. 2) Relative velocity is derived through inertial and multi-sensor fusion, integrating data from MEMS-based IMUs (accelerometers and gyroscopes) with supplementary inputs from GPS, barometric sensors, and visual odometry to construct accurate 3D velocity vectors. 3) Relative attitude and orientation are provided by phased-array radar systems, which utilize electronic beam steering to lock onto targets and determine their

azimuth and elevation angles relative to the host aircraft through phase analysis of returning signals. These sensing methodologies are operationally deployed in modern aerial systems, confirming that the observational inputs to our model are consistent with real-world capabilities and do not rely on unrealistic assumptions.

**Observation Space.** Each agent's observation space includes the ego-state $O_e$, observations relative to cooperative adjacent agents $O_c$, observations relative to opponents and encountered missiles $O_{opp}$. Concretely, $O_e$ comprises ego altitude, sine and cosine values of ego roll angle, sine and cosine values of ego pitch angle, and three velocity components in the body coordinate system; the observation relative to each neighbor includes three components $\{\Delta x_{i,j,t}, \Delta y_{i,j,t}, \Delta z_{i,j,t}\}_{j=1,...,2m}$ of relative position and three components $\{\Delta V x_{i,j,t}, \Delta V y_{i,j,t}, \Delta V z_{i,j,t}\}_{j=1,...,2m}$ of relative velocity in the northeast celestial coordinate system; in addition to the above information, $O_{opp}$ also includes antenna angle $\{ATA_{i,j,t}\}_{j=1,...,2m}$, aspect angle $\{AA_{i,j,t}\}_{j=1,...,2m}$, elevation angle $\{EA_{i,j,t}\}_{j=1,...,2m}$, horizontal angle $\{HA_{i,j,t}\}_{j=1,...,2m}$, and distance $\{\Delta D_{i,j,t}\}_{j=1,...,2m}$ relative to each opponent and missile.

**Action Space.** Each agent in a uav-game scenario has five continuous actions, including aileron angle, elevator angle, rudder angle, thrust, and sign of launching missiles. A sign value greater than 0 indicates that it can be launched, otherwise it will not be launched. The specific launch also depends on the attack angle, distance, and enemy survival number on the battlefield.

**Rewards.** Rewards primarily consist of distance-angle reward relative to opponents, height-angle reward relative to opponents, speed-angle reward relative to opponents, penalties for collisions (-5) and proximity between teammates, altitude safety reward, attack angle reward, crash penalties(-100), penalties for the number of missiles (-10), penalties for being killed (-100), rewards for killing opponents (+100), and survival rewards(+1). Some reward functions are given as follows:

1. get reward regarding position of planes

$$
\begin{aligned}
a &= \frac{\text{ATA} + \text{AA}}{2\pi} \\
dd &= \frac{\text{target\_dist} - \dfrac{\text{delt\_D}}{10000}}{\text{target\_dist}} \\
\text{reward} &= \begin{cases} e^{0.8+dd} \cdot (8 - 8a), & \text{if } a < 0.55 \\ e^{0.8-dd} \cdot (8 - 8a), & \text{otherwise} \end{cases}
\end{aligned}
\tag{9}
$$

2. get reward regarding position of missiles

$$
\begin{aligned}
\text{delt\_D} &= \frac{\text{delt\_D}}{10000} \\
\text{reward} &= -\max\left( -\frac{10}{\text{target\_dist}} \cdot \text{delt\_D} + 10,\ 0 \right)
\end{aligned}
\tag{10}
$$

3. get reward regarding potential of planes

$$
\begin{aligned}
a &= \frac{\text{ATA} + \text{AA}}{2\pi} \\
\text{orientation\_reward} &= f_{\text{orientation}}(\text{ATA}, \text{AA}) \\
\text{height\_range\_reward} &= f_{\text{range}}\left( \frac{|\text{delta\_H}|}{5000} \right) \\
\text{reward} &= \text{orientation\_reward} \cdot \text{height\_range\_reward}
\end{aligned}
\tag{11}
$$

4. get reward regarding potential of missiles

$$dd = \frac{\text{target\_dist} - \dfrac{\text{delta\_H}}{5000}}{\text{target\_dist}}$$

$$\text{reward} = \begin{cases} e^{0.7-dd} \cdot \left(2 - \dfrac{\text{missile\_v}}{\text{ego\_v}}\right), & \text{if } 2 - \dfrac{\text{missile\_v}}{\text{ego\_v}} > 0 \\[2ex] e^{-(0.7-dd)} \cdot \left(2 - \dfrac{\text{missile\_v}}{\text{ego\_v}}\right), & \text{otherwise} \end{cases} \tag{12}$$

5. get orientation function(v0)

$$f_{\text{orientation}}(\text{ATA}, \text{AA}) = \frac{1 - \tanh\left(9(\text{ATA} - \dfrac{\pi}{9})\right)}{3} + \frac{1}{3}$$
$$+ \min\left(\frac{\tanh^{-1}\left(1 - \max\left(\dfrac{2\text{AA}}{\pi}, 10^{-4}\right)\right)}{2\pi}, 0\right) + 0.5 \tag{13}$$

6. get range function (v0)

$$f_{\text{range}}(R) = \frac{\exp\left(-0.004(R - \text{target\_dist})^2\right)}{1 + \exp\left(-2(R - \text{target\_dist} + 2)\right)} \tag{14}$$

7. get reward regarding velocity of planes

$$a = \frac{\text{ATA} + \text{AA}}{2\pi}$$
$$\text{proj\_dist} = \delta_x \delta_{Vx} + \delta_y \delta_{Vy} + \delta_z \delta_{Vz}$$
$$\text{Angle} = \arccos\left(\text{clip}\left(\frac{\text{proj\_dist}}{\text{delt\_D} \cdot \delta_v + 10^{-8}}, -1, 1\right)\right)$$
$$a1 = \cos(\text{Angle})$$
$$dd = \frac{\text{angle\_max} - |\text{angle}|}{\text{angle\_max}}$$
$$\delta_v = \frac{\text{enemy\_v}}{\text{ego\_v}}$$
$$\text{reward} = \begin{cases} e^{0.8+dd} \cdot (2 - \delta_v), & \text{if } a1 > 0 \wedge \delta_v \leq 1 \\ e^{0.8-dd} \cdot (2 - \delta_v), & \text{if } a1 > 0 \wedge \delta_v > 1 \wedge (2 - \delta_v) > 0 \\ e^{-(0.8-dd)} \cdot (2 - \delta_v), & \text{if } a1 > 0 \wedge \delta_v > 1 \wedge (2 - \delta_v) \leq 0 \\ e^{0.8+dd} \cdot (2 - \delta_v), & \text{if } a1 < 0 \wedge \delta_v > 1 \wedge a \leq 0.25 \wedge (2 - \delta_v) > 0 \\ e^{-(0.8-dd)} \cdot (2 - \delta_v), & \text{if } a1 < 0 \wedge \delta_v > 1 \wedge a \leq 0.25 \wedge (2 - \delta_v) \leq 0 \\ 5\left(1 - \dfrac{|\text{ATA}|}{\text{angle\_max}}\right), & \text{if } a1 < 0 \wedge \delta_v > 1 \wedge a > 0.25 \\ e^{0.8-dd} \cdot (2 - \delta_v), & \text{if } a1 < 0 \wedge \delta_v \leq 1 \wedge a > 0.75 \\ 5\left(1 - \dfrac{|\text{ATA}|}{\text{angle\_max}}\right), & \text{otherwise} \end{cases} \tag{15}$$

8. get reward regarding velocity of missiles

$$v_{\text{decrease}} = \frac{\|\mathbf{v}_{\text{missile}}^{\text{previous}}\| - \|\mathbf{v}_{\text{missile}}\|}{340}$$
$$\theta = \frac{\mathbf{v}_{\text{missile}} \cdot \mathbf{v}_{\text{aircraft}}}{\|\mathbf{v}_{\text{missile}}\| \cdot \|\mathbf{v}_{\text{aircraft}}\|}$$
$$\text{reward} = \begin{cases} \dfrac{\theta}{\max(v_{\text{decrease}}, 0) + 1}, & \text{if } \theta < 0 \\ \theta \cdot \max(v_{\text{decrease}}, 0), & \text{otherwise} \end{cases} \tag{16}$$

9. get reward regarding proximity

$$
\begin{aligned}
p &= -\frac{10}{\text{target\_dist}} \cdot \delta_D + 10 \\
c &= \left\{ \begin{array}{ll} 0, & \text{if } p < 0 \\ p, & \text{otherwise} \end{array} \right. \\
\text{penalty\_proximity} &= c \\
\text{reward} &= -c
\end{aligned}
\tag{17}
$$

10. get reward regarding safety altitude

$$
\begin{aligned}
P_v &= \left\{ \begin{array}{ll} -\text{clip}\left(\frac{v_z}{K_v} \cdot \frac{\text{safe\_altitude} - z}{\text{safe\_altitude}},\ 0,\ 1\right), & \text{if } z \le \text{safe\_altitude} \\ 0, & \text{otherwise} \end{array} \right. \\
P_H &= \left\{ \begin{array}{ll} \text{clip}\left(\frac{z}{\text{danger\_altitude}},\ 0,\ 1\right) - 2, & \text{if } z \le \text{danger\_altitude} \\ 0, & \text{otherwise} \end{array} \right. \\
\Delta h &= z - z_{\text{initial}} \\
\Delta H &= \left\{ \begin{array}{ll} 10 \cdot \dfrac{\Delta h}{z_{\text{initial}}} - 0.5 \cdot [\text{elevator} < 0] + & \\ 1 \cdot [\text{elevator} > 0 \wedge \text{altitude\_change} > 0] - 1, & \text{if } \Delta h < 0 \\ 0.8, & \text{otherwise} \end{array} \right. \\
\text{reward} &= P_v + P_H + \Delta H
\end{aligned}
\tag{18}
$$

11. get reward regarding attack angle

$$
\text{reward} = \left\{ \begin{array}{ll} -1, & \text{if } |\alpha| \ge 30° \\ 0, & \text{otherwise} \end{array} \right.
\tag{19}
$$

## A.4   HIERARCHICAL VARIANCE INFERENCE

To capture these evolving dynamics, we approximate higher-level and lower-level transition models using $p_{\psi_I}$ and $p_{\psi_L}$, respectively. The H2TE-MITD module estimates the high-level posterior distribution $q_{\phi_I}(z_{I,i,t}|H_{opp,t}, O_{opp,i,t})$ to infer the opponent's intention $z_{I,i,t}$ based on historical and current observations relative to opponents $H_{opp,t} = \{O_{opp,i,t}\}_{t=t_0,\dots,t-1}^{i=1,\dots,N}$, $O_{opp,i,t}$. The LHTE-MLTD module approximates the low-level posterior $q_{\phi_L}(z_{L,i,t}|H_{c,t}, O_{c,i,t}, z_{I,i,t})$ to estimate multi-latent strategy queries $z_{L,i,t}$ based on historical and current observations relative to teammates $H_{c,i,t} = \{O_{c,i,t}\}_{t=t_0,\dots,t-1}^{i=1,\dots,N}$, $O_{c,i,t} = \{O_{i,l,t}\}_{l=1,\dots,n(l \ne i)}$, and inferred intent queries $z_{I,i,t}$, reflecting how intentions impact strategies. The observation models $p_{\theta_I}(O_{opp,i,t}|z_{I,i,t}, h_{I,i,t})$ and $p_{\theta_L}(O_{c,i,t}|z_{L,i,t}, h_{L,i,t})$ predict observations regarding opponents' trajectories and cooperative agents' trajectories. The hierarchical evidence lower bound (HELBO) is derived via Jensen's inequality as follows:

$$
\begin{aligned}
&\log p(O_{opp,1:N,1:T}, O_{c,1:N,1:T}, a_{1:N,1:T}, h_{I,1:N,1:T}, z_{I,1:N,1:T}, h_{L,1:N,1:T}, z_{L,1:N,1:T}) \\
&= \log E_{q(z_{1:N,1:T}|H_{1:T},O_{1:N,1:T})}\left[\frac{p(O_{opp,1:N,1:T}, O_{c,1:N,1:T}, a_{1:N,1:T}, h_{I,1:N,1:T}, z_{I,1:N,1:T}, h_{L,1:N,1:T}, z_{L,1:N,1:T})}{q(z_{1:N,1:T}|H_{1:T},O_{1:N,1:T})}\right] \\
&\ge E_{q(z_{1:N,1:T}|H_{1:T},O_{1:N,1:T})} \log \left[\frac{p(O_{opp,1:N,1:T}, O_{c,1:N,1:T}, a_{1:N,1:T}, h_{I,1:N,1:T}, z_{I,1:N,1:T}, h_{L,1:N,1:T}, z_{L,1:N,1:T})}{q(z_{1:N,1:T}|H_{1:T},O_{1:N,1:T})}\right] \\
&= \sum_{t=1}^{T} \sum_{i=1}^{N} \begin{array}{l} E_{q(z_{I,i,1:t}|H_{opp,1:t},O_{opp,i,1:t})}(\log[p(O_{opp,i,t}|h_{I,i,t}, z_{I,i,t})]) + E_{q(z_{L,i,1:t}|H_{c,i,1:t},O_{c,i,1:t},z_{I,i,1:t})} \\ (\log[p(O_{c,i,t}|h_{L,i,t}, z_{L,i,t})]) + E_{q(z_{I,i,1:t}|H_{opp,1:t},O_{opp,i,1:t})q(z_{L,i,1:t}|H_{c,i,t},O_{c,i,1:t},z_{I,i,1:t})} \\ \log[p(a_{i,t}|O_{opp,i,t}, O_{c,i,t}, z_{I,i,t}, z_{L,i,t})] - E_{q(z_{I,i,1:t}|H_{opp,1:t},O_{opp,i,1:t})} KL(q(z_{I,i,t}|H_{opp,t}, O_{opp,i,t})|| \\ p(z_{I,i,t}|z_{I,i,t-1}, z_{I,n_i,t-1}, a_{i,t-1}, a_{n_i,t-1})) - E_{q(z_{L,i,1:t}|H_{c,1:t},O_{c,i,1:t},z_{I,i,1:t})} \\ KL(q(z_{L,i,t}|H_{c,t}, O_{c,i,t}, z_{I,i,t})||p(z_{L,i,t}|z_{L,i,t-1}, z_{L,n_i,t-1}, a_{i,t}, a_{n_i,t-1}, z_{I,i,t})) \end{array}
\end{aligned}
\tag{20}
$$

Please refer to Appendix A.10 for the full derivation. The third term is omitted because of the joint policy. To reduce cumulative prediction error of opponents' intentions and strategies, we

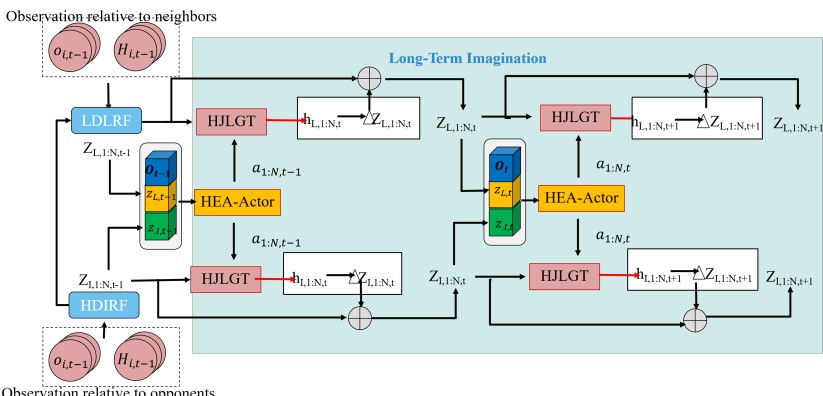

Figure 5: Inference phase.

minimize two KL divergences between the prior and posterior of the hierarchical world model. The reconstruction loss can be written compactly as:$E_{q(z_I|H_{opp},O_{opp})}(\log[p(O_{opp}|h_I, z_I)]) + E_{q(z_L|H_c,O_c,z_I)}(\log[p(O_c|h_L, z_L)])$. The priors $h_I$ and $h_L$ play dual roles: guiding trajectory prediction and shaping posterior learning via the reparameterization trick. Thus, reconstructed trajectories and posterior updates remain tightly coupled to evolving opponent mental states, unlike in a standard VAE. Because opponent intentions influence both their own trajectories and the cooperative agents' lower-level strategies, intention variables are updated through two backpropagation rounds within the hierarchical world model. Comparisons with RSSM, TSSM, and HyperHD2TSSM are given in Appendix A.5. Also, the hierarchical intention-strategy decomposition about H2TE-MITD and LHTE-MLTD, and the transition model are detailed in Appendix 3.2.1, 3.2.2, and A.6. And any-time-step update process can be found in Appendix A.7.

**Inference Process:** As shown in Figure 1, both allies and opponents use the same H2IL-MBOM and HyperHD2TSSM to estimate each other's mental states. For instance, collaborative agents model opponents using historical observations $H_{opp,t}$ and current observations $O_{opp,i,t}$, and vice versa. Here, $O_{opp,i,t} = \{O_{i,j,t}\}_{j=1,...,m}$ represents the observations relative to $m$ opponents within agent $i$'s scope, and $H_{opp,t} = \{O_{opp,i,t}\}_{t=t_0,...,t-1}^{i=1,...,N}$. The agent $i$ uses the H2TE-MITD to estimate the high-level posterior $q_{\phi_I}(z_{I,i,t}|H_{opp,t}, O_{opp,i,t})$ for multi-intent queries $z_{I,i,t}$ of opponents. It also uses a deterministic model HJLGT and a Gaussian stochastic model to approximate the high-level prior $p_{\psi_I}(z_{I,i,t}|z_{I,i,t-1}, z_{I,n_i,t-1}, a_{i,t-1}, a_{n_i,t-1})$, which reflects intent $z_{I,n_i,t-1}$ from the $n$ neighbors $n_i = \{1,..,n\}_{\neq i}$ of agent $i$, as well as the actions of those neighbors, to infer future multi-intent queries. The observation model $p_{\theta_I}(O_{opp,i,t}|z_{I,i,t}, h_{I,i,t})$ predicts opponents' trajectories, incorporating both current and historical intentions, which reveal how intentions influence trajectories. At the low-level world model, LHTE-MLTD approximates the low-level posterior $q_{\phi_L}(z_{L,i,t}|H_{c,t}, O_{c,i,t}, z_{I,i,t})$ to estimate multi-latent strategy queries $z_{L,i,t}$ from historical observations $H_{c,i,t} = \{O_{c,i,t}\}_{t=t_0,...,t-1}^{i=1,...,N}$, current observations $O_{c,i,t} = \{O_{i,l,t}\}_{l=1,...,n(l\neq i)}$, and current intent queries $z_{I,i,t}$, reflecting how intentions impact strategies. It uses a deterministic model HJLGT and a Gaussian model to approximate the low-level prior $p_{\psi_L}(z_{L,i,t}|z_{L,i,t-1}, z_{L,n_i,t-1}, a_{i,t-1}, a_{n_i,t-1}, z_{I,i,t})$ based on latent strategies $z_{L,n_i,t-1}$ from neighbors and their actions, along with predicted intent queries $z_{I,i,t}$. The observation model $p_{\theta_L}(O_{c,i,t}|z_{L,i,t}, h_{L,i,t})$ predicts the trajectories of cooperative agents based on the estimated latent strategies $z_{L,i,t}$ of opponents, revealing how these strategies influence the trajectories of cooperative agents. Once $z_{I,i,t}$ and $z_{L,i,t}$ are estimated at each step, agent $i$ can make decisions $a_{i,t} = \pi(O_{opp,i,t}, O_{c,i,t}, z_{I,i,t}, z_{L,i,t})$ and infer rewards $p_{\theta_r}(r_{i,t}|z_{I,i,t}, h_{I,i,t}, z_{L,i,t}, h_{L,i,t})$.

## A.5   HyperHD2TSSM

In the RSSM, hidden states are sequentially derived to accommodate sequential learning. By contrast, the TSSM deviates from this processing by concurrently computing each hidden state through the utilization of past states and actions, thereby facilitating parallelized training. It is important to

Table 4: Comparison of RSSM, TSSM, and HyperHD2TSSM

| | Rssm | Tssm | HyperHD2TSSM |
|---|---|---|---|
| Representation model | $z_t \sim q(z_t|h_t, O_t)$ | $z_t \sim q(z_t|O_t)$ | $z_{I,i,t} \sim q(z_{I,i,t}|H_{opp,t}, O_{opp,i,t})$, $z_{L,i,t} \sim q(z_{L,i,t}|H_{c,t}, O_{c,i,t}, z_{I,i,t})$ |
| Deterministic model | $h_{t+1} = gru(h_t, z_t, a_t)$ | $h_{t+1} = Transformer(z_{1:t}, a_{1:t})$ | $w_{I,i,t+1}, w_{I,n_i,t+1} = Hyper(z_{I,i,t}, a_{i,t}, z_{I,n_i,t}, a_{n_i,t})$, $h_{I,i,t+1} = HJLGT_{w_{I,n_i,t+1}}(z_{I,i,t}, z_{I,n_i,t}, a_{i,t}, a_{n_i,t})$ $w_{L,i,t+1}, w_{L,n_i,t+1} = Hyper(z_{L,i,t}, a_{i,t}, z_{L,n_i,t}, a_{n_i,t})$, $h_{L,i,t+1} = HJLGT_{w_{L,n_i,t+1}}(z_{L,i,t}, z_{L,n_i,t}, a_{i,t}, a_{n_i,t}, z_{I,i,t})$ |
| Stochastic model | | $\hat{z}_{t+1} \sim p(\hat{z}_{t+1}|h_{t+1})$ | $\Delta\hat{z}_{I,i,t+1} \sim p(\Delta\hat{z}_{I,i,t+1}|h_{I,i,t+1})$, $\hat{z}_{I,i,t+1} = \Delta\hat{z}_{I,i,t+1} + z_{I,i,t}$ $\Delta\hat{z}_{L,i,t+1} \sim p(\Delta\hat{z}_{L,i,t+1}|h_{L,i,t})$, $\hat{z}_{L,i,t+1} = \Delta\hat{z}_{L,i,t+1} + z_{L,i,t}$ |
| Observation model | | $p(O_{t+1}|z_{t+1}, h_{t+1})$ | $p(O_{opp,i,t+1}|z_{I,i,t+1}, h_{I,i,t+1})$, $p(O_{c,i,t+1}|z_{L,i,t+1}, h_{L,i,t+1})$ |
| Reward model | | $p(r_{t+1}|z_{t+1}, h_{t+1})$ | $p(r_{i,t+1}|z_{I,i,t+1}, h_{I,i,t+1}, z_{L,i,t+1}, h_{L,i,t+1})$ |

acknowledge, however, that as the temporal extent (T) expands, so too does the volume of requisite historical information, consequently escalating the computational demands. In our transition model design, we posit that the historical joint latent state-action of the $n$ adjacent agents is crucial, so we utilize a hypernetwork to interactively generate latent weights across agents based on the estimated state from the last step and further predict the state change at the next step. With reasoning, the latent weights at each step implicitly contain the historical information about neighbors from the beginning of reasoning to the desired time, leading to the $O(1)$ complexity. The comparison with RSSM, TSSM, and HyperHD2TSSM can be found in Table4.

Here, we utilize $HJLGT_I$, $HJLGT_L$, and a Gaussian model to approximate $p(z_{I,i,t}|z_{I,i,t-1}, z_{I,n_i,t-1}, a_{i,t-1}, a_{n_i,t-1})$ and $(z_{L,i,t}|z_{L,i,t-1}, z_{L,n_i,t-1}, a_{i,t-1}, a_{n_i,t-1}, z_{I,i,t})$. Within this framework, $w_{I,i,t}$, $w_{I,n_i,t}$ are the separate latent neural network weights generated by the hypernetwork for each agent and their corresponding neighbors, which are used for interactively estimating intentions of opponents. Similarly, $w_{L,i,t}$, $w_{L,n_i,t}$ are the neural network weights used for interactively estimating latent strategies of opponents for each agent and their corresponding neighbors. In other words, each agent updates the estimations for these mental states by considering the estimations of their neighbors, in which latent weights are adaptively adjusted based on the specific agent and inference time step, allowing for personalized and temporally sensitive representation learning. In addition, all agents within the same team share a common hierarchical world model. Through a hypernetwork, they can construct transition models HJLGT with distinct latent weights for each agent without increasing neural network parameters. This eliminates the need to for building individual decentralized world models for each agent, which is different from the centralized, shared, and decentralized world models, improving representation ability and scalability.

A.6   HJLGT

As shown in Figure 6, the HJLGT is defined as follow:

$$i.e., h_{i,t+1} \leftarrow w_{i,t+1}, w_{n_i,t+1} \leftarrow z_{i,t}, z_{n_i,t}, a_{i,t}, a_{n_i,t}$$
$$h_{i,t+1} = HJLGT_{w_{n_i,t}}(z_{i,t}, z_{n_i,t}, a_{i,t}, a_{n_i,t}):$$
$$z_{i+n_i,t} = hstack(z_{i,t}, z_{n_i,t})$$
$$w_{i,t+1}, w_{n_i,t+1} = Hyper(z_{i,t}, a_{i,t}, z_{n_i,t}, a_{n_i,t})$$
$$w_{i+n_i,t+1} = hstack(w_{i,t+1}, w_{n_i,t+1})$$
$$Q_{i,t} = z_{i+n_i,t}, K_{i,t} = \text{Tanh}(z_{i+n_i,t}@w_{i+n_i,t+1}), V_{i,t} = K_{i,t}W_i^V \qquad (21)$$
$$x = MHA(Q_{i,t}, K_{i,t}, V_{i,t})$$
$$y = Gate1(x, x)$$
$$z_{i,t} = Gate2(y, PositionWiseMlp(LayerNorm(y)))$$
$$E_{i,t} = Gate3(x, FCLayer(z_{i,t}))$$
$$h_{i,t+1} = FCLayer(Concat(E_{i,t}, x))$$

where the hstack operation involves stacking elements in a horizontal manner, MHA is the multi-head attention. It can be seen that the proposed transition model is designed for interactive prediction

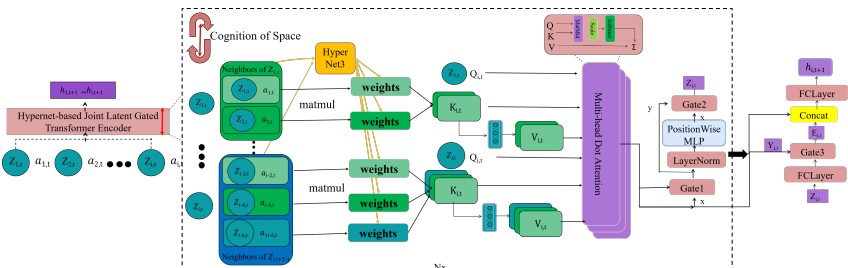

Figure 6: Architecture of the HJLGT.

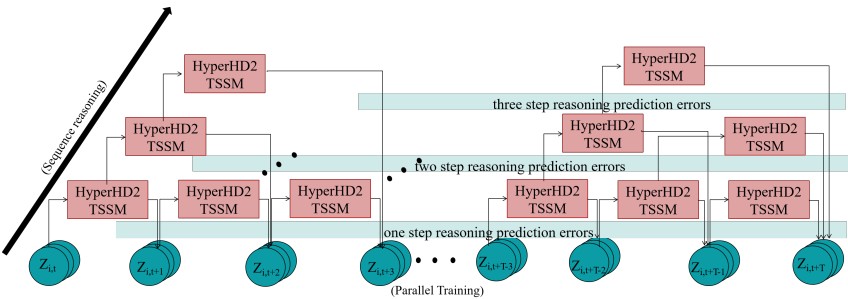

Figure 7: Any-time-step update process of model: Our approach allows for updating any state over arbitrary time intervals (with $k = 1, 2, 3$), reducing accumulative errors and eliminating the need to perform inference sequentially from the initial state to the target time step. Furthermore, multiple arbitrary-step updates applied to a single model can be interpreted as an implicit approximation of an ensemble of models with fixed horizons $k = 1$, $k = 2$, and $k = 3$, effectively reducing model complexity while enabling parallel training.

rather than independent prediction in a multi-agent system and can adaptively establish transition models for each agent without increasing model parameters, which makes it more adaptable and scalable. Most importantly, as inference progresses, each agent interactively updates its latent neural weights and estimates of mental states through continuous interaction with its neighbors.

## A.7 ANY-TIME-STEP UPDATE

As shown in Figure 7, we assume that the latent states at any given time can be inferred not only from the latent state and action at the most recent time step but also from a sequence of latent states and actions observed over the preceding interval. Given that the latent weights are capable of compressing historical information, the transition model is able to perform any-time-step updates:

$$
\begin{aligned}
J_{prior,z_I,z_L} = \min_{\phi_I,\psi_I,\phi_L,\psi_L} & \frac{1}{H}\frac{1}{T}\frac{1}{N}\sum_{k=1}^{H}\sum_{T=1}^{\infty}\sum_{t_s=t}^{t+T-k}\sum_{i=1}^{N} \begin{array}{l}(q_{\phi_I}(z_{I,i,t_s+k}|H_{opp,t_s+k},O_{opp,i,t_s+k})- \\ q_{\phi_I}(z_{I,i,t_s}|H_{opp,t_s},O_{opp,i,t_s})- \\ (p_{\psi_I}(\Delta|H_{opp,t_s},O_{opp,i,t_s},a_{i,t_s:t_s+k-1})))^2\end{array} \\
+\frac{1}{H}\frac{1}{T}\frac{1}{N}\sum_{k=1}^{H}\sum_{T=1}^{\infty}\sum_{t_s=t}^{t+T-k}\sum_{i=1}^{N} & \begin{array}{l}(q_{\phi_L}(z_{L,i,t_s+k}|H_{c,t_s+k},O_{c,i,t_s+k},z_{I,i,t_s+k})-q_{\phi_L}(z_{L,i,t_s}|H_{c,t_s},O_{c,i,t_s},z_{I,i,t_s})- \\ (p_{\psi_L}(\Delta|H_{c,t_s},O_{c,i,t_s},a_{i,t_s:t_s+k-1})))^2\end{array}
\end{aligned}
\tag{22}
$$

where $\phi_I, \psi_I, \phi_L, \psi_L$ are parameters of high-level world model and low-level world model, respectively. This approach eliminates the necessity of explicitly requiring all previous states up to time $T$, as is the case with TSSM. Additionally, it avoids the need for sequential inference from the initial state to the target time step, which is characteristic of RSSM. By enabling updates over arbitrary time intervals, our method reduces accumulative errors and computational complexity compared to these models. Moreover, since any state can be updated over arbitrary time spans, it facilitates parallel training. Furthermore, multiple arbitrary-step updates within a single model are equivalent to an implicit averaging over an ensemble of models with different horizons ($k = 1, 2, ..., H$, $H \sim$ random(maximum horizon)), thereby further reducing model complexity.

## A.8 THE DEFINITIONS OF OPERATORS

### A.8.1 HYPER OPERATOR

The Hyper operator is defined as follow:

$$
\begin{aligned}
x &= za_{i,t} = Concat(z_{i,t}, a_{i,t}); \\
w_{i,t} &= HyperNet(x; \theta_{hyper}); \\
y &= f(x; w_{i,t}) = f(x; HyperNet(x; \theta_{hyper}));
\end{aligned}
\tag{23}
$$

where we assume that the dimensions of concatenation $za_{i,t}$ of $z_{i,t}$ and $a_{i,t}$ are $[n, d_z + d_a]$. Initially, the hypernetwork with $\theta_{hyper}$ is sized as $[d_z + d_a, (d_z + d_a) \times d_h]$, and it is multiplied by $za_{i,t}$ to produce weights of size $[n, (d_z + d_a) \times d_h]$. To automate weight assignment and create a reduced neural network, $za_{i,t}$ is reshaped to $[n, 1, d_z + d_a]$ using the unsqueeze operator and weights with the size of $[n, (d_z + d_a) \times d_h]$ is reshaped to $[n, d_z + d_a, d_h]$. Finally, we multiply and activate them to obtain results while the the size of results is transformed into dimensions $[n, d_h]$. This process is denoted as $w_{i,t} = Hyper(z_{i,t}, a_{i,t})$

### A.8.2 GATE OPERATOR

The Gate operator is defined as follow:

$$
\begin{aligned}
Gate(y, x) &= (1 - z) \odot y + z \odot h; \\
z &= \sigma(W_z x + U_z y - b_g); \\
h &= \tanh(W_g x + U_g(r \odot y)); \\
r &= \sigma(W_r x + U_r y);
\end{aligned}
\tag{24}
$$

where $\odot$ is the hadamard product, which refers to the element-wise multiplication of two matrices of the same size; $\sigma$ is the sigmoid operation; the linear weights $W_z$, $U_z$, $W_g$, $U_g$, $W_r$, and $U_r$, along with the bias $b_g$, are components used in the model.

## A.9 IMPLEMENTATION OF MSOAR-PPO

## A.10 DERIVATION OF THE HIERARCHICAL VARIATIONAL LOWER BOUND

The joint probability and the hierarchical evidence lower bound (HELBO) are derived as follows:

$$
\begin{aligned}
&p(O_{opp,1:N,1:T}, O_{c,1:N,1:T}, a_{1:N,1:T}, h_{I,1:N,1:T}, z_{I,1:N,1:T}, h_{L,1:N,1:T}, z_{L,1:N,1:T}) \\
&= \prod_{t=1}^{T} \left[ \begin{matrix} p(h_{I,1:N,t}, z_{I,1:N,t}|z_{I,1:N,t-1}, a_{1:N,t-1})p(O_{opp,1:N,t}|h_{I,1:N,t}, z_{I,1:N,t}) \\ p(h_{L,1:N,t}, z_{L,1:N,t}|z_{L,1:N,t-1}, a_{1:N,t-1}, z_{I,1:N,t})p(O_{c,1:N,t}|h_{L,1:N,t}, z_{L,1:N,t}) \\ p(a_{1:N,t}|O_{opp,1:N,t}, O_{c,1:N,t}, z_{I,1:N,t}, z_{L,1:N,t}) \end{matrix} \right] \\
&= \prod_{t=1}^{T} \left[ \begin{matrix} p(z_{I,1:N,t}|h_{I,1:N,t})p(h_{I,1:N,t}|z_{I,1:N,t-1}, a_{1:N,t-1})p(O_{opp,1:N,t}|h_{I,1:N,t}, z_{I,1:N,t}) \\ p(z_{L,1:N,t}|h_{L,1:N,t})p(h_{L,1:N,t}|z_{L,1:N,t-1}, a_{1:N,t-1}, z_{I,1:N,t})p(O_{c,1:N,t}|h_{L,1:N,t}, z_{L,1:N,t}) \\ p(a_{1:N,t}|O_{opp,1:N,t}, O_{c,1:N,t}, z_{I,1:N,t}, z_{L,1:N,t}) \end{matrix} \right] \\
&= \prod_{t=1}^{T} \left[ \begin{matrix} p(z_{I,1:N,t}|z_{I,1:N,t-1}, a_{1:N,t-1})p(O_{opp,1:N,t}|h_{I,1:N,t}, z_{I,1:N,t}) \\ p(z_{L,1:N,t}|z_{L,1:N,t-1}, a_{1:N,t-1}, z_{I,1:N,t})p(O_{c,1:N,t}|h_{L,1:N,t}, z_{L,1:N,t}) \\ p(a_{1:N,t}|O_{opp,1:N,t}, O_{c,1:N,t}, z_{I,1:N,t}, z_{L,1:N,t}) \end{matrix} \right] \\
&= \prod_{t=1}^{T} \left[ \begin{matrix} p(z_{I,1,t}|z_{I,1,t-1}, z_{I,n_1,t-1}, a_{1,t-1}, a_{1,n_1,t-1})...p(z_{I,N,t}|z_{I,N,t-1}, z_{I,n_N,t-1}, a_{N,t-1}, a_{N,n_N,t-1}) \\ p(O_{opp,1,t}|h_{I,1,t}, z_{I,1,t})...p(O_{opp,N,t}|h_{I,N,t}, z_{I,N,t}) \\ p(z_{L,1,t}|z_{L,1,t-1}, z_{L,n_1,t-1}, a_{1,t-1}, a_{1,n_1,t-1}, z_{I,1,t})...p(z_{L,N,t}|z_{L,N,t-1}, z_{L,n_N,t-1}, a_{N,t-1}, \\ a_{N,n_N,t-1}, z_{I,N,t}) \\ p(O_{c,1,t}|h_{L,1,t}, z_{L,1,t})...p(O_{c,N,t}|h_{L,N,t}, z_{L,N,t}) \\ p(a_{1,t}|O_{opp,1,t}, O_{c,1,t}, z_{I,1,t}, z_{L,1,t})...p(a_{N,t}|O_{opp,N,t}, O_{c,N,t}, z_{I,N,t}, z_{L,N,t}) \end{matrix} \right] \\
&= \prod_{t=1}^{T} \prod_{i=1}^{N} \left[ \begin{matrix} p(z_{I,i,t}|z_{I,i,t-1}, z_{I,n_i,t-1}, a_{i,t-1}, a_{n_i,t-1})p(O_{opp,i,t}|h_{I,i,t}, z_{I,i,t}) \\ p(z_{L,i,t}|z_{L,i,t-1}, z_{L,n_i,t-1}, a_{i,t}, a_{n_i,t-1}, z_{I,i,t})p(O_{c,i,t}|h_{L,i,t}, z_{L,i,t}) \\ p(a_{i,t}|O_{opp,i,t}, O_{c,i,t}, z_{I,i,t}, z_{L,i,t}) \end{matrix} \right]
\end{aligned}
\tag{25}
$$

---

**Algorithm 1** MSOAR-PPO.

---

**Require:** $\leq step_{max}$, total numbers $N$, observable numbers $n$, and missile numbers $n$ of red team agents,and total numbers $M$, observable numbers $m$, and missile numbers $m$ of blue team agents;

Initialize the network parameters of H2IL-MBOM of two teams: $\{\phi_I, \psi_I, \theta_I, \theta_r, \phi_L, \psi_L, \theta_L\}$, and $\{\phi_I, \psi_I, \theta_I, \theta_r, \phi_L, \psi_L, \theta_L\}$, actor policies of two teams: $\pi_\theta$ and $\pi_\theta$, critic networks of two teams: $V_\psi$ and $V_\psi$;

Initialize the opponents' intentions $\{z_{I,i}\}_{i=1}^{N}$ reasoned by red team, and opponents' intentions $\{z_{I,j}\}_{j=1}^{M}$ reasoned by blue team;

Set learning rate $\alpha_{rl}$ of RL for red team and the learning rate $\alpha_m$ of their H2IL-MBOM, and learning rates $\alpha_{rl}, \alpha_m$ of blue team;

Initialize memory buffers $\{D_{env,t}\}_{t=1}^{T}$, $\{D_{env,t}\}_{t=1}^{T}$ and historical buffers $\{H_{opp,t}\}_{t=1}^{512}$, $\{H_{c,t}\}_{t=1}^{512}$, $\{H_{opp,t}\}_{t=1}^{512}$, $\{H_{c,t}\}_{t=1}^{512}$;

**while** $step \leq step_{max}$ **do**

    Reinitialize the environment;

  **while** not done **do**

    **for** red team agents $i = 1, ..., N$ **do**

      Obtain the current observations $O_{opp,i,t} = \{O_{i,j,t}\}_{j=1}^{2m}$ and $O_{c,i,t} = \{O_{i,l,t}\}_{l=1}^{n}$ of each agent, and gather historical observations $H_{opp,t}$ and $H_{c,t}$;

      Infer opponents' intentions $\{z_{I,i,j,t}\}_{j=1}^{2m}$ with $q(z_{I,i,t}|H_{opp,t}, O_{opp,i,t})$ by eq.equation 1-equation 5;

      Infer opponents' latent strategies $\{z_{L,i,j,t}\}_{j=1}^{2m}$ with $q(z_{L,i,t}|H_{c,t}, O_{c,i,t}, z_{I,i,t})$ by eq.equation 6-equation 8;

      Select actions according to the policy $\pi_\theta(\cdot|O_{opp,i,t}, O_{c,i,t}, z_{I,i,t}, z_{L,i,t})$ with HEA;

    **end for**

    **for** blue team agents $j = 1, ..., M$ **do**

      Obtain the current observations $O_{opp,j,t} = \{O_{j,i,t}\}_{i=1}^{2n}$ and $O_{c,j,t} = \{O_{j,l,t}\}_{l=1}^{m}$ of each agent, and gather historical observations $H_{opp,t}$ and $H_{c,t}$;

      Infer opponents' intentions $\{z_{I,j,i,t}\}_{i=1}^{2n}$ with $q(z_{I,j,t}|H_{opp,t}, O_{opp,j,t})$ by eq.equation 1-equation 5;

      Infer opponents' latent strategies $\{z_{L,j,i,t}\}_{i=1}^{2n}$ with $q(z_{L,j,t}|H_{c,t}, O_{c,j,t}, z_{I,j,t})$ by eq.equation 6-equation 8;

      Select actions according to the policy $\pi_\theta(\cdot|O_{opp,j,t}, O_{c,j,t}, z_{I,j,t}, z_{L,j,t})$ with HEA;

    **end for**

    Execution actions, and obtain rewards and next states;

    Add transitions to $D_{env} \leftarrow D_{env} \cup (O_{i,t}, a_{i,t}, r_{i,t}, O_{i,t+1}, z_{I,i,t}, z_{L,i,t})$ and $D_{env} \leftarrow D_{env} \cup (O_{i,t}, a_{i,t}, r_{i,t}, O_{i,t+1}, z_{I,i,t}, z_{L,i,t})$;

  **end while**

  Train H2IL-MBOM of both teams by eqs.20 and 25, in which $H \sim$random(maximum horizon) and $k = 1, ..., H$;

  **for** $epoch = 1$ to num-epoch **do**

    **// Update policy and critic of both teams by PPO, respectively**:

    Computer loss $J_\pi, J_c$ and $J_\pi, J_c$ of both teams from PPO;

    $\theta \leftarrow \theta + \alpha_{rl}\nabla_\theta J_\pi(O_t, z_{I,t}, z_{L,t})$;

    $\psi \leftarrow \psi - \alpha_{rl}\nabla_\psi J_c(O_t, z_{I,t}, z_{L,t})$;

    $\theta \leftarrow \theta + \alpha_{rl}\nabla_\theta J_\pi(O_t, z_{I,t}, z_{L,t})$;

    $\psi \leftarrow \psi - \alpha_{rl}\nabla_\psi J_c(O_t, z_{I,t}, z_{L,t})$;

  **end for**

  Clear up the respective memories;

**end while**

---

$$\log p(O_{opp,1:N,1:T}, O_{c,1:N,1:T}, a_{1:N,1:T}, h_{I,1:N,1:T}, z_{I,1:N,1:T}, h_{L,1:N,1:T}, z_{L,1:N,1:T})$$

$$= \log E_{q(z_{1:N,1:T}|H_{1:T},O_{1:N,1:T})} \left[ \frac{p(O_{opp,1:N,1:T}, O_{c,1:N,1:T}, a_{1:N,1:T}, h_{I,1:N,1:T}, z_{I,1:N,1:T}, h_{L,1:N,1:T}, z_{L,1:N,1:T})}{q(z_{1:N,1:T}|H_{1:T}, O_{1:N,1:T})} \right]$$

$$\geq E_{q(z_{1:N,1:T}|H_{1:T},O_{1:N,1:T})} \log \left[ \frac{p(O_{opp,1:N,1:T}, O_{c,1:N,1:T}, a_{1:N,1:T}, h_{I,1:N,1:T}, z_{I,1:N,1:T}, h_{L,1:N,1:T}, z_{L,1:N,1:T})}{q(z_{1:N,1:T}|H_{1:T}, O_{1:T})} \right]$$

$$= \int q(z_{1:N,1:T}|H_{1:T}, O_{1:N,1:T}) \log \left[ \frac{p(O_{opp,1:N,1:T}, O_{c,1:N,1:T}, a_{1:N,1:T}, h_{I,1:N,1:T}, z_{I,1:N,1:T}, h_{L,1:N,1:T}, z_{L,1:N,1:T})}{q(z_{1:N,1:T}|H_{1:T}, O_{1:T})} \right]$$
$$dz_{1:N,1:T}$$

$$= \int \sum_{t=1}^{T} \log \left[ \begin{array}{c} q(z_{I,1:N,1:T}|H_{opp,1:T}, O_{opp,1:N,1:T}) q(z_{L,1:N,1:T}|H_{c,1:T}, O_{c,1:N,1:T}, z_{I,1:N,1:T}) \\ \frac{p(z_{I,1:N,t}|z_{I,1:N,t-1}, a_{1:N,t-1}) p(O_{opp,1:N,t}|h_{I,1:N,t}, z_{I,1:N,t})}{} \\ p(z_{L,1:N,t}|z_{L,1:N,t-1}, a_{1:N,t-1}, z_{I,1:N,t}) p(O_{c,1:N,t}|h_{L,1:N,t}, z_{L,1:N,t}) \\ \frac{p(a_{1:N,t}|O_{opp,1:N,t}, O_{c,1:N,t}, z_{I,1:N,t}, z_{L,1:N,t})}{q(z_{I,1:N,t}|H_{opp,t}, O_{opp,1:N,t}) q(z_{L,1:N,t}|H_{c,t}, O_{c,1:N,t}, z_{I,1:N,t})} \end{array} \right] dz_{1:N,1:T}$$

$$= \sum_{t=1}^{T} \left\{ \begin{array}{l} \int \begin{array}{l} q(z_{I,1:N,1:t}|H_{opp,1:t}, O_{opp,1:N,1:t}) q(z_{L,1:N,1:t}|H_{c,1:t}, O_{c,1:N,1:t}, z_{I,1:N,1:t}) \\ \log[p(O_{opp,1:N,t}|h_{I,1:N,t}, z_{I,1:N,t})] \end{array} dz_{I,1:N,1:t} \\ + \int \begin{array}{l} q(z_{I,1:N,1:t}|H_{opp,1:t}, O_{opp,1:N,1:t}) q(z_{L,1:N,1:t}|H_{c,1:t}, O_{c,1:N,1:t}, z_{I,1:N,1:t}) \\ \log[p(O_{c,1:N,t}|h_{L,1:N,t}, z_{L,1:N,t})] \end{array} dz_{L,1:N,1:t} \\ + \int \begin{array}{l} q(z_{I,1:N,1:t}|H_{opp,1:t}, O_{opp,1:N,1:t}) q(z_{L,1:N,1:t}|H_{c,1:t}, O_{c,1:N,1:t}, z_{I,1:N,1:t}) \\ \log[p(a_{1:N,t}|O_{opp,1:N,t}, O_{c,1:N,t}, z_{I,1:N,t}, z_{L,1:N,t})] \end{array} dz_{1:N,1:t} \\ + \int \begin{array}{l} q(z_{I,1:N,1:t}|H_{opp,1:t}, O_{opp,1:N,1:t}) q(z_{L,1:N,1:t}|H_{c,1:t}, O_{c,1:N,1:t}, z_{I,1:N,1:t}) \\ \log \left[ \frac{p(z_{I,1:N,t}|z_{I,1:N,t-1}, a_{1:N,t-1})}{q(z_{I,1:N,t}|H_{opp,t}, O_{opp,1:N,t})} \right] \end{array} dz_{I,1:N,1:t} \\ + \int \begin{array}{l} q(z_{I,1:N,1:t}|H_{opp,1:t}, O_{opp,1:N,1:t}) q(z_{L,1:N,1:t}|H_{c,1:t}, O_{c,1:N,1:t}, z_{I,1:N,1:t}) \\ \log \left[ \frac{p(z_{L,1:N,t}|z_{L,1:N,t-1}, a_{1:N,t-1}, z_{I,1:N,t})}{q(z_{L,1:N,t}|H_{c,t}, O_{c,1:N,t}, z_{I,1:N,t})} \right] \end{array} dz_{L,1:N,1:t} \end{array} \right\}$$

$$= \sum_{t=1}^{T} \left\{ \begin{array}{l} \int q(z_{I,1:N,1:t}|H_{opp,1:t}, O_{opp,1:N,1:t}) \log[p(O_{opp,1:N,t}|h_{I,1:N,t}, z_{I,1:N,t})] dz_{I,1:N,1:t} \\ + \int q(z_{L,1:N,1:t}|H_{c,1:t}, O_{c,1:N,1:t}, z_{I,1:N,1:t}) \log[p(O_{c,1:N,t}|h_{L,1:N,t}, z_{L,1:N,t})] dz_{L,1:N,1:t} \\ + \int \begin{array}{l} q(z_{I,1:N,1:t}|H_{opp,1:t}, O_{opp,1:N,1:t}) q(z_{L,1:N,1:t}|H_{c,1:t}, O_{c,1:N,1:t}, z_{I,1:N,1:t}) \\ \log[p(a_{1:N,t}|O_{opp,1:N,t}, O_{c,1:N,t}, z_{I,1:N,t}, z_{L,1:N,t})] \end{array} dz_{1:N,1:t} \\ + \int q(z_{I,1:N,1:t}|H_{opp,1:t}, O_{opp,1:N,1:t}) \log \left[ \frac{p(z_{I,1:N,t}|z_{I,1:N,t-1}, a_{1:N,t-1})}{q(z_{I,1:N,t}|H_{opp,t}, O_{opp,1:N,t})} \right] dz_{I,1:N,1:t} \\ + \int q(z_{L,1:N,1:t}|H_{c,1:t}, O_{c,1:N,1:t}, z_{I,1:N,1:t}) \log \left[ \frac{p(z_{L,1:N,t}|z_{L,1:N,t-1}, a_{1:N,t-1}, z_{I,1:N,t})}{q(z_{L,1:N,t}|H_{c,t}, O_{c,1:N,t}, z_{I,1:N,t})} \right] dz_{L,1:N,1:t} \end{array} \right\}$$

$$= \sum_{t=1}^{T} \left\{ \begin{array}{l} \int \begin{array}{l} q(z_{I,1,1:t}|H_{opp,1:t}, O_{opp,1,1:t}) ... q(z_{I,N,1:t}|H_{opp,1:t}, O_{opp,N,1:t}) \log[p(O_{opp,1,t}|h_{I,1,t}, z_{I,1,t}) ... \\ p(O_{opp,N,t}|h_{I,N,t}, z_{I,N,t})] dz_{I,1:N,1:t} \end{array} \\ + \int \begin{array}{l} q(z_{L,1,1:t}|H_{c,1:t}, O_{c,1,1:t}, z_{I,1,1:t}) ... q(z_{L,N,1:t}|H_{c,1:t}, O_{c,N,1:t}, z_{I,N,1:t}) \log[p(O_{c,1,t}|h_{L,1,t}, \\ z_{L,1,t}) ... p(O_{c,N,t}|h_{L,N,t}, z_{L,N,t})] dz_{L,1:N,1:t} \end{array} \\ + \int \begin{array}{l} q(z_{I,1,1:t}|H_{opp,1:t}, O_{opp,1,1:t}) q(z_{L,1,1:t}|H_{c,1:t}, O_{c,1,1:t}, z_{I,1,1:t}) ... q(z_{I,N,1:t}|H_{opp,1:t}, \\ O_{opp,N,1:t}) q(z_{L,N,1:t}|H_{c,1:t}, O_{c,N,1:t}, z_{I,N,1:t}) \log[p(a_{1,t}|O_{opp,1,t}, O_{c,1,t}, z_{I,1,t}, z_{L,1,t}) ... \\ p(a_{N,t}|O_{opp,N,t}, O_{c,N,t}, z_{I,N,t}, z_{L,N,t})] dz_{1:N,1:t} \end{array} \\ + \int \left[ \begin{array}{c} q(z_{I,1,1:t}|H_{opp,1:t}, O_{opp,1,1:t}) ... q(z_{I,N,1:t}|H_{opp,1:t}, O_{opp,N,1:t}) \\ p(z_{I,1,t}|z_{I,1,t-1}, z_{I,n_1,t-1}, a_{1,t-1}, a_{n_1,t-1}) ... \\ \log \frac{p(z_{I,N,t}|z_{I,N,t-1}, z_{I,n_N,t-1}, a_{N,t-1}, a_{n_N,t-1})}{q(z_{I,1,t}|H_{opp,t}, O_{opp,1,t}) ... q(z_{I,N,t}|H_{opp,t} O_{opp,N,t})} \end{array} \right] dz_{I,1:N,1:t} \\ + \int \left[ \begin{array}{c} q(z_{L,1,1:t}|H_{c,1:t}, O_{c,1,1:t}, z_{I,1,1:t}) ... q(z_{L,N,1:t}|H_{c,1:t}, O_{c,N,1:t}, z_{I,N,1:t}) \\ p(z_{L,1,t}|z_{L,1,t-1}, z_{L,n_1,t-1}, a_{1,t-1}, a_{n_1,t-1}, z_{I,1,t}) ... \\ \log \frac{p(z_{L,N,t}|z_{L,N,t-1}, z_{L,n_N,t-1}, a_{N,t-1}, a_{n_N,t-1}, z_{I,N,t})}{q(z_{L,1,t}|H_{c,t}, O_{c,1,t}, z_{I,1,t}) ... q(z_{L,N,t}|H_{c,t}, O_{c,N,t}, z_{I,N,t})} \end{array} \right] dz_{L,1:N,1:t} \end{array} \right\}$$

$$
\begin{aligned}
&\{\int \sum_{i=1}^{N} q(z_{I,i,1:t}|H_{opp,1:t},O_{opp,i,1:t}) \log[p(O_{opp,i,t}|h_{I,i,t},z_{I,i,t})]dz_{I,i,1:t} \\
&+ \int \sum_{i=1}^{N} q(z_{L,i,1:t}|H_{c,1:t},O_{c,i,1:t},z_{I,i,1:t}) \log[p(O_{c,i,t}|h_{L,i,t},z_{L,i,t})]dz_{L,i,1:t} \\
= \sum_{t=1}^{T} &+ \int \sum_{i=1}^{N} \begin{array}{l} q(z_{I,i,1:t}|H_{opp,1:t},O_{opp,i,1:t})q(z_{L,i,1:t}|H_{c,1:t},O_{c,i,1:t},z_{I,i,1:t}) \log[p(a_{i,t}|O_{opp,i,t},O_{c,i,t}, \\ z_{I,i,t},z_{L,i,t})]dz_{i,1:t} \end{array} \\
&+ \int \sum_{i=1}^{N} q(z_{I,i,1:t}|H_{opp,1:t},O_{opp,i,1:t}) \log \left[ \frac{p(z_{I,i,t}|z_{I,i,t-1},z_{I,n_i,t-1},a_{i,t-1},a_{n_i,t-1})}{q(z_{I,i,t}|H_{opp,t},O_{opp,i,t})} \right] dz_{I,i,1:t} \\
&+ \int \sum_{i=1}^{N} q(z_{L,i,1:t}|H_{c,1:t},O_{c,i,1:t},z_{I,i,1:t}) \log \left[ \frac{p(z_{L,i,t}|z_{L,i,t-1},z_{L,n_i,t-1},a_{i,t},a_{n_i,t-1},z_{I,i,t})}{q(z_{L,i,t}|H_{c,t},O_{c,i,t},z_{I,i,t})} \right] dz_{L,i,1:t}\} \\
&\quad E_{q(z_{I,i,1:t}|H_{opp,1:t},O_{opp,i,1:t})}(\log[p(O_{opp,i,t}|h_{I,i,t},z_{I,i,t})]) + E_{q(z_{L,i,1:t}|H_{c,i,1:t},O_{c,i,1:t},z_{I,i,1:t})} \\
&\quad (\log[p(O_{c,i,t}|h_{L,i,t},z_{L,i,t})]) + E_{q(z_{I,i,1:t}|H_{opp,1:t},O_{opp,i,1:t})q(z_{L,i,1:t}|H_{c,1:t},O_{c,i,1:t},z_{I,i,1:t})} \\
= \sum_{t=1}^{T}\sum_{i=1}^{N} &\quad \log[p(a_{i,t}|O_{opp,i,t},O_{c,i,t},z_{I,i,t},z_{L,i,t})] - E_{q(z_{I,i,1:t}|H_{opp,1:t},O_{opp,i,1:t})}KL(q(z_{I,i,t}|H_{opp,t},O_{opp,i,t})|| \\
&\quad p(z_{I,i,t}|z_{I,i,t-1},z_{I,n_i,t-1},a_{i,t-1},a_{n_i,t-1})) - E_{q(z_{L,i,1:t}|H_{c,1:t},O_{c,i,1:t},z_{I,i,1:t})} \\
&\quad KL(q(z_{L,i,t}|H_{c,t},O_{c,i,t},z_{I,i,t})||p(z_{L,i,t}|z_{L,i,t-1},z_{L,n_i,t-1},a_{i,t},a_{n_i,t-1},z_{I,i,t}))
\end{aligned}
\tag{26}
$$

## A.11  ANALYSIS OF OPPONENTS' MULTIPLE INTENTIONS AND LATENT STRATEGIES

We analyze cumulative prediction errors of opponent intentions and strategies for two opposing teams, along with the t-SNE Van der Maaten & Hinton (2008) distributions of each agent's mental-state representations over three episode segments ($\leq$500 steps, 1500–2500 steps, 5000–6000 steps). As shown in Figures. 8a and 8b, both teams rapidly infer opponent mental states and exhibit a striking pattern: after reward convergence, sharp error drops occur at 6, 8.1, 11, and 12.5 M steps, signaling sudden recognition of key features. This stems from a prediction challenge in the early stage: the model initially struggles to infer opponent states and is prone to local minima. As training progresses and strategies converge, the explored state space gradually narrows, enabling more stable feature extraction and improved predictions.

The t-SNE visualizations in Figures. 8c and 8d reveal several notable patterns. Using agent 0's predictions as an example, opponent intentions form multiple continuous strip-like distributions across three stages rather than discrete clusters, while predicted strategies remain separable within each stage. This indicates that H2IL effectively captures features of opponents' mental states. Within smaller time intervals, the reduced representations preserve a sequential structure, reflecting the temporal coherence of mental states. The multiple distributions align with key tactical phases (e.g., nose-to-nose approach, tailing, evasion, missile launch), highlighting both diversity and smooth transitions of intentions and strategies. Across the three stages, opponents exhibit 11, 7, and 3 intention transitions, whereas low-level strategies vary more smoothly. Appendix A.16 further shows that the average predicted intention changes (3, 2, 1 per UAV across the three stages) match the actual intention changes. Overall, our method provides both global prediction of opponents' intentions and fine-grained tracking of evolving latent strategies, enhancing interpretability.

## A.12  TESTING RESULTS

The win rate (WR) and survival rate (SR) are as evaluation metrics. We first confront the opponents who adopt the baseline strategy that includes straight fly, rectangular trajectory maneuver and evasion of missiles, and pursuing the tail of our aircraft. The results show that our SR is the highest and achieves a 100% WR in 4 vs. 4 scenarios as presented in Table 5. We then test the effectiveness of our method against our method and our method against MAPPO under different numbers of agents as presented in Table 6 and 7. We use SR to evaluate performance because a group with fewer agents may sacrifice less or equal to the other group. In most cases, both teams make equal sacrifices because of the same reasoning ability of both teams, and in a small number of cases (e.g., 4 vs. 6, 4 vs. 8, 6 vs. 8) where the quantity is at a disadvantage, the red team still destroys one more aircraft than the blue team. In situations where the red team has a numerical advantage, it can achieve 100% superiority (e.g., 8 vs. 4, 10 vs. 4, 10 vs. 6). Additionally, the advantage ranges are further expanded when our method against MAPPO. (e.g., 4 vs.4, 6, 8,10; 6 vs. 4, 8,10; 8 vs. 4; 10 vs. 4, 10 vs. 6). The results also demonstrate our method is endowed with good generalization ability. Due to the fixed dimensions of other MARLs, it is not possible to complete adversarial tasks in different quantities.

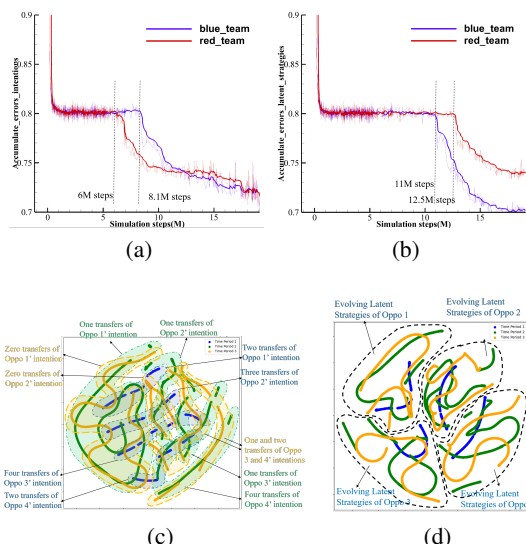

Figure 8: Accumulate error and t-SNE distribution of opponents' multiple intentions and latent strategies reasoned by Agent0 across three time periods. The total number of intention transitions observed for all opponents across various stages is 11, 7, and 3, respectively. In contrast, the low-level strategies employed by the opponents exhibit a more consistent and distinguishable performance.

As shown in Figure 8(c), the relevant opponent modeling methods are unable to complete this task, so there is no adversarial testing with these methods.

Table 5: The results of our method against the baseline strategy in 4 vs. 4 scenarios.

| SR(WR) | Straight fly | Maneuver | Pursue |
|---|---|---|---|
| Ours | 4:1(100 %) | 4:2(100 %) | 4:0(100 %) |

Table 6: The results of the confrontation of different number agents of our method.

Table 7: The results of our method vs. MAPPO under different numbers of agents.

| SR (Ours vs. Ours) | 4 | 6 | 8 | 10 | SR (Ours vs. MAPPO) | 4 | 6 | 8 | 10 |
|---|---|---|---|---|---|---|---|---|---|
| 4 | 2:2 | 3:4 | 3:6 | 3:9 | 4 | 3:2 | 3:4 | 3:6 | 3:8 |
| 6 | 3:1 | 3:3 | 3:4 | 3:7 | 6 | 3:0 | 3:3 | 3:4 | 3:6 |
| 8 | 8:0 | 3:1 | 3:3 | 3:5 | 8 | 8:0 | 3:1 | 3:3 | 3:5 |
| 10 | 10:0 | 10:0 | 3:1 | 3:3 | 10 | 10:0 | 10:0 | 3:1 | 3:3 |

### A.13 EXPERIMENTS ABOUT HYPERPARAMETERS

We vary the dimensions of intentions from 4 to 64 and evaluate the impact of different dimensions on the performance of our method, as shown in Figure 9a. We observe that there is an optimal dimensions of intentions, 8, which maximizes the performance of the model. When the dimension of intentions is below 8 or above 32, it takes twice the time to converge, and the convergence speed is significantly reduced. Based on our experiments, the optimal number of attention heads is 8. At this optimal number, the model achieves the highest performance with lowest complexity.

Similarly, we vary the number of attention heads from 2 to 16 and measure the performance using the average rewards. As shown in Figure 1 9b, we observe that there is an optimal number of attention heads, 4, which maximizes the performance of the model. When the number of attention heads is below 4 or above 8, it also takes twice the time to converge, and the convergence speed is significantly

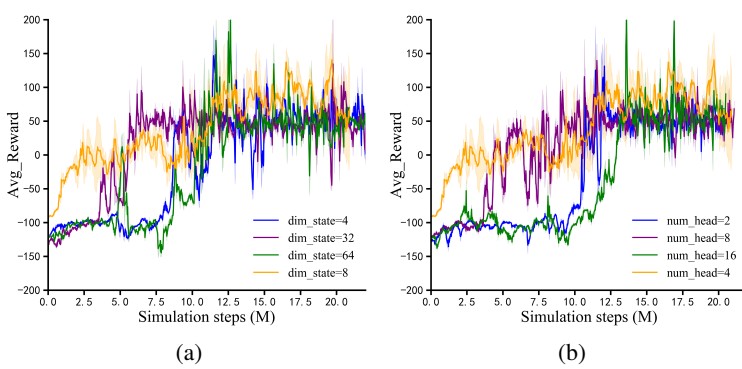

Figure 9: The results on experiments with different hyperparameters: a) different dimensions of the mental states; b) different numbers of attention head

reduced. Based on our experiments, the optimal number of attention heads is 4. At this optimal number, the model achieves the highest performance with lowest complexity.

In summary, the dimensions of the intention space and numbers of attention head are chosen based on the best balance between performance and computational efficiency.

## A.14 HYPERPARAMETERS

The hyperparameters are summarized in Tables 8–12.

Table 8: Hyperparameters of Ours

| Parameter | Value |
|---|---|
| Interaction steps | $2 \times 10^7$ (20M) |
| Training steps | $1.58 \times 10^5$ |
| Learning rate | $3 \times 10^{-4}$ |
| Discount factor | 0.99 |
| Policy initialization | Xavier uniform |
| Optimizer | Adam |
| Gradient norm clipping | 5.0 |
| Rollout Length | 128 |
| Batch size | 1024 |
| Number of training epochs | 1 |
| Number of head | 4 |
| Attention size | 32 |
| Hidden state dimensions | 128 |

Table 9: Hyperparameters of world models

| Parameter | Value |
|---|---|
| Training steps | $1.58 \times 10^5$ |
| Learning rate | $1 \times 10^{-4}$ |
| Discount factor | 0.99 |
| Optimizer | Adam |
| Gradient norm clipping | 5.0 |
| Number of head | 4 |
| Attention size | 32 |
| Intention $z_I$ and latent strategy $z_L$ dimensionality | 8 |
| Hidden state dimension | 32 |
| Number of layers ($N_M$ and $N_H$) | 4 |

Table 10: Hyperparameters of HAPPO and MAPPO

| Parameter | Value |
|---|---|
| Interaction steps | $2 \times 10^7$ (20M) |
| Training steps | $1.58 \times 10^5$ |
| Learning rate | $2 \times 10^{-4}$ |
| Discount factor | 0.99 |
| Policy initialization | Xavier uniform |
| Optimizer | Adam |
| Gradient norm clipping | 10.0 |
| PPO epoch | 5 |
| Rollout threads | 20 |
| Episode length | 1500 |

Table 11: Hyperparameters of HADDPG and MADDPG

| Parameter | Value |
|---|---|
| Interaction steps | $2 \times 10^7$ (20M) |
| Learning rate | $1 \times 10^{-4}$ |
| Discount factor | 0.99 |
| Optimizer | Adam |
| Gradient norm clipping | 5.0 |
| Buffer size | $1 \times 10^6$ |
| Batch size | 1000 |
| Rollout threads | 20 |
| Hidden state dimension | 128 |

Table 12: Hyperparameters of opponent modeling baselines

| Parameter | Value |
|---|---|
| Interaction steps | $2 \times 10^7$ (20M) |
| Learning rate | $3 \times 10^{-4}$ |
| Discount factor | 0.99 |
| Optimizer | Adam |
| Gradient norm clipping | 5.0 |
| Buffer size | $1 \times 10^6$ |
| Episode length | 1500 |
| Batch size | 3000 |
| Rollout threads | 20 |
| Hidden state dimension | 256 |

### A.15 COMPUTE RESOURCE

In our study, we performed simulations utilizing 36 parallel environments on a computer workstation equipped with dual Intel(R) Xeon(R) 40-core CPUs, 128 GB of RAM, and two NVIDIA RTX A4500 GPUs. Each environment completed 1500 maximum steps per episode at a simulation frequency of 60Hz. In total, there were roughly four days for training the uav-game environment.

### A.16 VISUAL RESULTS

We first visualize the engagement scenarios using the full reward function under different adversarial settings: ours (red) vs. MAPPO (blue) and ours (red) vs. ours (blue). Notably, MAPPO is treated as an unseen opponent during testing, as it was not encountered during the training of our red agent. We then present visualizations of our method without height-correlated reward to analyze their impact on the learned policies.

#### A.16.1 TACTICAL BEHAVIOR VISUALIZATION USING COMPLETE REWARD SETTINGS

As shown in Figures 10 and 11, the visualization of scenarios depicting engagements between our method and MAPPO, as well as engagements between our method and itself, was conducted. The figures illustrate that during combat with MAPPO, our maneuver decisions were more agile and rapid, resulting in achieving a high altitude and angle advantage with a smaller flight radius, ultimately leading to a SR of 3:1. In confrontations with our own method, both sides exhibited similar reasoning capabilities, leading to primarily engaging in double loop motion, which represents a classic tactic in close-range aerial combat.

Combining Figures 10,8c, and 8d, in the initial stage, the feature distribution range is relatively small, indicating both teams frequently make rapid maneuver transitions in a small space (such as climbing, making large turns to enter angles, and engaging in single-loop maneuvers). In the middle stage, both teams enter the engagement phase, conducting double-loop maneuvers (nose-to-nose approach and departure), and missile launches within a larger range. In the final stage, only alive agents engage in extensive pursuit and escape strategies. This is consistent with the average number of changes in the opponent's intention predicted by each UAV on average across three stages.

As shown in Figure 11, in the initial phase, the red team launches missiles first and rapidly dives downward at an airspeed of Mach 0.73 to gain kinetic energy. Afterward, it quickly climbs and performs a turn. During this phase, one blue aircraft is shot down, while the remaining blue aircraft evade the attack by diving and executing counterclockwise yaw maneuvers.

At this point, the red formation positions itself behind the blue formation, gaining a tactical advantage. The red team then accelerates and launches a second missile. In response to the incoming threat, the blue formation performs a rapid 180° counterclockwise turn to evade the second wave of attack.

The red formation maintains high maneuverability at Mach 0.87, achieves angular superiority for the second time, and launches a third missile. The blue formation again executes a swift 180° counterclockwise turn to avoid the third wave of attack.

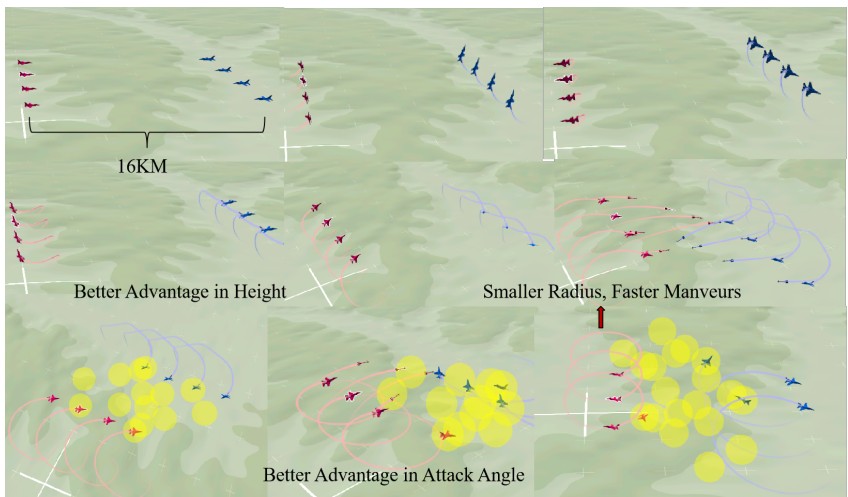

Figure 10: Snapshot of our method vs. MAPPO.

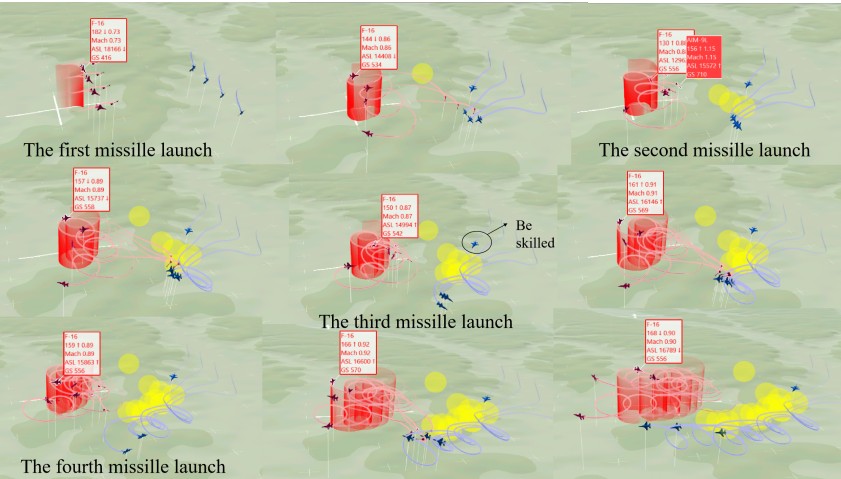

Figure 11: Snapshot of ours vs. ours.

While the blue team is turning to evade the missile, the red formation simultaneously performs aggressive turning maneuvers at Mach 0.92. This ensures that as soon as the blue aircraft complete their evasion, the red aircraft are already in a favorable angular position to launch the fourth missile.

Throughout the engagement, both teams perform turning maneuvers near their respective initial positions. The red formation is accompanied by diving and climbing movements, whereas the blue formation generally descends while maneuvering counterclockwise. Importantly, the red team consistently maintains angular superiority throughout the entire engagement.

### A.16.2 NO HEIGHT-CORRELATED REWARD

We remove height-correlated reward components and visualize the maneuvering policies and trajectories of both agents, as shown in the Figure 12.

In the initial phase, the red agent rapidly yaws to the right at Mach 0.94 and launches a missile. In response, the blue agent climbs quickly without access to height-correlated rewards; however, one of its aircraft is shot down during this phase. Chai2023A To pursue a joint advantage in range and angle-again without relying on height-correlated rewards, the red agent also initiates a rapid climb while maintaining proximity to the tail of the blue agent.

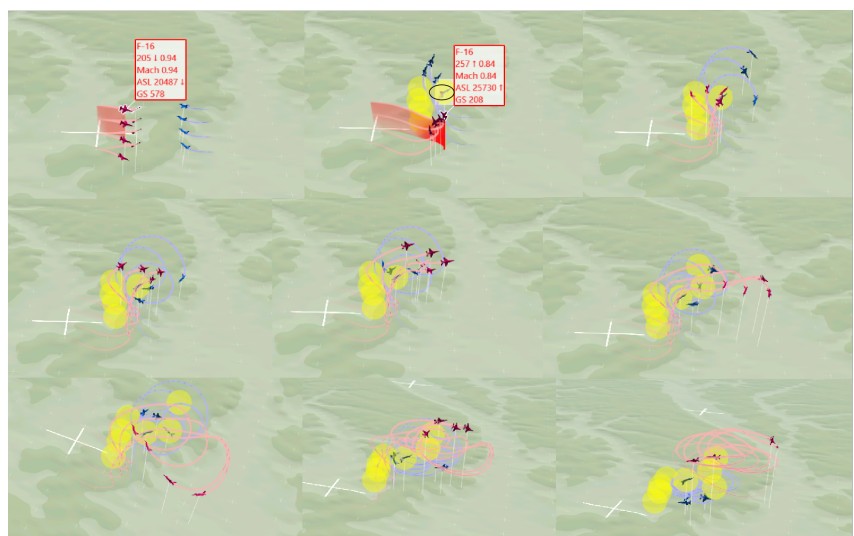

Figure 12: Maneuvering strategies without height-correlated reward.

Subsequently, in an attempt to reverse the joint range-angle advantage, the blue agent dives after climbing and performs a right yaw maneuver. The red agent promptly launches another missile, blocks the blue agent's climb, increases the distance, and then yaws to the right. Notably, each red aircraft exhibits a distinct pull-away distance and turning radius: those with shorter pull-away distances execute tighter turns, while those with longer distances perform wider turns. As a result, the red formation flies in a head-to-tail configuration.

Given that the blue agent attempts to gain angular advantage by diving and yawing to the right after climbing, the red agent responds with timely maneuvers and downward missile deployment. This forces the blue agent to perform tight turns for evasion, resulting in a disadvantage in altitude.

Importantly, by delaying the turn until after increasing the distance, the red formation avoids being exploited by the blue agent's fast, small-radius maneuvering. Moreover, the diversity in turning radii ensures that not all red aircraft fall into an angular disadvantage simultaneously. In contrast, the blue agent ends up in a clear angular disadvantage.

It's noting that despite the absence of height-based rewards, agents can still adopt strategies such as climbing to indirectly achieve a combined advantage in position and angular through position-angle rewards. Overall, the red agent achieves a favorable combined advantage in altitude, position, and attack angle by the end of the engagement.

