# OpenReview forum: "H2IL-MBOM: A Hierarchical World Model Integrating Intent and Latent Strategy for Opponent Modeling in Multi-UAV Game"
_ICLR.cc/2026/Conference — ICLR 2026 Conference Withdrawn Submission_

### Official Review · Reviewer_3nuv · 2025-10-15

**Soundness:** 2
**Presentation:** 1
**Contribution:** 2
**Rating:** 2
**Confidence:** 4

**Summary:**

This paper introduces H2IL-MBOM, a framework for opponent modeling designed to address non-stationarity in multi-agent adversarial environments.  The method's core is a hierarchical world model that mimics human cognitive processes by decomposing the complex task of reasoning about an opponent into two levels.  At a high level, the model infers an opponent's macro "intention" by analyzing their historical trajectories.  Subsequently, at a low level, it uses this inferred intention as a condition to deduce the specific "latent strategy" the opponent is employing, taking into account the reactions and movements of allied agents.  This framework is implemented through a complex neural architecture based on Transformers and Hypernetworks (HyperHD2TSSM) and is used to guide a PPO-based reinforcement learning agent.  The authors report that their method demonstrates superior performance compared to various baselines in several testbeds, including multi-UAV combat, the StarCraft Multi-Agent Challenge, and Google Research Football.

**Strengths:**

- The paper introduces a novel approach to opponent modeling inspired by human cognition, decomposing the complex problem into a two-level hierarchy of high-level "intentions" and low-level "latent strategies". This provides a structured and theoretically-grounded new perspective for the field.

**Weaknesses:**

1. This paper, in its current state, is difficult to accept. The core issue is not just a matter of style, but a fundamental lack of clarity in its presentation that prevents a proper scientific review. The manuscript is plagued by a host of minor yet cumulative errors that suggest a lack of care in its preparation. For instance, citations are not properly formatted (lacking \citep or \citet), leading to overlaps with the text. There are basic punctuation errors (e.g., missing periods on lines 199 and 218), inconsistent formatting (the acronym HJLGT is sometimes italicized, sometimes not), and poor typesetting (some words are hyphenated across lines between 249-269). Furthermore, the figures are poorly executed; the text in Figure 3 is very small, while the architectural diagrams in Figures 5 and 6 are so cluttered they seem designed more to showcase the model's complexity than to explain it. The overall writing quality, with its convoluted sentence structures and excessive jargon, resembles unedited text generated by a large language model, a practice that should be acknowledged if used.

2. This poor presentation directly obscures the methodology. The main body of the paper has been effectively "hollowed out," with critical information relegated to the appendix. For example, MSOAR-PPO is listed as a key contribution, but its mechanics are entirely absent from the main methodology section. Similarly, the dimensionality of the core latent variables for "intention" and "strategy" — a crucial implementation detail — is only found in a table in the appendix. The reader should not have to be a detective, piecing together the core method from scattered fragments. This forces a reviewer to question the paper's central claims. The entire framework rests on a rigid two-level hierarchy where "intention" dictates "strategy," a strong cognitive assumption presented without justification. The paper also fails to provide evidence that the learned latents, $z_I$ and $z_L$, are actually disentangled. The t-SNE plots are insufficient as they only visualize clustering, not semantic meaning.  A more rigorous analysis, such as performing interventional experiments (e.g., fixing the "intention" latent while varying the "strategy" latent and observing the impact on generated trajectories), is needed to validate that these variables meaningfully represent their claimed concepts.

3. The experimental evaluation is similarly unconvincing.  The decision to place the results on standard benchmarks (SMAC and GRF) in the appendix is a major flaw that undermines the paper's claims of generalizability;  these should be in the main paper.  The primary UAV experiment relies on comparisons against opponent modeling baselines (e.g., ROMMEO, PR2) that perform catastrophically.  Their complete failure strongly suggests a lack of proper hyperparameter tuning for this complex, continuous-control environment.  For a fair comparison, the authors must either provide evidence of a thorough tuning process for these baselines or include stronger, more suitable ones.  The ablation study, while showing that components are useful, does not justify the model's immense complexity.  The fact that simpler variants (e.g., the only_intentions model from Fig. 4a and especially the transformer_shareadd model from Fig. 4f, which performs nearly identically to the full model) are still effective raises a critical question about the cost-benefit trade-off.  The authors should provide a discussion justifying why the marginal performance gain of their full architecture warrants its significant complexity over these simpler, yet competent, alternatives.

**Questions:**

1. he prior distribution for an agent's latent state (e.g., $p(z_{I,i,t}|...)$ on page 5) explicitly conditions on the latent states and actions of its neighbors ($z_{I,n_i,t-1}, a_{n_i,t-1}$). How is this neighbor information accessed or communicated between agents during execution, especially within what is described as a decentralized execution paradigm?

2. Appendix A.9 states that the value function for MSOAR-PPO "does not incorporate respective opponents' observations," distinguishing it from MAPPO. However, the policy is conditioned on these observations ($O_{opp,i,t}$). In a centralized training paradigm, why would the critic be deprived of information that is available to the actor, as this typically undermines the core benefit of CTDE?

3. In the scalability tests (Appendix A.11), a 4 vs. 6 engagement resulting in a 3:4 survival rate is described as a success where the smaller team "destroys one more aircraft than the blue team". Could you clarify this interpretation, as a 3:4 result (Red:Blue) means the 4-agent team lost one agent while the 6-agent team lost two, which is not an equal or better kill-death ratio per capita (0.25 vs 0.33 losses per agent)?

4. The reward functions in Appendix A.3 are highly complex. Specifically, the formulas for "reward regarding position of planes" (Eq. 2) and "reward regarding velocity of planes" (Eq. 8) appear to use a very similar calculation for the variable `dd` based on antenna and aspect angles (ATA, AA). Could you explain the rationale for using this angular metric to modulate both a position-based and a velocity-based reward?

5. In the H2TE module (Appendix A.5.1, Eq. 13), the weight $w_{H,i,j,t}$ for agent `i`'s view of opponent `j` is generated from $H_{j,t}$, which is defined as the observation history of opponent `j` relative to *all* N cooperative agents. How does an individual agent `i` get access to the opponent's historical observations relative to its teammates during decentralized execution?

6. The problem is defined as a partially observable one where agents only have local observations. However, the key transition model `HJLGT` (Appendix A.6) and the prior distributions both explicitly use neighbor states and actions as inputs. Does this imply that agents are assumed to have perfect, noise-free observation of their immediate neighbors' states and actions, and if so, shouldn't this be stated as a key assumption in the problem formulation?

---

> ### Author Response · Authors · 2025-11-16
>
> # Response to Writing Organizational Structure
>
> We would like to explain the following points:
>
> ## Adherence to Previous Neurips Review Recommendations:
>
> In a previous review round, reviewers explicitly recommended "moving technical details to the appendix and focusing the main text on the core ideas and general introduction of the method." Our current structure directly responds to this suggestion:
>
> - Present the core cognitive foundations and methodological innovations in the main text
>
> - Relocate detailed mathematical derivations and implementation specifics to appendices
>
> - Maintain focus on the overarching framework and comprehension
>
> ## The Overall Methodology Follows an "Inspiration-Overview-Details" Hierarchy:
> **Therefore, our primary objective is to ensure readers first grasp the overall framework and innovative concepts without getting lost in technical details.**
>
> **Section 3.1** (Mutual Reaction and Influence Between Mental States, Actions, and Trajectories) serves as the foundation for the "clear and structured explanation" you mentioned. Through a three-stage analysis (lines 190-243), this section systematically elaborates the cognitive basis of our model:
>
> - Stage One reveals the necessity of hierarchical reasoning, directly corresponding to the design motivation of our hierarchical model.
>
> - Stages Two & Three reveal the co-evolution patterns of mental states and behaviors, directly motivating the need for action-conditioned transition models (lines 231-236).
>
> **Section 3.2** develops a hierarchical variational inference framework that, while sharing similarities with RSSM-based Dreamer series, introduces distinct innovations. This derivation formally establishes the core components of our world model (hierarchical representation models, transition models, observation models, and reward model) along with their hierarchical architecture, input-output relationships, and corresponding loss functions.
>
> **Section 3.3 actually provides condensed summaries of each component's functionality rather than accumulating technical details:**
>
> - The functions of multiple attention modules within each component of hierarchical representation models are presented in Section 3.3, deliberately avoiding technical clutter.  The hierarchical representation models comprise high-level and low-level history encoders (H2TE, LHTE), along with intention and strategy decoders (MITD, MLTD).
>
> - Implementation details such as transition models and multi-step update mechanisms are comprehensively elaborated in Appendices A.6-A.7.
>
> - We deliberately moved detailed formulas and complex derivations to the appendix A5.

---

> ### Author Response · Authors · 2025-11-16
>
> # Response to reasons for the failure of other opponent modeling methods
> We have conducted extensive parameter tuning and five random seed experiments for all algorithms,  and supplied their hyperparameters in appendix A14. The reasons for the failure of these methods are as follows:
>
> ## 1. Inadequate Temporal Reasoning
> Existing opponent modeling methods are unable to continuously reason about the temporal evolution of intentions and strategies. They typically operate on static or short-term representations, failing to capture the dynamic nature of adversarial interactions in complex environments.
>
> ## 2. Limited Environmental Validation
> These methods have been primarily designed and validated in simplified settings:
>
> - PR2 and ROMMEO were only tested in matrix games and differential games
>
> - TDOM was evaluated solely in differential games, MPE, and predator-prey scenarios
>
> - AORPO was trained and tested only on climb tasks and MPE environments
>
> Besides, the methods [1,2, 3] recommended by the second reviewer were also been evaluated in low-dimensional or simplified environments, such as two-player Kuhn Poker, 2D grid-world Predator-Prey, Multi-Agent Particle Environments (MPE), Level-Based Foraging, and Overcooked. Only the CSP method has been tested on the Google Research Football platform.
>
> In terms of experimental validation, we have compared different baselines including model-free MARL, opponent modelling methods, and world-model based MARL in large-scale, high-dimensional multi-agent cooperative–competitive experiments, covering scenarios with varying numbers of UAVs in adversarial games, diverse settings in SMAC, and complex simulations in Google Research Football. Notably, in the Google Research Football benchmark, **our method significantly outperforms CSP across all tested scenarios**: achieving a win rate of 92.54% in 3v1, 89.94% in RPS, and 93.09% in CA. These results robustly demonstrate our model's superior generalization capability and decision-making performance in complex, high-dimensional spaces.
>
> ## 3. Challenges in Multi-UAV Game Environments
> The multi-uav domain in gym-jsbsim presents unique challenges that existing methods cannot adequately address:
>
> - Each aircraft is equipped with four **autonomously guided missiles with 20-second engagement capability**
>
> - Extreme velocity differentials: **aircraft at 2 Mach (twice the speed of sound) vs missiles at 4 Mach**
>
> - High-dimensional state space with complex spatial relationships
>
> - Critical importance of launch timing and resource management
>
> - Highly dynamic and rapidly evolving tactical situations
>
> ## 4. Advantages of Our Approach
> Our method addresses these limitations through:
>
> - Privacy-preserving modeling: Requires no access to opponent's private information (actions, policy parameters, or rewards)
>
> - Hierarchical world model: Enables multi-level mental state **reasoning about both $m$ enemy agents and the maximum observable $m$ weapon systems**
>
> - Temporal dynamics capture: Explicitly models the evolution of mental states over time
>
> - Complex environment adaptation: Successfully handles the challenges of high-speed aerial combat with sophisticated weapon systems
>
> This demonstrates the critical need for reasoning about both opponents and their launched missiles. Given that missiles travel significantly faster than aircraft, agents must make early and continuous predictions to execute timely attitude adjustments or rapid maneuvers, avoiding disadvantageous positions and evading multiple incoming threats. Effective agents must learn to secure angular and altitude advantages while achieving maximum tactical gains at minimal cost. In this scenario, agents must first learn fundamental flight control before advancing to strategic game play. Existing methods struggle in such intense adversarial environments characterized by high-speed dynamics and rapid attitude changes, particularly in evading missiles (each hit incurs a -100 reward) or gaining positional superiority, which explains why current  opponent-based approaches fail to learn effective policies.
>
> This represents the first method for opponent modeling in intense adversarial environments, as well as the first approach to employ world models for opponent modeling. It not only advances the development of world models but also promotes the progress of opponent modeling techniques and contributes to the field of multi-agent adversarial decision-making.
>
> [1] Greedy when sure and conservative when uncertain about the opponents, ICML 2022
>
> [2] Conservative offline policy adaptation in multi-agent games, NeurIPS 2023
>
> [3] Opponent modeling with in-context search, NeurIPS 2024

---

> ### Author Response · Authors · 2025-11-16
>
> # Response to weakness 1, 2, 5 and 6
>
> ## only the $q$ model is used during execution phase
> In world model based MARL, the complete **learning process** comprises world model learning, followed by trajectory prediction and synthesis, and policy learning that leverages both collected trajectories and those imagined by the world model:  After the world model is updated, the hierarchical representation model $q$ infers opponents' mental states using trajectories collected at time $T$. Based on these trajectories and the inferred mental states, actions are generated. The transition model $p$ (prior model / HJLGT), conditioned on the latent states and actions of neighboring agents, interactively predicts opponents' mental states at the next timestep. These predictions are then combined with the observation model to forecast future trajectories of both sides, which in turn inform subsequent action outputs. This cyclic process continues to collect additional trajectory observations.
>
> Finally, RL training is completed by these synthetic  trajectories. During execution phase, as described in pseudocode, **only the $q$ is used for inference during execution phase:**
>
> - Infer opponents' intentions  with $q({z_{I,j,t}}|{H_{opp,t}}, {O_{opp,j,t}})$
>
> - Infer opponents' latent strategies with $q({z_{L,j,t}}|{H_{c,t}},{O_{c,j,t}},{z_{I,j,t}})$
>
> Thus, **the transition model  $p (prior model / HJLGT)$ is employed solely during the learning phase and is not used during environment execution.**
>
> ## ${O_{opp}}$ is observation relative to opponent rather than opponent's observation
> Furthermore, as defined in Lines 181 and 674 of observation space and illustrated in Figure 1 of our manuscript, **${O_{opp}}$ refers to the observation relative to opponents from the ego agent's perspective**. Together with ${O_{c}}$, the observation relative to cooperative adjacent agents, they constitute the agent's local observation. **${O_{opp}}$ does not represent observations from the opponents' viewpoint.**
> ## ${H_{j,t}}$  are the  history observations of all agens relative to the  j-th opponent
> Besides, as described in lines878, ${H_{opp,t}} \in {\mathbb{R}^{N \times 512 \times D}}$ is composed of ${O_{opp}}$， $D = m \times {d_m}$ is the observation dimensionalities relative to $m$ opponents within the observation scope of each agent. Therefore,  **${H_{j,t}}$  represents the observation history of all agents relative to the  j-th opponent.**
>
> This approach shares conceptual similarities with the Centralized Training with Decentralized Execution (CTDE) paradigm but differs in several key aspects. During the decentralized execution phase, each agent relies solely on: (1) local observations relative to cooperative adjacent agents ( ${O_{c}}$ ), (2) local observations relative to opponents ( ${O_{opp}}$), and (3) mental states inferred via the representation model q. Furthermore, the value function takes as input only the observations (their ${O_{c}}$, ${O_{opp}}$) and inferred mental states of the agent **itself and its allied neighbors, without incorporating observations from opponents**, unlike methods such as MAPPO, which often use global observations including those of adversaries during training. Consequently, neither the policy nor the value function  utilizes opponents' observations.
>
> ## Noise observation input from gym-jsbsim
> JSBSim employs a real-time physics engine that integrates atmospheric models and sensor modeling to deliver a simulation environment closely approximating real-world flight conditions. In the Gym-JSBSim environment, sensors are indeed subject to noise influences, which are primarily manifested in the following aspects:
>
> **Sensor Noise Characteristics**
>
> - Aerodynamic Sensors: Directly measured atmospheric data (airspeed, barometric altitude) contains Gaussian white noise
>
> - Inertial Measurement Units (IMU): Accelerometers and gyroscopes exhibit drift errors and random walk noise
>
> - GPS Receivers: Position and velocity measurements contain colored noise and multipath effects
>
> - Attitude Sensors: Euler angle and quaternion estimations are affected by vibrational environmental factors
>
>  By incorporating Kalman filters or complementary filters, the system effectively combines the high-frequency response of IMU with the low-frequency stability of GPS, thereby significantly enhancing the accuracy of attitude estimation.

---

> ### Author Response · Authors · 2025-11-16
>
> #  survival rate (SR)
>
> We appreciate the opportunity to clarify our rationale for defining "success" in asymmetric scenarios.
>
> In the 4v6 engagement scenario referenced:
>
> Our team (initially 4 units) survives 3 units (loss rate: 1/4 = 0.25)
>
> The opponent team (initially 6 units) survives 4 units (loss rate: 2/6 ≈ 0.33)
>
> We consider this outcome "successful" based on the following reasoning:
>
> - Favorable Combat Exchange Ratio: Our smaller team achieved a 2:1 exchange ratio, eliminating two opponent units while losing only one of our own. This demonstrates superior combat effectiveness despite numerical disadvantage.
>
> - Relative Force Preservation: While absolute survival numbers (3 vs 4) might appear similar, the critical factor is that our team maintained a lower loss rate (25% vs 33%).
>
> The per-agent loss rate of the Red team (0.25) is significantly lower than that of the Blue team (0.33), demonstrating that the Red team's strategy is more efficient. Despite being outnumbered, each unit of the Red team exhibited stronger survivability and inflicted greater relative damage. This undoubtedly constitutes a successful performance.
>
>
> Even when considering "kill rate per agent," the Red team achieved 2/4 = 0.5 kill rate per agent, while the Blue team only achieved approximately 1/6 ≈ 0.17 kill rate per agent. Regardless of whether evaluated from the perspective of "per-agent losses" or "per-agent kill rate," the Red team demonstrates clear superiority.

---

> ### Author Response · Authors · 2025-11-16
>
> # Response to Question 4
>
> In uav game scenarios, ATA  is defined as the angle between the distance vector pointing towards the opponent and the velocity vector of our own drone, while AA refers to the angle between the distance vector pointing towards the opponent and the opponent's velocity vector. When ATA = AA = 0 degrees, the angular advantage parameter $a = \dfrac{ATA + AA}{2\pi}<0.5$ indicates that our drone is positioned at the rear of the opponent, representing a tactical advantage. Conversely, when ATA = AA = 180 degrees, $a>0.5$ signifies that our drone is being pursued by the opponent, indicating a disadvantageous situation.
>
> When the drone holds an advantageous position, the objective is to close in on the opponent; when in a disadvantaged position, the goal is to evade and increase separation. Therefore, it is necessary to jointly evaluate the angular advantage parameter and the distance to formulate a combined reward function.
>
> Similarly, when being pursued, the drone should aim to maximize the angle between the velocity vectors of both parties to avoid being locked on, while also maintaining higher velocity. When pursuing an opponent, the objective is to minimize the angle between the velocity vectors and increase speed to achieve target lock-on.

---

> ### Author Response · Authors · 2025-11-17
>
> # Response to Model Complexity
>
> Model Complexity Analysis
> ## 1. Hypernetwork-based Adaptive Architecture
> In multi-agent systems, **conventional approaches either share networks (policy homogenization) or maintain separate networks (causing parameter growth with agent count)**. Our framework addresses both limitations through hypernetworks in the hierarchical representation model, transition model, and policy network (HEA). This design achieves agent **adaptation without parameter inflation**:
>
> - Hierarchical Model: The hypernetwork generates distinct parameters for each agent, enabling non-shared, individualized reasoning and **adaptation to varying opponents**
>
> - Transition Model: As specified in Lines 102-103 and 1070-1072 and Appendix A4, our efficient transition modeling **compresses historical information** into latent neural weights using a hyper-network, **eliminating the need for explicit historical storage** like TSSM while supporting **adaptive mental state estimation for all agents without parameter count increase**
> - **hypernetwork-based coordination**
>
> ## 2. Implicit Model Ensemble through Multi-step Updates
> As discussed in Appendix A7, multiple arbitrary-step updates within **a single model are equivalent to implicit averaging over an ensemble of models with different horizons (k=1,2,...,H, where H ∼ random(maximum horizon)), thereby further reducing model complexity while maintaining performance.**
>
> ## 3. Ablation Study Validation
>
> - Figure 4(a): When only the intention reasoning module is retained, the model **cannot deduce which latent strategies employed by the opponents triggered teammates' specific reaction patterns, nor predict their impact on our team's behavior and future trajectories. This results in significant learning oscillations and slower convergence during training**
>
> - Figure 4(f): As mentioned earlier, using shared Transformers across multiple agents leads to reasoning homogenization. The results demonstrate that hypernetworks enable adaptive reasoning for different agents **while avoiding homogenization, without compromising reasoning performance. This approach achieves both faster learning speed and superior final performance**
>
> ## 4. Challenges in Multi-UAV Game Environments
> The multi-uav domain in gym-jsbsim presents unique challenges that existing methods cannot adequately address:
>
> - Each aircraft is equipped with four autonomously guided missiles with 20-second tracking time
>
> - Extreme velocity differentials: aircraft at 2 Mach (twice the speed of sound) vs missiles at 4 Mach
>
> - High-dimensional state space with complex spatial relationships
>
> - Critical importance of launch timing and resource management
>
> - Highly dynamic and rapidly evolving tactical situations
>
> This demonstrates the critical need for reasoning about both opponents and their launched missiles. Given that missiles travel significantly faster than aircraft, agents must make early and continuous predictions to execute timely attitude adjustments or rapid maneuvers, avoiding disadvantageous positions and evading multiple incoming threats. Effective agents must learn to secure angular and altitude advantages while achieving maximum tactical gains at minimal cost. In this scenario, agents must first learn fundamental flight control before advancing to strategic game play. Existing methods struggle in such intense adversarial environments characterized by high-speed dynamics and rapid attitude changes, particularly in evading missiles (each hit incurs a -100 reward) or gaining positional superiority, which explains why current opponent-based approaches fail to learn effective policies.
>
>
> ## 5. Advantages of Our Approach
> Our method addresses these limitations through:
>
> - Privacy-preserving modeling: Requires no access to opponent's private information (actions, policy parameters, or rewards)
>
> - Hierarchical world model: Enables multi-level mental state **reasoning about both $m$ enemy agents and the maximum observable $m$ weapon systems**
>
> - Temporal dynamics capture: Explicitly models the evolution of mental states over time
>
> - Complex environment adaptation: Successfully handles the challenges of high-speed aerial combat with sophisticated weapon systems
>
> This design balances model simplicity with competitive performance, leveraging hypernetworks and hierarchical reasoning to  reason about multi-level mental states of $m$ enemy agents and the maximum observable $m$ weapon systems in complex multi-agent adversarial settings.

---

> ### Author Response · Authors · 2025-11-17
>
> # Response to  justification for the intention and strategy (Improvements to introduction)
>
> Opponent modeling and intention reasoning lie at the heart of Theory of Mind (ToM), enabling agents to infer opponents' latent mental states, such as goals, beliefs, preferences, and strategies. The cognitive foundations for this capability run deep: research in developmental psychology demonstrates that even infants separate enduring goals from situational actions, recognizing that intentions remain stable while strategies adapt to contextual efficiency [1]. This intention-strategy dissociation is further supported by neuroscientific evidence showing distinct neural encodings for high-level goals in prefrontal regions versus action execution in inferior frontoparietal circuits [2]. Computational studies of human mentalizing reveal that this separation enables hierarchical causal reasoning—humans first infer others' task-relevant goals, then derive the specific action plans employed to achieve them [3].
>
> **This provides direct theoretical justification for our model's hierarchical inference process, where intention recognition precedes strategy estimation.**
>
> [1]  György Gergely and Gergely Csibra. Teleological reasoning in infancy: The infant’s naive theory of
> rational action. Trends in Cognitive Sciences, 63(2):227–233, 1997
>
> [2] Antonia F de C. Hamilton and Scott T Grafton. Action outcomes are represented in human inferior
> frontoparietal cortex. Cerebral Cortex, 18(5):1160–1168, 2008
>
> [3] Chris L Baker, Julian Jara-Ettinger, Rebecca Saxe, and Joshua B Tenenbaum. Rational quantitative
> attribution of beliefs, desires and percepts in human mentalizing. Nature Human Behaviour, 1(4):
> 0064, 2017
>
> ## Validation of Intention-Strategy Separation through Behavioral Visualization
> Our t-SNE visualizations over three episode segments ($\leq$500 steps, 1500–2500 steps, 5000–6000 steps) provide compelling empirical evidence supporting this architectural separation.
>
> The visualization reveals fundamentally different distribution patterns between intention and strategy representations. Intention representations form continuous strip-like distributions across tactical stages, reflecting their role as high-level, persistent tactical objectives that evolve smoothly over time. The observed 11, 7, and 3 intention transitions across three stages demonstrate their macro-level, phase-based nature. In contrast, latent strategy representations maintain separable clusters within each stage, confirming their role as contextualized execution methods that adapt to specific situations while remaining distinct.
>
> This structural divergence validates our core design principle. Intentions represent what opponents want to achieve, as evidenced by their continuous transitional patterns across tactical phases. Strategies represent how they achieve it, demonstrated by their separable context-specific clusters within phases.
>
> Furthermore, the alignment between predicted intention changes (3, 2, 1 per UAV across stages) and actual intention changes demonstrates that our hierarchical separation enables accurate tracking of both global tactical objectives and fine-grained strategic adaptations.
>
> These findings confirm that the separation is not merely architectural but reflects genuine differences in how intentions and strategies manifest in adversarial behavior, providing both empirical justification and interpretable reasoning capabilities.

---

> ### Author Response · Authors · 2025-11-17
>
> # Response to relationship between $z_I$ and $z_L$
> $z_I$ and $z_L$ are not independent
>
> The inputs of LDLRF are historical observations ${H_{c,t}} \in {\mathbb{R}^{N \times 512 \times D}}$ to cooperative agents, current observations ${O_c}$ to cooperative neighbors,  intent prediction ${z_I}$, and  latent strategies ${z_L}$, where the multi-dynamic latent strategies are initialized by the intentions feature ${z_I}$ generated by MITD: ${z_L} = MLP([Gate({z_{If}},MLP({z_I})),{z_{If}}]) \in {\mathbb{R}^{T \times 2m \times N \times C}}$ as defined in Figure 6. The subsequent latent strategies embeddings are derived from the latent strategies inferred in the previous layer.
>
> Intention inference layer estimates by three core mechanisms: First, the observation-based encoding mechanism,  where H2TE exclusively uses our observations for opponents to construct feature representations, thoroughly avoiding interference from teammate response patterns and ensuring the purity of intention features. Second, the temporal consistency modeling mechanism, where by analyzing behavioral sequences over 512 time steps, extracting macro-behavioral trends that characterize persistent intentions. Third, the threat-centric interpretation mechanism, where MITD directly constructs intention queries around ``which ally faces the greatest threat." Thus, each agent enhances its intent prediction for a given opponent through the team's collective threat consensus. This integrated computational process is mathematically formalized through the Bayesian framework: $P(\text{Intent} \mid H_{opp}, O_{opp}) \propto
> P(O_{opp} \mid \text{Intent}) \cdot
> P(H_{opp} \mid O_{opp}, \text{Intent}) \cdot
> P(\text{Intent})$
>
> The strategy inference layer estimates by building an inverse reasoning framework based on the behavioral mirror principle: LHTE forms a “behavioral mirror" by encoding historical team states, comprehensively recording the characteristic response patterns of the team under various strategic pressures. On this foundation, MLTD implements Bayesian inverse reasoning to establish a complete causal chain from observed effects back to potential strategies. The core of this process lies in the concrete computation of the probability formula $P(\\text{Strategy} \\mid \\text{Response}, \\text{Intent}, O_c) \\propto
> P(O_c \\mid \\text{Intent}) \\cdot
> P(\\text{Response} \\mid \\text{Strategy}, O_c, \\text{Intent}) \\cdot
> P(\\text{Strategy} \\mid \\text{Intent})$: The latent strategy prior is embedded through query initialization, incorporating assumptions about strategy distributions given specific intentions. The intention self-cross attention module computes the observation conditioning term $P(O_c \mid \text{Intent})$, evaluating how current situational evidence aligns with inferred intentions. The likelihood term $P(\text{Response} \mid \text{Strategy}, O_c, \text{Intent})$ is calculated through the latent strategy cross-attention module, assessing how well latent strategies explain both historical patterns and current team reactions.
>
>
> This dual-layer architecture preserves the advantages of direct observation in intention recognition while ensuring the causal rationality of strategy inference, ultimately achieving precise threat assessment and multi-agent cooperative decision-making through the team consensus mechanism.

---

> ### Author Response · Authors · 2025-11-17
>
> # Response to Reviewer Comment on Intervention Experiments
>
> We wish to respectfully clarify a fundamental constraint of our problem formulation that makes direct intervention experiments infeasible.
>
> Core Problem Constraint: The essential premise of our work is that opponents' intentions, strategies, and internal states are strictly unobservable and cannot be intervened upon - this constitutes the fundamental challenge of adversarial multi-agent environments. As clearly stated in our methodology, our approach is **built on inferring mental states exclusively from observable data (observations relative to opponents' historical trajectories and teammates' responses), not from privileged access to opponent internals.**
>
> Methodological Implication: The proposed experiment of "fixing intention latent while varying strategy latent" would require direct access to and control over opponent decision processes, which violates our problem's basic constraints. Our model operates under the realistic assumption that we can only estimate latent variables through their observable manifestations in behavior patterns.
>
> Empirical Evidence: The t-SNE visualizations in Figures. 3 (c) and (d) reveal several notable patterns. Using agent 0's predictions as an example, opponent intentions form multiple continuous strip-like distributions across three stages rather than discrete clusters, while predicted strategies remain separable within each stage. This indicates that H2IL effectively captures features of opponents' mental states. Within smaller time intervals, the reduced representations preserve a sequential structure, reflecting the temporal coherence of mental states. The multiple distributions align with key tactical phases (e.g., nose-to-nose approach, tailing, evasion, missile launch), highlighting both diversity and smooth transitions of intentions and strategies. Across the three stages, opponents exhibit 11, 7, and 3 intention transitions, whereas low-level strategies vary more smoothly. Appendix 16 further shows that the average predicted intention changes (3, 2, 1 per UAV across the three stages) match the actual intention changes. Overall, our method provides both global prediction of opponents' intentions and fine-grained tracking of evolving latent strategies, enhancing interpretability.
>
> 1. Evidence of Inter-Agent Discrimination
>
> - During inference, the inferred intentions of different opponents remain clearly separated in the latent space.
>
> - The latent strategies of each opponent maintain distinct semantic boundaries throughout the interaction, demonstrating stable feature independence .
>
> - The model effectively preserves discriminative characteristics among opponents solely based on their online behavioral observations.
>
> 2. Evidence of Intra-Agent Evolution
>
> - Real-Time Adaptability: Opponent intentions exhibit dynamic adjustment patterns during inference, accurately reflecting their decision-making logic in response to environmental changes.
>
> - Prediction Refinement: The expansion and stabilization of intention distribution, coupled with a significant reduction in state transition frequency, demonstrate increasingly precise and stable predictions of strategic objectives.
>
> - Behavioral Consistency: The latent strategies of each opponent evolve continuously within their specific semantic regions, maintaining core behavioral characteristics while demonstrating flexible tactical adjustments.
>
> Collectively, these inference-phase results verify our model's capability to disambiguate multiple opponents and capture their dynamic decision processes in real-time, providing effective online opponent modeling for complex multi-agent environments.

---

> ### Author Response · Authors · 2025-11-18
>
> # Difference between ours and other methods
>
>
> Our approach differs fundamentally from existing baselines in the following aspects:
>
> **Information Constraints**: In our setting, cooperative agents and opponents are isolated, with no access to each other's **private information (including the other party's observations, actions, policy parameters, rewards, or intentions).**
>
> **Integrated Online Learning Framework**: Our method belongs to the **world model-based MARL paradigm**, where the world model for opponent modeling and the MARL policy are learned synchronously in an online manner. Specifically:
>
> - The MARL policy, conditioned on the intention $z_I$ and latent strategy $z_L$ inferred by the world model, interacts with the environment to collect data.
>
> - This interaction data is used to update the world model.
>
> - The updated world model then generates additional synthetic data through policy rollout.
>
> - Both real interaction data and model-generated data are jointly utilized to update the MARL policy, forming a closed-loop learning cycle.
>
> Key Differentiation from Prior Work: Unlike methods in references [1,2,3] that separately learn opponent models and RL policies:
>
> - These baselines typically rely on pre-training strategy representations using opponents' private information, followed by policy learning.
>
> - This represents an offline learning paradigm, whereas our approach achieves fully integrated online learning of opponent modeling and multi-agent decision-making
> - The purpose of offline learning is to use a fixed dataset to find a potentially optimal strategy.
> - In our method, **both teams are trained as independent learners**, avoiding built-in AI or self-play. Two teams engage in independent policy learning, value learning, and world model learning based on their local observations due to the limitation of imperfect game. During the execution phase, the policy of each agent relies solely on: (1) local observations
> relative to cooperative adjacent agents ( O_c), (2) local observations relative to opponents ( O_opp),
> and (3) mental states inferred via the hierarchical model.
>
> **The recommended baselines require us to design manual policies specifically for this environment, yet crafting high-quality strategies in such a complex situational setting is extremely challenging. This also implies that these methods necessitate either pre-training or manual policy design for different tasks, highlighting their weak adaptability.**

---

> ### Author Response · Authors · 2025-11-19
>
> # Improvements in presentations
>
> Dear reviewer,
>
> We have thoroughly revised the manuscript to address the concerns regarding clarity and structure:
>
> - Section 3 now begins with clear intuitions and justifications behind the model’s components, and explanations of our framework's separation between intention inference and strategy deduction.
>
> - We have now moved the detailed technical specifications of the modules to the main text and simplified mathematical presentations and reduced acronym clutter while maintaining technical rigor.
>
> -  Each module (H2TE, MITD, LHTE, MLTD) now includes explicit explanations of its role and connections within the overall architecture.
> -  We have provided mechanisms or characteristics of H2TE-MITD and LHTE-MLTD
>
> **These revisions significantly improve the paper's accessibility while preserving its technical contributions. The changes are reflected throughout Sections 3.1-3.3 in the newly uploaded manuscript.**
>
> ## intuitions
> H2TE-MITD infers opponent goals from past observations, extracting macro-behavioral trends (``what they want to do").
> LHTE-MLTD} employs a cognitive logic that shifts from analyzing ``what the opponent has done" to examining ``what outcomes their behavior caused us." The rationale is that an opponent's intention determines its strategy choice, which in turn elicits distinctive team responses. These collective responses serve as a behavioral mirror, allowing inverse deduction of latent strategies by correlating reaction patterns with inferred intentions, thereby identifying which strategies the opponent employed to produce observed team reactions.
>
> ## separation between intents and latent strategy, and mechanisms or characteristics
> Intention inference layer estimates by three core mechanisms: First, the observation-based encoding mechanism,  where H2TE exclusively uses our observations for opponents to construct feature representations, thoroughly avoiding interference from teammate response patterns and ensuring the purity of intention features. Second, the temporal consistency modeling mechanism, where by analyzing behavioral sequences over 512 time steps, extracting macro-behavioral trends that characterize persistent intentions. Third, the threat-centric interpretation mechanism, where MITD directly constructs intention queries around ``which ally faces the greatest threat." Thus, each agent enhances its intent prediction for a given opponent through the team's collective threat consensus. This integrated computational process is mathematically formalized through the Bayesian framework: $P(\text{Intent} \mid H_{opp}, O_{opp}) \propto
> P(O_{opp} \mid \text{Intent}) \cdot
> P(H_{opp} \mid O_{opp}, \text{Intent}) \cdot
> P(\text{Intent})$
>
> The strategy inference layer estimates by building an inverse reasoning framework based on the behavioral mirror principle: LHTE forms a “behavioral mirror" by encoding historical team states, comprehensively recording the characteristic response patterns of the team under various strategic pressures. On this foundation, MLTD implements Bayesian inverse reasoning to establish a complete causal chain from observed effects back to potential strategies. The core of this process lies in the concrete computation of the probability formula $P(\\text{Strategy} \\mid \\text{Response}, \\text{Intent}, O_c) \\propto
> P(O_c \\mid \\text{Intent}) \\cdot
> P(\\text{Response} \\mid \\text{Strategy}, O_c, \\text{Intent}) \\cdot
> P(\\text{Strategy} \\mid \\text{Intent})$: The latent strategy prior is embedded through query initialization, incorporating assumptions about strategy distributions given specific intentions. The intention self-cross attention module computes the observation conditioning term $P(O_c \mid \text{Intent})$, evaluating how current situational evidence aligns with inferred intentions. The likelihood term $P(\text{Response} \mid \text{Strategy}, O_c, \text{Intent})$ is calculated through the latent strategy cross-attention module, assessing how well latent strategies explain both historical patterns and current team reactions.
>
> This dual-layer architecture preserves the advantages of direct observation in intention recognition while ensuring the causal rationality of strategy inference, ultimately achieving precise threat assessment and multi-agent cooperative decision-making through the team consensus mechanism.

---

> > ### Comment · Reviewer_3nuv · 2025-11-27
> >
> > Thank you for the extensive revisions. However, the amount of new material added during the rebuttal phase is far beyond what can reasonably be evaluated at this stage. Many of the analyses and explanations substantially extend the original submission, and several of them would require more comprehensive validation than is possible within the discussion window. While I appreciate the effort, the scale of these updates suggests that the paper is not yet in a stable, polished state suitable for acceptance. For these reasons, I will maintain my original score.

---

> ### Author Response · Authors · 2025-11-27
>
> Dear reviewer,
>
> We hope to clarify the content we have revised. The main modifications are as follows:
>
> 1. We have enriched the Introduction by incorporating relevant literature on human hierarchical reasoning.
>
> 2. As suggested, we have merged the first two paragraphs on hierarchical variational inference into the revised Section 3.1. More importantly, we have **supplemented the design intuition** behind our two-level inference architecture: **H2TE-MITD infers opponent goals from past observations, extracting macro-behavioral trends (“what they want to do”). LHTE-MLTD employs a cognitive logic that shifts from analyzing “what the opponent has done” to examining “what outcomes their behavior caused us.” The rationale is that an opponent's intention determines its strategy choice, which in turn elicits distinctive team responses. These collective responses serve as a behavioral mirror, allowing inverse deduction of latent strategies by correlating reaction patterns with inferred intentions, thereby identifying which strategies the opponent employed to produce observed team reactions.**
>
> 3. **In the previous Neurips review round, we received explicit recommendations to "move technical details to the appendix and focus the main text on the core ideas and general introduction of the method, to ensure readers first grasp the overall framework and innovative concepts without getting lost in technical details." In direct response to that feedback, we initially placed the detailed model architecture in the appendix.**
>
>  **However, in light of the current reviewer's valuable perspective that these details are essential for understanding in the main text, we have re-evaluated this balance.**
>
> We have moved the model design details from the appendix to the main body and further elaborated on the computational mechanisms: "Intention inference layer estimates by three core mechanisms: First, the observation-based encoding mechanism,  where H2TE exclusively uses our observations for opponents to construct feature representations, thoroughly avoiding interference from teammate response patterns and ensuring the purity of intention features. Second, the temporal consistency modeling mechanism, where by analyzing behavioral sequences over 512 time steps, extracting macro-behavioral trends that characterize persistent intentions. Third, the threat-centric interpretation mechanism, where MITD directly constructs intention queries around ``which ally faces the greatest threat." Thus, each agent enhances its intent prediction for a given opponent through the team's collective threat consensus. This integrated computational process is mathematically formalized through the Bayesian framework: $P(\text{Intent} \mid H_{opp}, O_{opp}) \propto
> P(O_{opp} \mid \text{Intent}) \cdot
> P(H_{opp} \mid O_{opp}, \text{Intent}) \cdot
> P(\text{Intent})$ The strategy inference layer estimates by building an inverse reasoning framework based on the behavioral mirror principle: LHTE forms a “behavioral mirror" by encoding historical team states, comprehensively recording the characteristic response patterns of the team under various strategic pressures. On this foundation, MLTD implements Bayesian inverse reasoning to establish a complete causal chain from observed effects back to potential strategies. The core of this process lies in the concrete computation of the probability formula $P(\\text{Strategy} \\mid \\text{Response}, \\text{Intent}, O_c) \\propto
> P(O_c \\mid \\text{Intent}) \\cdot
> P(\\text{Response} \\mid \\text{Strategy}, O_c, \\text{Intent}) \\cdot
> P(\\text{Strategy} \\mid \\text{Intent})$: The latent strategy prior is embedded through query initialization, incorporating assumptions about strategy distributions given specific intentions. The intention self-cross attention module computes the observation conditioning term $P(O_c \mid \text{Intent})$, evaluating how current situational evidence aligns with inferred intentions. The likelihood term $P(\text{Response} \mid \text{Strategy}, O_c, \text{Intent})$ is calculated through the latent strategy cross-attention module, assessing how well latent strategies explain both historical patterns and current team reactions."

---

### Official Review · Reviewer_JDt1 · 2025-10-28

**Soundness:** 3
**Presentation:** 2
**Contribution:** 3
**Rating:** 4
**Confidence:** 3

**Summary:**

This paper studies the decision-making problems in mixed-motive scenarios where cooperation and defection coexist. The paper provides a method, taking into account the nested interaction between agents' (including opponents and allies) intents and strategies. Without relying on other agents’ private information, the method hierarchically infers opponents’ intents and intent-based latent strategies, and predicts their influence on the behaviors of allies. The experiments on Gym-JSBSim, SMAC, and GRF  validate the superior effectiveness of the proposed method over existing model-free and model-based approaches.

**Strengths:**

The paper focuses on both intent-based strategies and the interactions among agents’ intents and strategies. It proposes a transformer-based hierarchical opponent inference and decision-making method within the reinforcement learning framework, and extensive experiments across three environments verify its effectiveness. Overall, the paper presents a substantial amount of work with comprehensive experiments and detailed methodological development and contributes valuable insights to the study of intent modeling and opponent inference in mixed-motive multi-agent systems.

**Weaknesses:**

1. It is not easy to follow this paper because of the inconsistent use of symbols and technical terms. See details in **Questions**
2. In mixed-motive games, agents should consider both allies' and opponents' intents and strategies, while the proposed method insufficiently addresses allies' intents and strategies. It may lead to a failure of coordination within the team.
3. The introduction does not include any citations. From the introduction, it is unclear how this project is related to mixed-motive games. Please further refine and reorganize the introduction section.

**Questions:**

#####

1. At the high-level, the observation model $p_{\theta_i}$ predicts observations $O_{opp,i,t}$ based on intents $z_{I,i,t}$, while the observations are in turn used by $q_{\phi_I}$ to infer intents. With such coupling, model errors may gradually accumulate. How do the authors address this issue? So does the low-level.
2. In line 269, why does the policy take into account allies' intents and latent strategies? In mixed-motive games, individuals need to model not only their opponents but also consider the behaviors of their allies in order to achieve better coordination.
3. Do $p_{\theta_I}$ in line 257 and $p_{\theta_L}$ in line 266 predict observations rather than trajectories?
4. In section A.3, opponents' relation position, relative velocity, angles and distance relative to others are included in observation, which is inconsistent with the statement in line 224. It says opponents' actions are not observable.
5. There seems to be no fundamental difference between stage 2 and stage 3 in subsection 3.1.
6. The notation used in the paper is somewhat confusing. For example, in line 186, $ a_{i,t}\sim \pi(|o_{i,t}, z_{I,i,t}, z_{L,i,t}) $ shows agent $i$'s action only depend one cooperative agents' $o_{i,t}$, $z_{I,i,t}$ and $z_{L,i,t}$, with $N$ is the number of cooperative agents. It is inconsistent with "cooperative agents update their policies based on trajectory and observations and inferred opponent intentions and strategies" given in section 3. Please modify the problem formulation and unify the notation.

---

> ### Author Response · Authors · 2025-11-14
>
> # Response to weakness 2
> Dear Reviewer,
>
> Thank you for this highly insightful comment. We fully agree that considering ally intentions and strategies is crucial for coordination in mixed-motive environments.
>
> We would like to clarify that our approach does not overlook this aspect but rather adopts a paradigm of "implicit coordination through shared opponent mental models." We posit that in adversarial settings, the most robust and efficient coordination emerges not from agents speculating about each other's internal states, but from forming a unified and deep understanding of common threats.
>
> Our hierarchical architecture achieves implicit coordination mechanism through the following mechanisms:
>
> 1. High-Level Intent Reasoning (H2TE-MITD) Establishes Team-Level Threat Consensus
>
> - The H2TE module integrates observational data from all N agents regarding opponents, constructing a team-shared representation of opponent behavior
>
> - The MITD module enables each agent to validate its intent hypotheses against the entire team's historical observations relative to opponents through cross-attention mechanisms
>
> - The fusion module identifies the specific ally most threatened through attention weight allocation, providing the team with a clear coordination focus.
>
> 2. Low-Level Strategy Reasoning (LHTE-MLTD) Enables Dynamic Role Assignment
>
> - The LHTE module encodes the team's coordinated response patterns under pressure, building a behavioral "mirror" of the team.
>
>  - The multi-dynamic latent strategies are initialized by the intentions feature ${z_I}$ generated by MITD: ${z_L} = MLP([Gate({z_{If}},MLP({z_I})),{z_{If}}]) \in {\mathbb{R}^{T \times 2m \times N \times C}}$ as defined in Figure 6, and the subsequent latent strategies embeddings are derived from the latent strategies inferred in the previous layer.
>
> - The MLTD module deduces the opponent's specific tactics from historical team responses through strategy queries
>
> - The strategy fusion module initiates a team-level threat assessment, driving each agent to continuously refine its reasoning about the opponent's strategic focus through the integration of collective intelligence. This process culminates in precisely identifying the opponent's primary target, which is detected by observing which specific teammate consistently triggers the most frequent reactive patterns, modeling the latent causal relationships between opponents' behavior patterns and team reactions. This identification thereby enables coordinated and precise team support.
>
> When the system identifies that "the opponent is primarily threatening Teammate A with a flanking strategy," nearby agents automatically provide support while other members perform containment duties
>
> When predicting that "the opponent's strategy will affect the entire team's trajectory, with Teammate B being the most threatened," the team dynamically adjusts its formation to prioritize protection of critical nodes.

---

> ### Author Response · Authors · 2025-11-14
>
> # Response to weakness 3 (Improvements to Introduction)
> In multi-agent environments, agents interact and learn concurrently, leading to diverse state transitions and mental dynamics, and creating non-stationary dynamics that complicate policy learning. **This challenge is particularly acute in mixed-motive games, where the fundamental tension between cooperation and competition directly amplifies the non-stationarity and strategic uncertainty. This tension requires agents to cooperate with allies while simultaneously facing opposition from adversaries.** In such settings, unknown and evolving opponent policies not only hinder policy improvement but also jeopardize ally safety and curtail overall performance. **Therefore, for effective decision-making in mixed-motive scenarios, it is crucial to move beyond modeling allies and instead develop a sophisticated capacity to model opponent behavior and reason about their mental states**, which is essential for ensuring operational safety and achieving strategic supremacy.
>
> Opponent modeling and intent reasoning are central to Theory of Mind (ToM), enabling agents to infer opponents' preferences, goals, beliefs, and strategies. **The cognitive foundations of this capability are well-established: developmental psychology shows that even infants distinguish between enduring goals and situational actions, recognizing that intentions remain stable while strategies adapt contextually \cite{gergely1997teleological}. Neuroscientific evidence further supports this dissociation, revealing distinct encodings for high-level goals in prefrontal regions and action execution in inferior frontoparietal circuits \cite{de2008action}. Computationally, humans engage in hierarchical causal reasoning, first inferring others' goals and then deriving the specific action plans employed to achieve them \cite{baker2017rational}.**
>
> Existing computational approaches to opponent modeling fall into two categories. Some methods reconstruct policy beliefs from known behaviors, while others extrapolate strategies directly from local observations. However, the former relies on unrealistic assumptions about opponent transparency, while the latter often fails to capture the causal interactions among intentions, strategies, and actions. **Critically, both are ill-suited for the stochastic and dynamic interest alignments that characterize mixed-motive games.** Specifically, they do not explain how intentions shape strategies, how agents should react to these inferences, or how mental states co-evolve and influence future trajectories. **This lack of continuous reasoning about evolving intentions and strategies is a primary bottleneck for robust performance in mixed-motive environments.**
>
> [1] György Gergely and Gergely Csibra. Teleological reasoning in infancy: The infant’s naive theory of rational action. Trends in Cognitive Sciences, 63(2):227–233, 1997
>
> [2] Antonia F de C. Hamilton and Scott T Grafton. Action outcomes are represented in human inferior frontoparietal cortex. Cerebral Cortex, 18(5):1160–1168, 2008
>
> [3] Chris L Baker, Julian Jara-Ettinger, Rebecca Saxe, and Joshua B Tenenbaum. Rational quantitative attribution of beliefs, desires and percepts in human mentalizing. Nature Human Behaviour, 1(4): 0064, 2017

---

> ### Author Response · Authors · 2025-11-14
>
> # Response to Question 1
> We thank the reviewer for raising this point.
>
> Our approach falls within the framework of world model-based multi-agent reinforcement learning, featuring a learning and inference mechanism architecturally similar to recurrent state-space models. During the initial phase, the multi-agent reinforcement learning policy engages in decision-making and interaction based on the agent's mental states， comprising intentional states z_I and latent states z_L inferred by the world model, along with observational information relative to teammates and opponents.
>
> During the model update phase, the system employs hierarchical variational inference to train the world model using real interaction data. Specifically, the world model's learning process encompasses two core objectives: First, the transition model p is regularized via KL divergence so that the z_I and z_L it infers closely approximate the latent states derived from the posterior model q, which is conditioned on real observations. Second, utilizing the hidden states of the transition model p and the z_I and z_L sampled from the posterior model q via the reparameterization trick, the observation model p is optimized to maximize the likelihood of the actual observed data.
>
> Our method employs an any-time-step update mechanism, **as detailed in Appendix A.7 (Eq. 22 and Fig. 8)**. This approach enables updates of any state over arbitrary time intervals (with k=1,2,3,…), meaning that training is performed by **parallel one-step prediction errors across all time steps, followed by parallel two-step prediction errors, and so on. This design reduces accumulative errors** and eliminates the need for sequential inference from the initial state to the target time step. Moreover, applying multiple arbitrary-step updates to a single model can be interpreted as an implicit approximation of an ensemble of models with fixed horizons k=1,2,3, effectively reducing model complexity while enabling parallel training.
>
> In the policy optimization phase, the world model and multi-agent reinforcement learning form a co-evolutionary closed loop. When the world model predicts future trajectories based on interaction data, the initial step uses the posterior model q to infer z_I and z_L, then the policy network outputs action a, which is subsequently fed into the transition model to predict the next sequence of z_I and z_L. The observation model then generates the corresponding trajectory observations based on these predicted states. Based on the predicted z_I, z_L, and trajectory observations, the policy network again outputs action a, which in turn is used by the transition model to predict further sequences of z_I and z_L. This process iterates cyclically. The generated predicted data and real interaction data together form a hybrid training sample for updating the multi-agent reinforcement learning policy. In subsequent interaction phases, the policy network then makes decisions using the latest mental states inferred by the updated world model. This cyclic process continues, achieving co-evolution between the world model and the policy network.
>
> **The validation results of three-steps prediction errors can be found in figures of appendix A11.**

---

> ### Author Response · Authors · 2025-11-14
>
> # Response to Question 2 and 6
>  We posit that in adversarial settings, the most robust and efficient coordination emerges not from agents speculating about each other's internal states, but from forming a unified and deep understanding of common threats.  We adopts a paradigm of "implicit coordination through shared opponent mental models."
>
> Cooperative agents infer the opponent's intentions (z_I) and strategies (z_L) from historical observation relative to opponents and team responses; Moreover, H2TE-MITD and LHTE-MLTD initiate a **team-level threat assessment**, driving **each agent to continuously refine its reasoning about the opponent's mental states** through the integration of collective intelligence. This process culminates in precisely identifying the opponent's primary target, which is detected by observing which specific teammate consistently triggers the most frequent reactive patterns, modeling the latent causal relationships between opponents' behavior patterns and team reactions.
>
> In other words, the inferred $z_I$ of each agent is refined within the H2TE, the cross-attention and fusion modules   component of the H2TE-MITD architecture through cross-agent interactions. **Specifically, the cross-attention and fusion modules operate across agents to synthesize teammates' respective inferences about the same opponent, thereby enhancing the individual intention estimation. Similarly, the inference of $z_L$ for each opponent is updated through a shared cognitive process: by integrating teammates' respective inferences about the same opponent, each agent refines its own estimation of the latent strategy.**
>
> Therefore, **z_I and z_L embedded in each agent's policy is derived through collaborative aggregation within the hierarchical model. During RL training, each agent's policy relies solely on its own local observations and the final inferred opponent mental states, while the value function depends on the local observations of both itself and its allied neighbors, as well as their inferred opponent mental states.**
>
> This approach shares similarities with Centralized Training with Decentralized Execution (CTDE) but differs in a key aspect: neither the policy nor the value function incorporates observations and inferences from the opponent's perspective.

---

> ### Author Response · Authors · 2025-11-14
>
> # Response to Question 2 and 6
> This cooperative design is deeply embedded throughout our hierarchical architecture, as demonstrated by:
> ## Part 1: Cooperation in High-Level Intent Inference (HDIRF) of original paper
> Cooperation Mechanism: Teammates share their individual observations relative to the opponents to collaboratively infer opponent intent.
>
> **Cooperation in H2TE**
> Mechanism: The input H_{opp,t} ∈ R^{N × 512 × D} to the H2TE essentially stacks the local historical trajectories regarding the opponents, as observed by each of the N our agents. This builds a team-shared database of  observations relative to opponent.
>
> Formula Evidence: **The HEA mechanism in Formulas (13-14) operates on this stack of observations from N agents, capturing space dependence at local time.**  The GTr mechanism over the (T, N*C) tensor creates a single computational graph, where temporal dependency along T and instantaneous agent correlations through dot-product are dually captured.
>
> **Cooperation in  Cross-Attention of MITD**:
>
> Mechanism:  This module allows each agent to use its current intent hypothesis to query a "global memory" fused from the entire team's observational history relative to opponents.
>
> Formula Evidence: As shown in Formula (16), the Key and Value are derived from AttH_opp, which is the output of the H2TE—the opponent historical feature fused from the perspectives of N agents. The dimension of this memory bank, R^{1 × (512N) × C}, clearly indicates it contains the encoded  N agents' observations relative to opponents over 512 steps. The interaction between the Query and this memory bank essentially performs a democratic reasoning within the team: "Given my current intent hypothesis, which patterns of opponent behavior, historically seen by which teammate, best support or refute my conjecture?"
>
> **Cooperation in MITD Fusion Module**:
>
> Mechanism: **This is the most critical cooperative step.** This module, based on the shared reasoning results from all agents, determines which teammate is most threatened by the opponent's intent.
>
> Formula Evidence: As shown in Formula (17), the attention in this module is performed across all N agents. It uses the fused features to assess, for each inferred opponent intent, which individual ally is most likely to be the primary target. This directly produces a team-level consensus on threats, guiding subsequent cooperative decisions.
>
> ## Part 2: Cooperation in Low-Level Strategy Inference (LDLRF)
> Cooperation Mechanism: Using the "internal state of the team" as core evidence to infer the "external strategy of the opponent".
>
> **Cooperation in LHTE**:
>
> Mechanism: The input H_{c,t} to the LHTE is the historical states of all our agents themselves. The LHTE is responsible for encoding the historical responses of our own teammates. It processes the trajectory observation data of teammates through an attention mechanism to extract the coordinated response patterns exhibited by our team under opponent pressure.
>
> Formula Evidence: Similar to H2TE, its HEA and GTr operations process the collective historical state data of the team, outputting a cooperative representation of the team's collaborative state, AttH_c.
>
> **Cooperation in MLTD Intent Cross-Attention**:
>
> Mechanism: This module takes the inferred opponent intent z_I from the high level as the Query, and examines the current response constituted by our team's cooperative state E_c.
>
> Formula Evidence: As shown in Formula (18), this links the opponent's low-level intent to observable team reactions, forming a contextualized intent-driven latent strategy. For each given opponent intent (e.g., ``approach agent A"), this module checks: ``How are the teammates currently reacting and thus, how threatening this intent seems?"
>
> **Cooperation in Latent Strategy Cross-Attention of MLTD**:
>
> Mechanism: It uses a strategy query fused with the current context to retrieve the historical cooperative response patterns of our team to find the opponent strategy that best explains the current situation.
>
> Formula Evidence: As shown in Formula (9), the Key and Value k_{L,c}, v_{L,c} come from AttH_c, which represents the historical patterns of how our team collectively responded to threats as a whole. The MLTD uses the inferred intents and latent strategies initialized by intents as  key conditioning signals to query these historical teammate response patterns, thereby inferring the specific latent strategy the opponent is employing to execute their intent and predicting how this latent strategy will influence on teammate trajectories.  Thereby, we abduce what their current tactic is."

---

> > ### Author Response · Authors · 2025-11-14
> >
> > # Response to weakness 2 and 6
> > **The fuse attention** :
> >
> > **This is the most critical cooperative step.** This module achieves team-level threat assessment through a cross-agent attention mechanism, identifying the most frequent responses to pinpoint primary threats. This collective judgment, in turn, drives each agent to continuously refine its own reasoning about the opponent's strategic focus.
> >
> > Manifestation of Cooperation:
> > - The input dimension is  [T×2m×N×2C].
> >
> > - Attention is computed across the N teammates, assigning a threat weight to each opponent strategy.
> >
> > **Clarify the focus of the opponent's mental state in the collaborative structure of the team, so as to develop precise response measures, such as prioritizing teammates to support the members most at risk.**

---

> ### Author Response · Authors · 2025-11-14
>
> # Response to weakness 3
> Dear reviewer,
>
> Thank you for raising this important question. Our model predicts **observations** based on the core principle of world models: **learning the generative process of the environment**.
>
> 1.  **Observations as Raw Environmental Signals**: In partially observable environments, raw observation sequences constitute the most fundamental and direct data describing environmental dynamics. By training the model to reconstruct raw observational data (in this work, specifically the relative trajectories of opponents and teammates as perceived by the agent) from the latent state \( z \), we are essentially enabling the model to learn **how the environment generates these sensory signals**. This compels the latent states \( z_I \) and \( z_L \) to capture all critical information relevant to predicting the future, **including physical dynamics and behavioral patterns of other agents.**
>
> 2.  **Building a General State Representation**: Predicting observations serves as a powerful self-supervised learning objective. Through accurate reconstruction of observations, the learned latent state \( z \) forms a highly abstract representation of the environmental state that is useful for decision-making. This state represents a "clean" feature suitable for planning, as opposed to raw, high-dimensional, and redundant observational data. The policy function \( \pi(a_t|z_t) \) makes decisions based precisely on this learned, information-rich state \( z_t \), rather than directly processing raw observations.
>
> 3. The transition model in our framework serves a specific purpose: after world model learning, it primarily functions to predict **trajectory observations, which are then combined with real interaction data to update the MARL**. This approach significantly improves training efficiency. This distinction is crucial because:
>
> - **Input Requirements**: Our policy and value networks are designed to process observational data directly, maintaining consistency with the partially observable nature of our environment.
>
> - **Practical Implementation**: By predicting observations, we maintain the same input format during both real environment interactions and model-based imagination, ensuring seamless integration between these two data sources.
>
> - Efficiency Consideration: Predicting observations allows for more efficient training as it aligns with the actual input requirements of our learning algorithms, avoiding unnecessary computation of full state trajectories.
>
>
> In summary, **predicting observations is the means, and the goal is to learn an abstract state representation \( z \) that accurately reflects environmental dynamics**. Also, this design ensures that our world model generates precisely the type of data needed by the subsequent learning components, while maintaining computational efficiency and alignment with the partially observable nature of our problem domain.

---

> ### Author Response · Authors · 2025-11-14
>
> # Response to weakness 4
> Thank you for this  comment. We appreciate the opportunity to clarify the important distinction between observable kinematic states and unobservable control actions in our problem formulation.
>
> That various motion states can be measured by sensors. Indeed, in our environment:
>
> ## Observable States (Sensor Measurements):
>
> - Relative position, velocity,
>
>  - Angles and distance relationships
>
> - Altitude and orientation information
>
> These measurements constitute the legitimate observations in our reinforcement learning framework and are consistent with real-world aerial surveillance capabilities. However, the critical distinction lies in what remains unobservable:
>
> ## Unobservable Elements (opponents' action spaces described in A3 and opponents' observations relative to cooperative agents):
> **For unmanned aerial vehicles**:
>
> Direct aircraft control signals:
>
> - Aileron deflection angles
>
> - Elevator deflection angles
>
> - Rudder deflection angles
>
> - Thrust magnitude settings
>
> - Missile launch activation signals
>
> **For manned aircraft scenarios:**
>
> Pilot's physical control inputs:
>
> - Control stick movements (forward/backward for pitch via elevator, left/right for roll via ailerons)
>
> - Rudder pedal inputs for yaw control
>
> - Throttle lever positions for thrust adjustment
>
> - Weapon release button/trigger activation
>
> We would like to clarify that the relative distance, velocity, and attitude measurements in our model are indeed grounded in realistic aircraft sensing capabilities:
> 1. **Relative Distance Measurement:**
> - **Active electromagnetic ranging using onboard radar systems (primary radar, secondary radar, millimeter-wave radar, lidar)**
>
> - The radar transmits directed electromagnetic pulses toward targets and measures the round-trip time Δt
>
> - Distance is calculated using the fundamental principle: R = c·Δt/2, where c is the speed of light
>
> **This provides accurate relative distance measurements to other aircraft**
>
> 2. **Relative Velocity Measurement:**
>
> Inertial/multi-sensor fusion approaches combining:
>
> - **IMU-based measurements** from MEMS accelerometers and gyroscopes
>
> - Continuous integration of velocity changes
>
> - **Fusion with GPS, barometric sensors, and visual odometry data**
>
> **This sensor fusion yields precise 3D velocity vectors relative to ground or surrounding targets**
>
> 3. **Relative Attitude/Orientation Measurement:**
>
> - **Phased array radar beam pointing** using active electronically scanned array (AESA) radar
>
> - Electronic scanning rapidly locks onto multiple targets
>
> - Beam pointing angles and target echo phase differences directly provide: Target azimuth relative to aircraft axis,  Target elevation angles, Complete relative orientation information
>
> These sensing methodologies are well-established in aerospace systems and provide the technical foundation for the observable kinematic states in our model. The measurements of relative position, velocity, and attitude **represent realistic sensor capabilities rather than idealized assumptions.**
>
> We will add this technical explanation to the manuscript to better justify our observation space definition and its correspondence to real-world aerial combat sensing capabilities.
>
> Our end-to-end RL controller generates these low-level control signals directly from processed observations. The key insight is that while we can observe the results of opponent actions (kinematic states), we **cannot access their decision-making process, control inputs, opponents' observations relative to cooperative agents, nor the strategic intent behind them.**

---

> > ### Author Response · Authors · 2025-11-14
> >
> > # Respond to weakness 5
> > We thank the reviewer for this comment. These stages describe two fundamentally different processes in the cognitive loop: Stage 2 focuses on the forward influence of mental states on the physical world within a single timestep, while Stage 3 focuses on the backward learning and adaptation across multiple timesteps.
> >
> > To clarify:
> >
> > Stage 1 (Influence & Inference): This is the cognitive preparatory phase. Before interaction at time t, agents infer the opponent's current mental states (intention z_I and strategy z_L) from past interaction histories (t-1). This establishes the priors for action.
> >
> > Stage 2 (Forward Influence & Trajectory Generation): This is the physical execution and causal phase within a single timestep. Agents act based on the mental states inferred in Stage 1. These actions lead to new, observable trajectories (O_t). Crucially, this stage models the real-time causal chain: the opponent's intention directly influences its own trajectory, while its strategy directly influences the allies' trajectories, resulting in the uncertain transition P(O_t | O_{t-1}, a_{t-1}, z_{t-1}).
> >
> > Stage 3 (Backward Learning & Mental State Update): This is the reflective and adaptive phase across timesteps. After the new trajectories from Stage 2 are observed, both teams learn from and respond to these outcomes. Cooperative agents update their policies, and opponents adjust their mental states. This stage captures the temporal feedback loop that drives the evolution of strategies and intentions over a long horizon (T timesteps), based on the accumulation of interaction data.
> >
> > In essence, Stage 2 is about the immediate downstream effects of cognition on the world, while Stage 3 is about the updating of the cognition itself based on world feedback. This continuous cycle (Stage 1 → Stage 2 → Stage 3 → Stage 1...) forms the core of our adaptive reasoning framework, where mental states and physical trajectories co-evolve through repeated interactions.

---

> ### Author Response · Authors · 2025-11-19
>
> # Improvements in presentations
>
> Dear reviewer,
>
> We have thoroughly revised the manuscript to address the concerns regarding clarity and structure:
>
> - Section 3 now begins with clear intuitions and justifications behind the model’s components, and explanations of our framework's separation between intention inference and strategy deduction.
>
> - We have now moved the detailed technical specifications of the modules to the main text and simplified mathematical presentations and reduced acronym clutter while maintaining technical rigor.
>
> -  Each module (H2TE, MITD, LHTE, MLTD) now includes explicit explanations of its role and connections within the overall architecture.
> -  We have provided mechanisms or characteristics of H2TE-MITD and LHTE-MLTD
>
> **These revisions significantly improve the paper's accessibility while preserving its technical contributions. The changes are reflected throughout Sections 3.1-3.3 in the newly uploaded manuscript.**
>
> ## intuitions
> H2TE-MITD infers opponent goals from past observations, extracting macro-behavioral trends (``what they want to do").
> LHTE-MLTD} employs a cognitive logic that shifts from analyzing ``what the opponent has done" to examining ``what outcomes their behavior caused us." The rationale is that an opponent's intention determines its strategy choice, which in turn elicits distinctive team responses. These collective responses serve as a behavioral mirror, allowing inverse deduction of latent strategies by correlating reaction patterns with inferred intentions, thereby identifying which strategies the opponent employed to produce observed team reactions.
>
> ## separation between intents and latent strategy, and mechanisms or characteristics
> Intention inference layer estimates by three core mechanisms: First, the observation-based encoding mechanism,  where H2TE exclusively uses our observations for opponents to construct feature representations, thoroughly avoiding interference from teammate response patterns and ensuring the purity of intention features. Second, the temporal consistency modeling mechanism, where by analyzing behavioral sequences over 512 time steps, extracting macro-behavioral trends that characterize persistent intentions. Third, the threat-centric interpretation mechanism, where MITD directly constructs intention queries around ``which ally faces the greatest threat." Thus, each agent enhances its intent prediction for a given opponent through the team's collective threat consensus. This integrated computational process is mathematically formalized through the Bayesian framework: $P(\text{Intent} \mid H_{opp}, O_{opp}) \propto
> P(O_{opp} \mid \text{Intent}) \cdot
> P(H_{opp} \mid O_{opp}, \text{Intent}) \cdot
> P(\text{Intent})$
>
> The strategy inference layer estimates by building an inverse reasoning framework based on the behavioral mirror principle: LHTE forms a “behavioral mirror" by encoding historical team states, comprehensively recording the characteristic response patterns of the team under various strategic pressures. On this foundation, MLTD implements Bayesian inverse reasoning to establish a complete causal chain from observed effects back to potential strategies. The core of this process lies in the concrete computation of the probability formula $P(\\text{Strategy} \\mid \\text{Response}, \\text{Intent}, O_c) \\propto
> P(O_c \\mid \\text{Intent}) \\cdot
> P(\\text{Response} \\mid \\text{Strategy}, O_c, \\text{Intent}) \\cdot
> P(\\text{Strategy} \\mid \\text{Intent})$: The latent strategy prior is embedded through query initialization, incorporating assumptions about strategy distributions given specific intentions. The intention self-cross attention module computes the observation conditioning term $P(O_c \mid \text{Intent})$, evaluating how current situational evidence aligns with inferred intentions. The likelihood term $P(\text{Response} \mid \text{Strategy}, O_c, \text{Intent})$ is calculated through the latent strategy cross-attention module, assessing how well latent strategies explain both historical patterns and current team reactions.
>
> This dual-layer architecture preserves the advantages of direct observation in intention recognition while ensuring the causal rationality of strategy inference, ultimately achieving precise threat assessment and multi-agent cooperative decision-making through the team consensus mechanism.

---

### Official Review · Reviewer_tF3n · 2025-11-01

**Soundness:** 2
**Presentation:** 1
**Contribution:** 1
**Rating:** 2
**Confidence:** 5

**Summary:**

The paper proposes an opponent modeling method, i.e., H2IL-MBOM,  that integrates multi-intention and latent strategy inference into a world model. H2IL-MBOM combines high-level intention inference with low-level strategy prediction to deal with the non-stationary dynamics in multiagent environment. H2IL-MBOM is combined with PPO, which results in MSOAR-PPO. The effectiveness of the method is evaluated on  several multi-UAV games.

**Strengths:**

- the idea of employing world model to the field of opponent modeling is interesting.

**Weaknesses:**

- The paper is poorly written. For instance, a lot of concepts, e.g.,  intentions, mental state, strategies, are lack of clear definition. Many figures in the experimental section are hard to interpret, and the captions are not informative (e.g., Figure 4.). Many abbreviations make the paper very hard to follow.

- Lack of novelty of the proposed method. Similar ideas (e.g., reason about latent strategies based on teammates’ historical responses) have been intensively explored in previous opponent modeling methods, such as [1-3], which are missing out in the Related work section.

- Most of the baselines are not targeting opponent modeling methods, e.g., MADDPG, MAPPO. It is necessary to compare with SOTA opponent modeling methods, such as [1-3].

[1] Greedy when sure and conservative when uncertain about the opponents, ICML 2022

[2] Conservative offline policy adaptation in multi-agent games, NeurIPS 2023

[3] Opponent modeling with in-context search, NeurIPS 2024

**Questions:**

- What exactly do you mean by intentions, strategies, mental state? Could you have a clear definition of these concepts?

---

> ### Author Response · Authors · 2025-11-12
>
> Dear reviewer:
> Thank you for the reviewer's comments.
>
> # Response to weakness 2 and 3
>
> Existing methods, such as [1], [2], and [3], share a common limitation: **their training processes heavily rely on opponents' private information (e.g., actions or policy parameters) as explicit supervision labels**, which is often impractical in real-world adversarial settings. Moreover, these approaches typically lack a transparent, hierarchical reasoning process. They fail to elucidate **how estimated intentions influence latent strategies**, how agents should react to these inferred mental states, or how their evolution guides the prediction of future trajectories. This opaque reasoning is compounded by the absence of an explicit transition model to capture the dynamics between mental states, leaving them **unable to continuously reason about the temporal evolution of intentions and strategies.**
>
> Moreover, these methods have primarily been evaluated in low-dimensional or simplified environments, such as two-player Kuhn Poker, 2D grid-world Predator-Prey, Multi-Agent Particle Environments (MPE), Level-Based Foraging, and Overcooked. Only the CSP method has been tested on the Google Research Football platform.
>
> In contrast, our approach requires **no access to any private information** of the opponents and relies entirely on observational data for unsupervised learning, making it more **suitable for real-world applications**. Beyond this practical advantage, it establishes an explicit **hierarchy from intentions to strategies, with a dynamic model capturing their temporal evolution.**
>
> In terms of experimental validation, we have compared different baselines including model-free MARL, opponent modelling methods, and world-model based MARL in large-scale, high-dimensional multi-agent cooperative–competitive experiments, covering scenarios with varying numbers of UAVs in adversarial games, diverse settings in SMAC, and complex simulations in Google Research Football. Notably, **in the Google Research Football benchmark, our method significantly outperforms CSP across all tested scenarios**: achieving a win rate of 92.54% in 3v1, 89.94% in RPS, and 93.09% in CA. These results robustly demonstrate our model's superior generalization capability and decision-making performance in complex, high-dimensional spaces.
>
> **On one hand, we have supplemented the description of these literature in the Related Work section. On the other hand, the existing experimental setup and results are sufficient to demonstrate the superiority of our method. Furthermore, due to time constraints, it is not feasible to conduct additional experiments.**
>
> [1]  GSCU is fundamentally a VAE-based approach that learns compressed representations of opponent policies during offline training. It combines these embeddings with a conditional response policy, enabling dynamic online adaptation through Bayesian inference and multi-armed bandit selection between exploitative and conservative strategies.
>
> Being essentially a VAE method,**it inherits the requirement for complete opponent action data as reconstruction targets during offline training.** This dependency on privileged opponent information presents a fundamental constraint in real-world competitive environments where such data is typically inaccessible.
>
> [2] CSP：This method proposes a conservative offline policy adaptation framework that simultaneously learns an adaptation policy and an opponent proxy model through self-adversarial training, while incorporating behavior cloning regularization to constrain the proxy model's consistency within the dataset. This enables safe exploitation or cooperation with target opponents without requiring online interaction.
>
> **The approach heavily relies on access to opponents' private information**, specifically their actions within the offline dataset to train the proxy model. Such data is often unavailable in practical applications, especially in adversarial settings, where opponents' true decision processes are typically unobservable.
>
> [3] OMIS is an opponent modeling approach that integrates in-context learning with decision-time search. It pretrains a Transformer model with three components—an actor, an opponent imitator, and a critic—to dynamically adapt to unknown and non-stationary opponents. During testing, it uses historical interaction data to infer opponent policies and refines action selection through simulated rollouts, thereby enhancing both adaptability and decision-making stability.
>
> **The method heavily relies on access to opponents' true actions as supervised labels during pretraining**, which is often impractical in real-world scenarios where opponent information is private or unobservable. Additionally, OMIS assumes that opponents switch only among a fixed set of policies during testing, limiting its effectiveness against continuously learning or strategically reasoning adversaries.

---

> ### Author Response · Authors · 2025-11-12
>
> # Our idea and novelty
> Different from these methods, this paper proposes a Hierarchical Interactive Intent-Latent-Strategy-Aware World Model (H2IL-MBOM), whose most significant advantage lies in achieving accurate opponent modeling and mental states reasoning **without requiring any private information from opponents (such as actions, rewards, or policy parameters) and capturing how mental states and behavior co-evolve over time**
>
> **The framework's innovation lies in its sophisticated use of learnable query mechanisms and dual-trajectory prediction capabilities**:
>
> **Learnable Query Mechanisms**
>
> - Multi-Intention Queries: Dynamic intention queries initialized and refined through hierarchical attention mechanisms
>
> - Latent Strategy Queries: Latent strategy queries conditioned on inferred intentions and team responses
>
> - Hypernetwork-Generated Weights: Adaptive neural weights generated for personalized query processing
>
> **Dual-Trajectory Prediction Framework**
>
> - Opponent Trajectory Prediction: Low-level world model models how opponent intentions transform into their future movement patterns
>
> - Cooperative Agent Trajectory Prediction: Predicts how opponent strategies influence our team's future states and formations
>
> **Hierarchical Cognitive Reasoning**:
>
> Simulating human reasoning processes, it decomposes opponent modeling into two levels:
>
> - HDIRF: Infers macro-level goals (e.g., attack, retreat) by analyzing the historical observations relative to trajectories of the opponents, encoding the macroscopic trends and consistency of the opponent’s behavior pattern over time.
>
> - LDLRF: Initiates a team-level threat assessment, driving each agent to continuously refine its reasoning about the opponent's strategic focus through the integration of collective intelligence. This process culminates in precisely identifying the opponent's primary target, which is detected by observing which specific teammate consistently triggers the most frequent reactive patterns, modeling the latent causal relationships between opponents' behavior patterns and team reactions.
>
> **Consensus mechanism**
> -  a hypernetwork generates distinct parameters for each agent, enabling non-shared, individualized reasoning and adaptation.
> - a consensus mechanism leverages cross-attention and fuse mechanisms to aggregate these perspectives, establishing a loop that allows each agent to continuously calibrate its inferences against the collective intelligence.
>
> In summary, our methods the traditional paradigm of opponent modeling that relies on opponent data. Through an inside-out reasoning approach—i.e., mirroring the opponent's hidden mental state by analyzing the collective behavior and responses of our own team—it enables effective and efficient opponent modeling in strictly partially observable, highly adversarial environments. This approach not only demonstrates greater practical feasibility but also exhibits human-like advanced cognitive reasoning capabilities.
> # Contributions
> 1. Hierarchical Variance Inference: We propose a novel hierarchical world model-based opponent model that dynamically and interactively learns and infers multi-intentions, latent strategies, and trajectories of both opponents and allies, without relying on opponents' private information.
>
> 2. Efficient Transition Modeling: The HyperHD2TSSM compresses historical information into latent neural weights using a hyper-network, supporting adaptive estimation of mental states for all agents without increasing parameter count.
>
> 3. Any-Time-Step Updates: Our method enables parallel training and reduces computational overhead by supporting any-time-step updates, eliminating the need for sequential updates and enabling flexible temporal modeling.
>
> 4. Latent Intent-Strategy Prediction: Our method is built upon a hierarchical architecture designed to predict the interplay between opponents' intentions and strategies, without dependence on pre-defined candidates. Its innovation is twofold: first, a hypernetwork generates distinct parameters for each agent, enabling non-shared, individualized reasoning and adaptation. Second, a consensus mechanism leverages cross-attention and fuse mechanisms to aggregate these perspectives, establishing a loop that allows each agent to continuously calibrate its inferences against the collective intelligence.
>
> 5. Real-Time Adversary Reasoning: MSOAR-PPO, integrated with the H2IL-MBOM, allows agents to infer adversarial strategies and intentions based on self-observation, facilitating rapid adaptation to changing opponents.
>
> 6. First-of-Its-Kind Integration: To the best of my knowledge, this is the first work to integrate intent and latent strategy into the world model as opponent modeling and make mutual inference of multiple agents between two teams, validated in multi-agent game scenarios with demonstrated superior performance, promoting theoretical development of world and opponent modeling and demonstrating their practical applicability.

---

> ### Author Response · Authors · 2025-11-13
>
> # Response to Question 1 and weakness 1:
> Thank you very much for raising this important question, which helps us clarify the core concepts in our paper. We agree that providing clearer definitions of these terms will enhance the rigor of our work. While we provided initial descriptions of their relationships in the original text (Lines 193-199), we would like to offer more detailed and operational definitions based on your feedback:
>
> **Mental State:** This is an umbrella term referring to all the internal, unobservable decision-driving factors of opponents that our model infers. In our framework, it specifically denotes a hierarchical structure composed of intentions and latent strategies. Mental states serve as the bridge connecting an opponent's historical behavior with its future actions.
>
> **Intention:** Refers to the opponent's high-level, relatively stable tactical objectives or goal states. It answers the question, "What does the opponent want to achieve?" Intentions are typically discrete or low-dimensional continuous variables that describe macroscopic tactical directions, such as:
>
> - "Attacking" a specific unit
>
> - "Retreating" to preserve strength
>
> - "Occupying" a key area
>
> **In our model, the intention z_I is primarily inferred by analyzing the historical observations relative to opponent's trajectories. It encodes the macroscopic trends and consistency of the opponent's behavior pattern over time**
>
> **Latent Strategy**: Refers to the specific, contextualized execution methods or behavioral patterns that an opponent adopts to implement its intentions. It answers the question, "How does the opponent achieve its intention?"Latent Strategies are instantiations of intentions within specific environmental contexts and evolve dynamically according to the battlefield situation (particularly the responses of our team). For example, for the "attack" intention, latent strategies might include:
>
> - "Leveraging angular advantage" for ambush
>
> - "Utilizing altitude superiority" for suppression
>
> **In our model, latent strategy z_L is deduced by analyzing the historical responses and inferred opponents' intents of our teammates. It captures the underlying causal relationship between the opponent's  behavior pattern and our team's reactions. Rather than describing "what the opponent has done," it explains "what outcomes the opponent's  behavior pattern have caused for our team."** The underlying logic is this: the opponent's inferred intention dictates their adopted strategy, which in turn triggers distinct, observable responses from our team. These responses act like a ``mirror", by analyzing how these responses vary with the opponent's shifting intentions and our own historical state, we can inversely reveal the latent strategy at play.
>
> To summarize the relationship between these three concepts: Mental state is a set comprising the intention and latent strategy  that drive the opponent's behavior. The intention determines the selection of strategies, and the execution of strategies collectively shapes the observable trajectories of both the opponent and our team.

---

> ### Author Response · Authors · 2025-11-18
>
> # Difference
>
> Our approach differs fundamentally from existing baselines in the following aspects:
>
> **Information Constraints**: In our setting, cooperative agents and opponents are isolated, with no access to each other's **private information (including the other party's observations, actions, policy parameters, rewards, or intentions).**
>
> **Integrated Online Learning Framework**: Our method belongs to the **world model-based MARL paradigm**, where the world model for opponent modeling and the MARL policy are learned synchronously in an online manner. Specifically:
>
> - The MARL policy, conditioned on the intention $z_I$ and latent strategy $z_L$ inferred by the world model, interacts with the environment to collect data.
>
> - This interaction data is used to update the world model.
>
> - The updated world model then generates additional synthetic data through policy rollout.
>
> - Both real interaction data and model-generated data are jointly utilized to update the MARL policy, forming a closed-loop learning cycle.
>
> Key Differentiation from Prior Work: Unlike methods in references [1,2,3] that separately learn opponent models and RL policies:
>
> - These baselines typically rely on pre-training strategy representations using opponents' private information, followed by policy learning.
>
> - This represents an **offline learning** paradigm, whereas our approach achieves fully integrated **online learning of opponent modeling and multi-agent decision-making**
> - The purpose of offline learning is to use a fixed dataset to find a potentially optimal strategy.
>
> - In our method, **both teams are trained as independent learners**, avoiding built-in AI or self-play. Two teams engage in independent policy learning, value learning, and world model learning based on their local observations due to the limitation of imperfect game. During the execution phase, the policy of each agent relies solely on: (1) local observations
> relative to cooperative adjacent agents ( O_c), (2) local observations relative to opponents ( O_opp),
> and (3) mental states inferred via the hierarchical model.
>
> **The recommended baselines require us to design manual policies specifically for this environment, yet crafting high-quality strategies in such a complex situational setting is extremely challenging. This also implies that these methods necessitate either pre-training or manual policy design for different tasks, highlighting their weak adaptability.**

---

> ### Author Response · Authors · 2025-11-19
>
> # Improvements in presentations
>
> Dear reviewer,
>
> We have thoroughly revised the manuscript to address the concerns regarding clarity and structure:
>
> - Section 3 now begins with clear intuitions and justifications behind the model’s components, and explanations of our framework's separation between intention inference and strategy deduction.
>
> - We have now moved the detailed technical specifications of the modules to the main text and simplified mathematical presentations and reduced acronym clutter while maintaining technical rigor.
>
> -  Each module (H2TE, MITD, LHTE, MLTD) now includes explicit explanations of its role and connections within the overall architecture.
> -  We have provided mechanisms or characteristics of H2TE-MITD and LHTE-MLTD
>
> **These revisions significantly improve the paper's accessibility while preserving its technical contributions. The changes are reflected throughout Sections 3.1-3.3 in the newly uploaded manuscript.**
>
> ## intuitions
> H2TE-MITD infers opponent goals from past observations, extracting macro-behavioral trends (``what they want to do").
> LHTE-MLTD} employs a cognitive logic that shifts from analyzing ``what the opponent has done" to examining ``what outcomes their behavior caused us." The rationale is that an opponent's intention determines its strategy choice, which in turn elicits distinctive team responses. These collective responses serve as a behavioral mirror, allowing inverse deduction of latent strategies by correlating reaction patterns with inferred intentions, thereby identifying which strategies the opponent employed to produce observed team reactions.
>
> ## separation between intents and latent strategy, and mechanisms or characteristics
> Intention inference layer estimates by three core mechanisms: First, the observation-based encoding mechanism,  where H2TE exclusively uses our observations for opponents to construct feature representations, thoroughly avoiding interference from teammate response patterns and ensuring the purity of intention features. Second, the temporal consistency modeling mechanism, where by analyzing behavioral sequences over 512 time steps, extracting macro-behavioral trends that characterize persistent intentions. Third, the threat-centric interpretation mechanism, where MITD directly constructs intention queries around ``which ally faces the greatest threat." Thus, each agent enhances its intent prediction for a given opponent through the team's collective threat consensus. This integrated computational process is mathematically formalized through the Bayesian framework: $P(\text{Intent} \mid H_{opp}, O_{opp}) \propto
> P(O_{opp} \mid \text{Intent}) \cdot
> P(H_{opp} \mid O_{opp}, \text{Intent}) \cdot
> P(\text{Intent})$
>
> The strategy inference layer estimates by building an inverse reasoning framework based on the behavioral mirror principle: LHTE forms a “behavioral mirror" by encoding historical team states, comprehensively recording the characteristic response patterns of the team under various strategic pressures. On this foundation, MLTD implements Bayesian inverse reasoning to establish a complete causal chain from observed effects back to potential strategies. The core of this process lies in the concrete computation of the probability formula $P(\\text{Strategy} \\mid \\text{Response}, \\text{Intent}, O_c) \\propto
> P(O_c \\mid \\text{Intent}) \\cdot
> P(\\text{Response} \\mid \\text{Strategy}, O_c, \\text{Intent}) \\cdot
> P(\\text{Strategy} \\mid \\text{Intent})$: The latent strategy prior is embedded through query initialization, incorporating assumptions about strategy distributions given specific intentions. The intention self-cross attention module computes the observation conditioning term $P(O_c \mid \text{Intent})$, evaluating how current situational evidence aligns with inferred intentions. The likelihood term $P(\text{Response} \mid \text{Strategy}, O_c, \text{Intent})$ is calculated through the latent strategy cross-attention module, assessing how well latent strategies explain both historical patterns and current team reactions.
>
> This dual-layer architecture preserves the advantages of direct observation in intention recognition while ensuring the causal rationality of strategy inference, ultimately achieving precise threat assessment and multi-agent cooperative decision-making through the team consensus mechanism.

---

> ### Author Response · Authors · 2025-11-20
>
> We have thoroughly studied the OMIS work and believe that its methodological paradigm differs fundamentally from our research objectives, making it a potentially unsuitable baseline for comparison.
>
> Specifically, OMIS critically relies on a key prerequisite: **a fixed population of opponent policies must be pre-trained offline with substantial computation. The "Best Response" in its pipeline is merely trained using a standard PPO algorithm against these pre-defined opponents, without incorporating any advanced opponent modeling mechanisms. The subsequent Transformer pre-training essentially learns to imitate and replicate the existing interaction data within this closed set of opponents. By treating opponent actions as explicit supervision labels, its strength lies in rapid retrieval and matching within a known context**, rather than online, strategic-level reasoning and understanding of unknown opponents in an open environment.
>
> Our work, in contrast, addresses a more challenging and general problem: how to perform online inference of opponents' latent intentions and strategies and make dynamic decisions through real-time interaction with the environment, without pre-knowledge of opponent strategies or pre-collected interaction data. This approach is more applicable to real-world adversarial scenarios (like multi-UAV engagements) where the strategy space is open and dynamically evolving. Consequently, comparing our online intention-reasoning model against a method that heavily depends on a pre-defined, offline opponent set and whose core learning component (BR) is just standard PPO may not fairly demonstrate the core advantages of our method in terms of strategic understanding, adaptability, and generalization.
>
>
> There exists a fundamental paradigm difference between GSCU's technical approach and our work. **GSCU relies on a dual pre-definition framework: it requires both a pre-defined opponent policy pool (handcrafted rule-based strategies) and a fixed ego policy during data collection. More critically, it adopts a strongly supervised learning paradigm, depending on accurate policy w labels for model training, which is impractical in real open environments.**
>
> Whether using PBT-trained policies (as in OMIS) or handcrafted rules (as in GSCU), the essence of this approach is to retrieve and respond from a "known opponent library." This introduces two fundamental limitations:
>
>     - Unrealistic Assumption: In real-world multi-agent adversarial scenarios (e.g., UAV game with missiles), we cannot predefine or access the private strategy information of opponents.
>
>     - Limited Adaptability: Such methods fail against novel opponents whose strategies lie outside the training distribution or can evolve autonomously, which is the core challenge in real adversarial settings.
>
> In contrast, our method focuses on solving a more challenging problem: performing real-time inference of opponents' latent intentions and strategy structures through online interactions, without any pre-defined policy knowledge. Our technical advantage lies precisely in handling dynamically evolving, unknown types of intelligent opponents, which is exactly the limitation of methods like GSCU. In fact, in our extensibility experiments section, we systematically evaluated our method's performance against various fixed-strategy opponents and achieved a 100% win rate.

---

> ### Comment · Reviewer_tF3n · 2025-11-27
>
> Thanks for the rebuttal. I have decided to maintain my rating.

---

### Official Review · Reviewer_fJ8q · 2025-11-01

**Soundness:** 2
**Presentation:** 1
**Contribution:** 2
**Rating:** 2
**Confidence:** 4

**Summary:**

The paper studies cooperative-competitive MARL. The paper proposes to learn both intention and strategy representations of opponents and utilizes such information to update beliefs and policies/strategies of the agents involved in the game. The paper conducts experiments in several MARL benchmarks including Gym-JSBSim, SMAC, and GRF, in comparison with multiple baselines including both opponent-model free and opponent-model-based MARL algorithms.

**Strengths:**

1. The paper introduces a new approach to model opponents' decision making (with the goal is to separate intentions from strategies) and integrate it into strategy/policy learning of the agents in the game.

2. Experiments show promising results as the proposed method is shown to perform better than other strong baselines in various benchmarks.

**Weaknesses:**

1. Writing and Clarity
The paper is not well written. In particular, Section 3—the main technical section—requires substantial revision. The section consists of long, dense paragraphs that lack a clear and structured explanation of the proposed model. The heavy use of acronyms and lengthy equations further obscures the main ideas rather than clarifying them. More importantly, the intuitions and justifications behind the model’s components, as well as how they connect, are not clearly explained. A concise, intuitive description of the model and its motivation is necessary to make the paper accessible and convincing.

2. Separation Between Intention and Strategy
The paper needs stronger justification for the proposed separation between intention and strategy. The key question is how the proposed components actually capture intention and distinguish it from strategy prediction. The manuscript does not clearly explain what specific mechanisms or characteristics of H2TE-MITD and LHTE-MLTD enable this distinction. The authors should provide clearer explanations to support the claim that these modules can meaningfully separate intentions from strategies.

3. Cooperative–Competitive Setting
Although the paper discusses a mixed cooperative–competitive setting, the proposed approach appears primarily designed to address competitive interactions. It remains unclear how the model effectively handles both cooperation and competition within the same framework.

4. Baseline Performance and Reliability of Results
The reported performance of baseline methods, such as MAPPO on SMAC environments, is significantly lower than in established works (e.g., the recent HAPPO paper). This discrepancy raises concerns about the experimental setup and the reliability of the reported results. The authors should verify their implementations, hyperparameters, and training conditions to ensure a fair and credible comparison.

5. Supplemental Material
The supplemental zip file could not be opened, preventing further examination of the additional materials. Please ensure that the supplementary files are correctly packaged and accessible.

**Questions:**

Please address my concerns raised in Weaknesses.

---

> ### Author Response · Authors · 2025-11-13
>
> # Response to weakness 1 and 2
> Dear Reviewer,
>
> Thank you very much for your valuable feedback on the writing clarity of our paper. We fully understand your concerns about the readability of Section 3 and would like to clarify that while we have followed a principle similar to previous suggestion of "concept first, details later" during the writing process, the current manuscript structure may not clearly reveal this logical progression.
>
> We would like to emphasize that our core contribution lies not in simply improving an existing model structure to achieve better results, but in proposing a fundamentally new approach to opponent modeling. Unlike most existing methods that rely on opponents' private information such as actions or policy parameters to learn opponents' representations, we draw inspiration from analyzing the hierarchical and recursive reasoning processes in human cognition from lines 71-83. This novel perspective has led us to develop:
>
> - Hierarchical variational inference
>
> - Hierarchical model architecture
>
> - Action-conditioned transition models
>
> - Any-time-step belief update mechanism
>
> - MSOAR-PPO algorithm
>
> We would like to explain the following points:
>
> ## Adherence to Previous Neurips Review Recommendations:
>
> In a previous review round, reviewers explicitly recommended "moving technical details to the appendix and focusing the main text on the core ideas and general introduction of the method." Our current structure directly responds to this suggestion:
>
> - Present the core cognitive foundations and methodological innovations in the main text
>
> - Relocate detailed mathematical derivations and implementation specifics to appendices
>
> - Maintain focus on the overarching framework and comprehension
>
> ## The Overall Methodology Follows an "Inspiration-Overview-Details" Hierarchy:
> **Therefore, our primary objective is to ensure readers first grasp the overall framework and innovative concepts without getting lost in technical details. To achieve this, we have adopted the following structured presentation strategy:**
>
> Cognitive processes of hierarchical thinking and recursive reasoning in the human brain are analyzed by lines 190-243.
>
> **Section 3.1** (Mutual Reaction and Influence Between Mental States, Actions, and Trajectories) serves as the foundation for the "clear and structured explanation" you mentioned. Through a three-stage analysis (lines 190-243), this section systematically elaborates the cognitive basis of our model:
>
> - Stage One reveals the necessity of hierarchical reasoning, directly corresponding to the design motivation of our hierarchical model.
>
> - Stages Two & Three reveal the co-evolution patterns of mental states and behaviors, directly motivating the need for action-conditioned transition models (lines 231-236).
>
> **Section 3.2** develops a hierarchical variational inference framework that, while sharing similarities with RSSM-based Dreamer series, introduces distinct innovations. This derivation formally establishes the core components of our world model (hierarchical representation models, transition models, observation models, and reward model) along with their hierarchical architecture, input-output relationships, and corresponding loss functions.
>
> **Formula Integration with Clear Purpose**
>
> The mathematical formulations serve specific pedagogical purposes:
>
> - They provide precise definitions of the variational inference framework
>
> - Establish clear optimization objectives through HELBO
>
> - Demonstrate how collaborative reasoning is mathematically instantiated
>
> - Each major equation is preceded by intuitive explanations  (lines 249-269)
>
> **Section 3.3 actually provides condensed summaries of each component's functionality rather than accumulating technical details:**
>
> - The functions of multiple attention modules within each component of hierarchical representation models are presented in Section 3.3, deliberately avoiding technical clutter.  The hierarchical representation models comprise high-level and low-level history encoders (H2TE, LHTE), along with intention and strategy decoders (MITD, MLTD).
>
> - Implementation details such as transition models and multi-step update mechanisms are comprehensively elaborated in Appendices A.6-A.7.
>
> - We deliberately moved detailed formulas and complex derivations to the appendix A5.
>
> As described in lines 979 and Figure 6, the LDLRF input includes green z_I and blue z_If from the corresponding outputs of HDIRF intent queries. The multi-dynamic latent strategies (orange) are initialized by the intentions feature: z_L = MLP([Gate(z_If, MLP(z_I)), z_If]). Subsequent latent strategy embeddings are derived from the latent strategies inferred in the previous layer.
>
> We believe the current structure already reflects a clear methodological hierarchy, **but we will further optimize the presentation according to your suggestions to ensure a more explicit logical flow.** Thank you for helping us improve the paper's quality.

---

> ### Author Response · Authors · 2025-11-13
>
> # Response to weakness 1 and 2 (Improvements in Section 3.1)
> We have improved the intuition, and mechanisms behind the model in lines 222-230, and a detailed introduction can be found in Appendix A5.
>
> **The three stages reveal that opponent modeling requires both hierarchical decomposition of mental states (Stage One) and dynamic modeling of their temporal evolution (Stages Two and Three). However, implementing this model faces three fundamental challenges that existing methods fail to address comprehensively. First, the unobservability of opponent's mental states and actions precludes the use of supervised approaches that require opponent actions as labeled data—an unrealistic assumption in real adversarial scenarios. Second, while some methods like VAEs \cite{qi2018intent,Shi2022Motion,wu2023intent} learn static representations from history, they cannot capture the recursive co-evolution of mental states and trajectories revealed in stages two and three. Third, existing world model approaches like \cite{xie2021learning}  still neglected the causal hierarchy and the influence of intentions identified in stage one, limiting their capacity for structured reasoning about opponents' latent strategies.**
>
> **Guided by hierarchical reasoning mechanisms inherent in human cognition (stage one), we propose a hierarchical Transformer architecture composed of H2TE-MITD and LHTE-MLTD for opponent modeling:**
>
> **H2TE-MITD corresponds to the human "intent inference" stage. The H2TE utilizes the historical trajectories relative to opponents to capture the spatiotemporal evolution of the opponent's behavior, then the MITD employs learnable intention queries to decode multiple probable intent hypotheses from this historical context, mirroring how humans judge ``what they want to do." Without this encoding of long-term observation history relative to opponent, intent inference would be myopic and unreliable.**
>
> **LHTE-MLTD corresponds to the human "latent strategy deduction" stage. Its innovation lies in analyzing the historical responses of our own teammates. The cognitive rationale is that an opponent's intent, manifested through their actions, elicits coordinated reactions from our team (e.g., evasion, interception). These reactions serve as a ``mirror" revealing the opponent's latent strategy. The LHTE is responsible for encoding the historical responses of our own teammates. It processes the trajectory observation data of teammates through an attention mechanism to extract the coordinated response patterns exhibited by our team under opponent pressure. The MLTD uses the inferred intents and latent strategies initialized by intents as  key conditioning signals to query these historical teammate response patterns, thereby inferring the specific latent strategy the opponent is employing to execute their intent and predicting how this latent strategy will influence teammate trajectories.**
>
> Consequently, the tight integration of this hierarchical structure (intent vs. latent strategy) with dedicated history encoding (opponent history vs. teammate history) is not a mere assembly of modules but a computational implementation of human hierarchical social reasoning.
>
> The hierarchical model provides a structure for inferring mental states. However, to capture the dynamic interactions between intentions, strategies, actions, and trajectories, we must overcome the limitation of static models (e.g., VAE based methods) and model the evolution. **\textbf{Based on insights from the stages two and three}, we recognize that it is necessary to introduce transition models along with actions. \textbf{Therefore, the mutual reactions between the opponents' intentions and the own team's actions are transformed into transitions of trajectory observation sequences relative to opponents, and the mutual reactions between the opponent's strategies and the own team's actions into transitions of trajectory observation sequences relative to cooperative agents}. This enables the model to capture how mental states and behavior co-evolve over time.** Consequently, our framework integrates these components: \textbf{(1) hierarchical intention-strategy decomposition (H2TE-MITD and LHTE-MLTD), (2) multi-step mental simulation (HJLGT), and (3) any-time-step belief update and dynamic co-adaptation (MSOAR-PPO)}. The dynamic co-adaptation loop between the world model and policy works as follows: The world model infers opponent behavior patterns using learnable intent $z_I$ and strategy $z_L$ queries, while the PPO-based policy leverages these inferences to guide actions, with state transitions recursively updating the world model. Operating through iterative observation-action-reflection cycles, the model mimics human cognitive reasoning, enabling continuous adaptation to new tactics and dynamic inference of evolving opponent intentions and latent strategies.

---

> ### Author Response · Authors · 2025-11-13
>
> # Response to weakness 3
> H2IL-MBOM embeds cooperation as a core mechanism rather than treating it as a separate function. Cooperation here is defined as teammates sharing and co-processing their inferred opponents' mental states to build comprehensive situational awareness beyond any single agent's view. This cooperative design is deeply embedded throughout our hierarchical architecture, as demonstrated by:
> ## Part 1: Cooperation in High-Level Intent Inference (HDIRF)
> Cooperation Mechanism: Teammates share their individual observations relative to the opponents to collaboratively infer opponent intent.
>
> **Cooperation in H2TE**
> Mechanism: The input H_{opp,t} ∈ R^{N × 512 × D} to the H2TE essentially stacks the local historical trajectories regarding the opponents, as observed by each of the N our agents. This builds a team-shared database of  observations relative to opponent.
>
> Formula Evidence:  The HEA captures per-agent spatial dependencies across opponents in local time. The GTr mechanism over the $(T, N*C)$ tensor creates a single computational graph, where temporal dependency along T and instantaneous agent interactions through dot-product are dually captured.
>
> **Cooperation in  Cross-Attention of MITD**:
>
> Mechanism:  This module allows each agent to use its current intent hypothesis to query a "global memory" fused from the entire team's observational history relative to opponents.
>
> Formula Evidence: As shown in Formula (16), the Key and Value are derived from AttH_opp, which is the output of the H2TE—the opponent historical feature fused from the perspectives of N agents. The dimension of this memory bank, R^{1 × (512N) × C}, clearly indicates it contains the encoded  N agents' observations relative to opponents over 512 steps. The interaction between each agent's ${q_{I,c}}$ and this memory bank aggregates entire team's historical perception: "Given my current intent hypothesis, which patterns of opponent behavior, historically seen by which teammate, best support or refute my conjecture?"
>
> **Cooperation in MITD Fusion Module**:
>
> Mechanism: **This is the most critical cooperative step.** This module, based on the shared reasoning results from all agents, determines which teammate is most threatened by the opponent's intent.
>
> Formula Evidence: As shown in Formula (17), the attention in this module is performed across all N agents. It uses the fused features to assess, for each inferred opponent intent, which individual ally is most likely to be the primary target. This directly produces a team-level consensus on threats, guiding subsequent cooperative decisions.
>
> ## Part 2: Cooperation in Low-Level Strategy Inference (LDLRF)
> Cooperation Mechanism: Using the "internal state of the team" as core evidence to infer the "external strategy of the opponent".
>
> **Cooperation in LHTE**:
>
> Mechanism: The input H_{c,t} to the LHTE is the historical states of all our agents themselves. The LHTE is responsible for encoding the historical responses of our own teammates. It processes the trajectory observation data of teammates through an attention mechanism to extract the coordinated response patterns exhibited by our team under opponent pressure.
>
> Formula Evidence: Similar to H2TE, its HEA and GTr operations process the collective historical state data of the team, outputting a cooperative representation of the team's collaborative state, AttH_c.
>
> **Cooperation in MLTD Intent Cross-Attention**:
>
> Mechanism: This module takes the inferred opponent intent z_I from the high level as the Query, and examines the current response constituted by our team's cooperative state E_c.
>
> Formula Evidence: As shown in Formula (18), this links the opponent's low-level intent to observable team reactions, forming a contextualized intent-driven latent strategy. For each given opponent intent (e.g., ``approach agent A"), this module checks: ``How are the teammates currently reacting and thus, how threatening this intent seems?"
>
> **Cooperation in Latent Strategy Cross-Attention of MLTD**:
>
> Mechanism: It uses a strategy query fused with the current context to retrieve the historical cooperative response patterns of our team to find the opponent strategy that best explains the current situation.
>
> Formula Evidence: As shown in Formula (19), the Key and Value k_{L,c}, v_{L,c} come from AttH_c, which represents the historical patterns of how our team collectively responded to threats as a whole. The MLTD uses the inferred intents and latent strategies initialized by intents as  key conditioning signals to query these historical teammate response patterns, thereby inferring the specific latent strategy the opponent is employing to execute their intent and predicting how this latent strategy will influence on teammate trajectories.  Thereby, we abduce what their current tactic is."

---

> ### Author Response · Authors · 2025-11-13
>
> **The fuse attention** :
>
> **This is the most critical cooperative step.** This module achieves team-level threat assessment through a cross-agent attention mechanism, identifying the most frequent responses to pinpoint primary threats. This collective judgment, in turn, drives each agent to continuously refine its own reasoning about the opponent's strategic focus.
>
> Manifestation of Cooperation:
> - The input dimension is  [T×2m×N×2C].
>
> - Attention is computed across the N teammates, assigning a threat weight to each opponent strategy.
>
> **Clarify the focus of the opponent's mental state in the collaborative structure of the team, so as to develop precise response measures, such as prioritizing teammates to support the members most at risk.**

---

> ### Author Response · Authors · 2025-11-13
>
> # Response to weakness 4
> Thank you for your concerns regarding the reliability of the baseline performance. We would like to clarify that the performance evaluation of all algorithms reported in our study (with win rate as the core metric) is based on a unified benchmark **aligned with the advanced world model MAZero, using interaction steps at the scale of 1e4 to 1e6.** This approach is designed to facilitate the most direct and fair comparison of model efficiency, and the results genuinely and strongly support our core conclusions.
>
> ## Experimental Setup Focused on Evaluating World Model Efficiency
> The primary contribution of this work is a superior world model. Consequently, the fairest approach for comparison is to evaluate our model against the current state-of-the-art world model (e.g., MAZero) under identical simulation steps (1e4-1e6 steps), assessing their efficiency in learning and planning from equivalent interaction data.
>
> ## Explicit Unified Benchmark and Core Metric
> **As explicitly stated in lines 1440–1441 of our paper and the last column of Table 6: "The total interactive steps are aligned with the settings used in MAZero to ensure a fair and valid comparison."** Under this benchmark of limited interaction steps (1e4-1e5), we use win rate as the core performance metric to evaluate all algorithms. This means that our method (H2IL-MBOM), the baseline world model (MAZero), and reference model-free methods (e.g., MAPPO) all compete for the same win rate objective under identical, constrained interaction steps.
>
> ## Rationale for Baseline Performance and Our Superiority
> The reported performance of MAPPO is reasonable, accurate, and **consistent with results reported in MAZero(mappo results are depicted by green lines of Figure3 in MAzero)** literature under our experimental settings because:
>
> - **The interaction steps (1e4-1e6) used in our comparison are significantly fewer than the 1e7-4e7 steps typically required for model-free methods like MAPPO to reach peak performance.**
>
> - This constrained setting fairly tests all methods' learning efficiency.
>
> - Despite these constraints, our method demonstrates clear superiority over both MAZero and MAPPO.
>
> Under this benchmark, the results incontrovertibly demonstrate the superiority of our Model
>
> - **First-Level Advantage (Superiority Among Peers)**: Our world model achieves a significantly higher win rate, performing markedly better than the baseline world model MAZero under the same interaction steps (1e4-1e6). This directly validates the advancement of our proposed hierarchical architecture.
>
> - **Second-Level Advantage (Efficiency Dominance)**: The win rate attained by our world model with limited data also substantially surpasses that of model-free methods (e.g., MAPPO) under the same step constraint. The relatively lower performance of MAPPO under this setting is expected and reasonable, which precisely underscores the remarkable sample efficiency of our world model, enabling it to learn and prevail in competition from scant data.
>
> These dual advantages collectively form robust evidence of the excellence of our approach.
>
> ## Experimental Reliability Assurance
>
> We affirm that the implementations and hyperparameters of all baseline algorithms adhere to their original papers or authoritative code repositories. Multiple runs with different random seeds were conducted to ensure statistical reliability.
>
> In summary, using the same limited interaction steps (1e4-1e6) as MAZero and measured by win rate, our proposed H2IL-MBOM framework demonstrates performance superiority over all types of baselines.

---

> ### Author Response · Authors · 2025-11-13
>
> # Response to weakness 1 about Intention-Strategy Separation and Justifications (Improvement in Section 3.1)
> Dear Reviewer,
>
> Thank you for raising this crucial question regarding the separation mechanism between intention and strategy. We would like to emphasize that this separation is an inherent characteristic of our model's architectural design, as explicitly stated in the core contribution outlined in lines 79-96 of our manuscript:
>
> "High-level History Transformer Encoder (H2TE) + Multi-Intention Transformer Decoder (MITD) for inferring multi-intention queries from opponents' past trajectories and predicting future ones. Low-level Dynamic Latent-Strategy-aware Representation Fusion (LDLRF): Low-level History Transformer Encoder (LHTE) + Multi-Latent-Strategy Transformer Decoder (MLTD) for predicting latent strategy queries and allied-agent trajectories based on estimated intentions and historical responses."
>
> This design embodies the **core philosophy of "inferring intentions from opponents' historical trajectories and understanding latent strategies from teammate responses."** Specifically, the components achieve the separation between intention and strategy through the following mechanisms:
>
> 1. Inherent Isolation of Data Sources and Information Pathways
>
> **Intention:** Refers to the opponent's high-level, relatively stable tactical objectives or goal states. It answers the question, "What does the opponent want to achieve?" Intentions are typically discrete or low-dimensional continuous variables that describe macroscopic tactical directions, such as:
>
> - "Attacking" a specific unit
>
> - "Retreating" to preserve strength
>
> - "Occupying" a key area
>
> **Latent Strategy**: Refers to the specific, contextualized execution methods or behavioral patterns that an opponent adopts to implement its intentions. It answers the question, "How does the opponent achieve its intention?"Latent Strategies are instantiations of intentions within specific environmental contexts and evolve dynamically according to the battlefield situation (particularly the responses of our team). For example, for the "attack" intention, latent strategies might include:
>
> - "Leveraging angular advantage" for ambush
>
> - "Utilizing altitude superiority" for suppression
>
> Intention Pathway: The H2TE-MITD components strictly process the historical observation relative to opponent's trajectory, with the optimization objective of reconstructing and predicting the opponent's future trajectories. **This compels the module to learn macro-level trends and consistent patterns in opponent behavior, forming explicit behavioral pattern representations.**
>
> Strategy Pathway: The LHTE-MLTD components specifically handle our teammates' historical response data, conditioned on pre-estimated intentions, to explain and **predict teammates' trajectory changes**. The underlying logic is this: the opponent's inferred intention dictates their adopted strategy, which in turn **triggers distinct, observable responses from our team. These responses act like a ``mirror", by analyzing how these responses vary with the opponent's shifting intentions and our own historical state, we can inversely reveal the latent strategy at play**. **This forces the module to learn the latent causal relationships between  patterns in opponent behavior and our team's reactions, forming implicit causal mechanism representations.**
>
> 2. Fundamental Differences in Learning Objectives
>
> The intention module serves as a temporal pattern recognizer, answering "where is the opponent most likely to appear next"
>
> The strategy module serves as a causal reasoning engine, answering "why are my teammates moving in this particular way"
>
> Through this separation mechanism, in data sources, and learning objectives, our components naturally capture mental states at different levels of abstraction: **intentions encode the inherent patterns of opponent behavior, while strategies encode the mapping relationships between behaviors pattern and responses.**

---

> ### Author Response · Authors · 2025-11-14
>
> # Response to justification  (Improvements in introduction and section 3.2)
> Opponent modeling and intention reasoning lie at the heart of Theory of Mind (ToM), enabling agents to infer opponents' latent mental states, such as goals, beliefs, preferences, and strategies. The cognitive foundations for this capability run deep: research in developmental psychology demonstrates that even infants separate enduring goals from situational actions, recognizing that intentions remain stable while strategies adapt to contextual efficiency [1]. This intention-strategy dissociation is further supported by neuroscientific evidence showing distinct neural encodings for high-level goals in prefrontal regions versus action execution in inferior frontoparietal circuits [2]. Computational studies of human mentalizing reveal that this separation enables hierarchical causal reasoning—humans first infer others' task-relevant goals, then derive the specific action plans employed to achieve them [3].
>
> **This provides direct theoretical justification for our model's hierarchical inference process, where intention recognition precedes strategy estimation.**
>
> [1]  György Gergely and Gergely Csibra. Teleological reasoning in infancy: The infant’s naive theory of
> rational action. Trends in Cognitive Sciences, 63(2):227–233, 1997
>
> [2] Antonia F de C. Hamilton and Scott T Grafton. Action outcomes are represented in human inferior
> frontoparietal cortex. Cerebral Cortex, 18(5):1160–1168, 2008
>
> [3] Chris L Baker, Julian Jara-Ettinger, Rebecca Saxe, and Joshua B Tenenbaum. Rational quantitative
> attribution of beliefs, desires and percepts in human mentalizing. Nature Human Behaviour, 1(4):
> 0064, 2017
>
> The inputs of LDLRF are historical observations ${H_{c,t}} \in {\mathbb{R}^{N \times 512 \times D}}$ to cooperative agents, current observations ${O_c}$ to cooperative neighbors,  intent prediction ${z_I}$, and  latent strategies ${z_L}$, where the multi-dynamic latent strategies are initialized by the intentions feature ${z_I}$ generated by MITD: ${z_L} = MLP([Gate({z_{If}},MLP({z_I})),{z_{If}}]) \in {\mathbb{R}^{T \times 2m \times N \times C}}$ as defined in Figure 6. The subsequent latent strategies embeddings are derived from the latent strategies inferred in the previous layer.
>
> Intention inference layer estimates by three core mechanisms: First, the observation-based encoding mechanism,  where H2TE exclusively uses our observations for opponents to construct feature representations, thoroughly avoiding interference from teammate response patterns and ensuring the purity of intention features. Second, the temporal consistency modeling mechanism, where by analyzing behavioral sequences over 512 time steps, extracting macro-behavioral trends that characterize persistent intentions. Third, the threat-centric interpretation mechanism, where MITD directly constructs intention queries around ``which ally faces the greatest threat." Thus, each agent enhances its intent prediction for a given opponent through the team's collective threat consensus. This integrated computational process is mathematically formalized through the Bayesian framework: $P(\text{Intent} \mid H_{opp}, O_{opp}) \propto
> P(O_{opp} \mid \text{Intent}) \cdot
> P(H_{opp} \mid O_{opp}, \text{Intent}) \cdot
> P(\text{Intent})$
>
> The strategy inference layer estimates by building an inverse reasoning framework based on the behavioral mirror principle: LHTE forms a “behavioral mirror" by encoding historical team states, comprehensively recording the characteristic response patterns of the team under various strategic pressures. On this foundation, MLTD implements Bayesian inverse reasoning to establish a complete causal chain from observed effects back to potential strategies. The core of this process lies in the concrete computation of the probability formula $P(\\text{Strategy} \\mid \\text{Response}, \\text{Intent}, O_c) \\propto
> P(O_c \\mid \\text{Intent}) \\cdot
> P(\\text{Response} \\mid \\text{Strategy}, O_c, \\text{Intent}) \\cdot
> P(\\text{Strategy} \\mid \\text{Intent})$: The latent strategy prior is embedded through query initialization, incorporating assumptions about strategy distributions given specific intentions. The intention self-cross attention module computes the observation conditioning term $P(O_c \mid \text{Intent})$, evaluating how current situational evidence aligns with inferred intentions. The likelihood term $P(\text{Response} \mid \text{Strategy}, O_c, \text{Intent})$ is calculated through the latent strategy cross-attention module, assessing how well latent strategies explain both historical patterns and current team reactions.
>
>
> This dual-layer architecture preserves the advantages of direct observation in intention recognition while ensuring the causal rationality of strategy inference, ultimately achieving precise threat assessment and multi-agent cooperative decision-making through the team consensus mechanism.

---

> ### Author Response · Authors · 2025-11-19
>
> # Improvements in presentations
>
> Dear reviewer,
>
> We have thoroughly revised the manuscript to address the concerns regarding clarity and structure:
>
> - Section 3 now begins with clear intuitions and justifications behind the model’s components, and explanations of our framework's separation between intention inference and strategy deduction.
>
> - We have simplified mathematical presentations and reduced acronym clutter while maintaining technical rigor.
>
> -  Each module (H2TE, MITD, LHTE, MLTD) now includes explicit explanations of its role and connections within the overall architecture.
> -  We have provided mechanisms or characteristics of H2TE-MITD and LHTE-MLTD
>
> **These revisions significantly improve the paper's accessibility while preserving its technical contributions. The changes are reflected throughout Sections 3.1-3.3 in the newly uploaded manuscript.**
>
> ## definitions
> Intention: The opponent's high-level tactical objectives, answering What does the opponent want to achieve?" (e.g.,``Attacking" a specific unit, ``Retreating").}
>
> Latent Strategy: Contextualized execution methods for implementing intentions, answering ``How does the opponent achieve its intention?" (e.g.,Leveraging angular advantage" for an attack intention).}
>
> ## intuitions
> H2TE-MITD infers opponent goals from past observations, extracting macro-behavioral trends (``what they want to do").
> LHTE-MLTD} employs a cognitive logic that shifts from analyzing ``what the opponent has done" to examining ``what outcomes their behavior caused us." The rationale is that an opponent's intention determines its strategy choice, which in turn elicits distinctive team responses. These collective responses serve as a behavioral mirror, allowing inverse deduction of latent strategies by correlating reaction patterns with inferred intentions, thereby identifying which strategies the opponent employed to produce observed team reactions.
>
> ## separation between intents and latent strategy
> Intention inference layer estimates by three core mechanisms: First, the observation-based encoding mechanism,  where H2TE exclusively uses our observations for opponents to construct feature representations, thoroughly avoiding interference from teammate response patterns and ensuring the purity of intention features. Second, the temporal consistency modeling mechanism, where by analyzing behavioral sequences over 512 time steps, extracting macro-behavioral trends that characterize persistent intentions. Third, the threat-centric interpretation mechanism, where MITD directly constructs intention queries around ``which ally faces the greatest threat." Thus, each agent enhances its intent prediction for a given opponent through the team's collective threat consensus. This integrated computational process is mathematically formalized through the Bayesian framework: $P(\text{Intent} \mid H_{opp}, O_{opp}) \propto
> P(O_{opp} \mid \text{Intent}) \cdot
> P(H_{opp} \mid O_{opp}, \text{Intent}) \cdot
> P(\text{Intent})$
>
> The strategy inference layer estimates by building an inverse reasoning framework based on the behavioral mirror principle: LHTE forms a “behavioral mirror" by encoding historical team states, comprehensively recording the characteristic response patterns of the team under various strategic pressures. On this foundation, MLTD implements Bayesian inverse reasoning to establish a complete causal chain from observed effects back to potential strategies. The core of this process lies in the concrete computation of the probability formula $P(\\text{Strategy} \\mid \\text{Response}, \\text{Intent}, O_c) \\propto
> P(O_c \\mid \\text{Intent}) \\cdot
> P(\\text{Response} \\mid \\text{Strategy}, O_c, \\text{Intent}) \\cdot
> P(\\text{Strategy} \\mid \\text{Intent})$: The latent strategy prior is embedded through query initialization, incorporating assumptions about strategy distributions given specific intentions. The intention self-cross attention module computes the observation conditioning term $P(O_c \mid \text{Intent})$, evaluating how current situational evidence aligns with inferred intentions. The likelihood term $P(\text{Response} \mid \text{Strategy}, O_c, \text{Intent})$ is calculated through the latent strategy cross-attention module, assessing how well latent strategies explain both historical patterns and current team reactions.
>
> This dual-layer architecture preserves the advantages of direct observation in intention recognition while ensuring the causal rationality of strategy inference, ultimately achieving precise threat assessment and multi-agent cooperative decision-making through the team consensus mechanism.

---

> ### Author Response · Authors · 2025-11-20
>
> # Response to weakness 1 about Justifications
> ## Validation of Intention-Strategy Separation through Behavioral Visualization
> Our t-SNE visualizations over three episode segments ($\leq$500 steps, 1500–2500 steps, 5000–6000 steps) provide compelling empirical evidence supporting this architectural separation.
>
> The visualization reveals fundamentally different distribution patterns between intention and strategy representations. Intention representations form continuous strip-like distributions across tactical stages, reflecting their role as high-level, persistent tactical objectives that evolve smoothly over time. The observed 11, 7, and 3 intention transitions across three stages demonstrate their macro-level, phase-based nature. In contrast, latent strategy representations maintain separable clusters within each stage, confirming their role as contextualized execution methods that adapt to specific situations while remaining distinct.
>
> This structural divergence validates our core design principle. Intentions represent what opponents want to achieve, as evidenced by their continuous transitional patterns across tactical phases. Strategies represent how they achieve it, demonstrated by their separable context-specific clusters within phases.
>
> Furthermore, the alignment between predicted intention changes (3, 2, 1 per UAV across stages) and actual intention changes demonstrates that our hierarchical separation enables accurate tracking of both global tactical objectives and fine-grained strategic adaptations.
>
> These findings confirm that the separation is not merely architectural but reflects genuine differences in how intentions and strategies manifest in adversarial behavior, providing both empirical justification and interpretable reasoning capabilities.

---

> ### Author Response · Authors · 2025-11-21
> **New videos are updated**
>
> As shown in Figures 10 and 11, the visualization of scenarios depicting engagements between our
> method and MAPPO, as well as engagements between our method and itself, was conducted. The
> figures illustrate that during combat with MAPPO, our maneuver decisions were more agile and rapid,
> resulting in achieving a high altitude and angle advantage with a smaller flight radius, ultimately
> leading to a SR of 3:1. In confrontations with our own method, both sides exhibited similar reasoning
> capabilities, leading to primarily engaging in double loop motion, which represents a classic tactic in
> close-range aerial combat.
>
> Combining Figures 10,8c, and 8d, in the initial stage, the feature distribution range is relatively small,
> indicating both teams frequently make rapid maneuver transitions in a small space (such as climbing,
> making large turns to enter angles, and engaging in single-loop maneuvers). In the middle stage, both
> teams enter the engagement phase, conducting double-loop maneuvers (nose-to-nose approach and
> departure), and missile launches within a larger range. In the final stage, only alive agents engage in
> extensive pursuit and escape strategies. This is consistent with the average number of changes in the
> opponent’s intention predicted by each UAV on average across three stages.
> As shown in Figure 11, in the initial phase, the red team launches missiles first and rapidly dives
> downward at an airspeed of Mach 0.73 to gain kinetic energy. Afterward, it quickly climbs and
> performs a turn. During this phase, one blue aircraft is shot down, while the remaining blue aircraft
> evade the attack by diving and executing counterclockwise yaw maneuvers.
> At this point, the red formation positions itself behind the blue formation, gaining a tactical advantage.
> The red team then accelerates and launches a second missile. In response to the incoming threat, the
> blue formation performs a rapid 180◦counterclockwise turn to evade the second wave of attack.
> The red formation maintains high maneuverability at Mach 0.87, achieves angular superiority for
> the second time, and launches a third missile. The blue formation again executes a swift 180◦
> counterclockwise turn to avoid the third wave of attack.
>
> While the blue team is turning to evade the missile, the red formation simultaneously performs
> aggressive turning maneuvers at Mach 0.92. This ensures that as soon as the blue aircraft complete
> their evasion, the red aircraft are already in a favorable angular position to launch the fourth missile.
> Throughout the engagement, both teams perform turning maneuvers near their respective initial
> positions. The red formation is accompanied by diving and climbing movements, whereas the
> blue formation generally descends while maneuvering counterclockwise. Importantly, the red team
> consistently maintains angular superiority throughout the entire engagement.

---

### Author Response · Authors · 2025-11-19
**Some clarifications**

# Baseline Performance and Reliability of Results
## SMAC
We would like to clarify that the performance evaluation of all algorithms reported in our study (with win rate as the core metric) is based on a unified benchmark **aligned with the advanced world model MAZero, using interaction steps at the scale of 1e4 to 1e6.** This approach is designed to facilitate the most direct and fair comparison of model efficiency, and the results genuinely and strongly support our core conclusions.

## Experimental Setup Focused on Evaluating World Model Efficiency
The primary contribution of this work is a superior world model. Consequently, the fairest approach for comparison is to evaluate our model against the current state-of-the-art world model (e.g., MAZero) under identical simulation steps (1e4-1e6 steps), assessing their efficiency in learning and planning from equivalent interaction data.

## Explicit Unified Benchmark and Core Metric
**As explicitly stated in lines 1440–1441 of our paper and the last column of Table 6: "The total interactive steps are aligned with the settings used in MAZero to ensure a fair and valid comparison."** Under this benchmark of limited interaction steps (1e4-1e5), we use win rate as the core performance metric to evaluate all algorithms. This means that our method (H2IL-MBOM), the baseline world model (MAZero), and reference model-free methods (e.g., MAPPO) all compete for the same win rate objective under identical, constrained interaction steps.

## Rationale for Baseline Performance and Our Superiority
The reported performance of MAPPO is reasonable, accurate, and **consistent with results reported in MAZero(mappo results are depicted by green lines of Figure3 in MAzero)** literature under our experimental settings because:

- **The interaction steps (1e4-1e6) used in our comparison are significantly fewer than the 1e7-4e7 steps typically required for model-free methods like MAPPO to reach peak performance.**

- This constrained setting fairly tests all methods' learning efficiency.

- Despite these constraints, our method demonstrates clear superiority over both MAZero and MAPPO.

Under this benchmark, the results incontrovertibly demonstrate the superiority of our Model

- **First-Level Advantage (Superiority Among Peers)**: Our world model achieves a significantly higher win rate, performing markedly better than the baseline world model MAZero under the same interaction steps (1e4-1e6). This directly validates the advancement of our proposed hierarchical architecture.

- **Second-Level Advantage (Efficiency Dominance)**: The win rate attained by our world model with limited data also substantially surpasses that of model-free methods (e.g., MAPPO) under the same step constraint. The relatively lower performance of MAPPO under this setting is expected and reasonable, which precisely underscores the remarkable sample efficiency of our world model, enabling it to learn and prevail in competition from scant data.

These dual advantages collectively form robust evidence of the excellence of our approach.

## Experimental Reliability Assurance

We affirm that the implementations and hyperparameters of all baseline algorithms adhere to their original papers or authoritative code repositories. Multiple runs with different random seeds were conducted to ensure statistical reliability.

In summary, using the same limited interaction steps (1e4-1e6) as MAZero and measured by win rate, our proposed H2IL-MBOM framework demonstrates performance superiority over all types of baselines.

---

### Author Response · Authors · 2025-11-19
**Some clarifications**

# Difference between ours and other methods


Our approach differs fundamentally from existing baselines in the following aspects:

**Information Constraints**: In our setting, cooperative agents and opponents are isolated, with no access to each other's **private information (including the other party's observations, actions, policy parameters, rewards, or intentions).**

**Integrated Online Learning Framework**: Our method belongs to the **world model-based MARL paradigm**, where the world model for opponent modeling and the MARL policy are learned synchronously in an online manner. Specifically:

- The MARL policy, conditioned on the intention $z_I$ and latent strategy $z_L$ inferred by the world model, interacts with the environment to collect data.

- This interaction data is used to update the world model.

- The updated world model then generates additional synthetic data through policy rollout.

- Both real interaction data and model-generated data are jointly utilized to update the MARL policy, forming a closed-loop learning cycle.

Key Differentiation from Prior Work: Unlike methods in references [1,2,3] that separately learn opponent models and RL policies:

- These baselines typically rely on **pre-training strategy representations** using opponents' private information, followed by policy learning.

- This represents an **offline learning paradigm**, whereas our approach achieves fully integrated **online learning** of opponent modeling and multi-agent decision-making
- The purpose of offline learning is to use a **fixed dataset** to find a potentially optimal strategy.
- In our method, **both teams are trained as independent learners**, avoiding built-in AI or self-play. Two teams engage in independent policy learning, value learning, and world model learning based on their local observations due to the limitation of imperfect game. During the execution phase, the policy of each agent relies solely on: (1) local observations
relative to cooperative adjacent agents ( O_c), (2) local observations relative to opponents ( O_opp),
and (3) mental states inferred via the hierarchical model.

Existing methods, such as [1], [2], and [3], share a common limitation: **their training processes heavily rely on opponents' private information (e.g., actions or policy parameters) as explicit supervision labels**, which is often impractical in real-world adversarial settings. Moreover, these approaches typically lack a transparent, hierarchical reasoning process. They fail to elucidate **how estimated intentions influence latent strategies**, how agents should react to these inferred mental states, or how their evolution guides the prediction of future trajectories. This opaque reasoning is compounded by the absence of an explicit transition model to capture the dynamics between mental states, leaving them **unable to continuously reason about the temporal evolution of intentions and strategies.**

Moreover, these methods have primarily been evaluated in low-dimensional or simplified environments, such as two-player Kuhn Poker, 2D grid-world Predator-Prey, Multi-Agent Particle Environments (MPE), Level-Based Foraging, and Overcooked. Only the CSP method has been tested on the Google Research Football platform.

In contrast, our approach requires **no access to any private information** of the opponents and relies entirely on observational data for unsupervised learning, making it more **suitable for real-world applications**. Beyond this practical advantage, it establishes an explicit **hierarchy from intentions to strategies, with a dynamic model capturing their temporal evolution.**

In terms of experimental validation, we have compared different baselines including model-free MARL, opponent modelling methods, and world-model based MARL in large-scale, high-dimensional multi-agent cooperative–competitive experiments, covering scenarios with varying numbers of UAVs in adversarial games, diverse settings in SMAC, and complex simulations in Google Research Football. Notably, **in the Google Research Football benchmark, our method significantly outperforms CSP across all tested scenarios**: achieving a win rate of 92.54% in 3v1, 89.94% in RPS, and 93.09% in CA. These results robustly demonstrate our model's superior generalization capability and decision-making performance in complex, high-dimensional spaces.

**On one hand, we have supplemented the description of these literature in the Related Work section. On the other hand, the existing experimental setup and results are sufficient to demonstrate the superiority of our method.**

---

> ### Author Response · Authors · 2025-11-20
> **Why OMIS Is Not a Suitable Baseline**
>
> We have thoroughly studied the OMIS work and believe that its methodological paradigm differs fundamentally from our research objectives, making it a potentially unsuitable baseline for comparison.
>
> Specifically, OMIS critically relies on a key prerequisite: **a fixed population of opponent policies must be pre-trained offline with substantial computation. The "Best Response" in its pipeline is merely trained using a standard PPO algorithm against these pre-defined opponents, without incorporating any advanced opponent modeling mechanisms. The subsequent Transformer pre-training essentially learns to imitate and replicate the existing interaction data within this closed set of opponents. By treating opponent actions as explicit supervision labels, its strength lies in rapid retrieval and matching within a known context**, rather than online, strategic-level reasoning and understanding of unknown opponents in an open environment.
>
> Our work, in contrast, addresses a more challenging and general problem: how to perform online inference of opponents' latent intentions and strategies and make dynamic decisions through real-time interaction with the environment, without pre-knowledge of opponent strategies or pre-collected interaction data. This approach is more applicable to real-world adversarial scenarios (like multi-UAV engagements) where the strategy space is open and dynamically evolving. Consequently, comparing our online intention-reasoning model against a method that heavily depends on a pre-defined, offline opponent set and whose core learning component (BR) is just standard PPO may not fairly demonstrate the core advantages of our method in terms of strategic understanding, adaptability, and generalization.

---

> ### Author Response · Authors · 2025-11-20
> **Why GSCU Is Not a Suitable Baseline**
>
> We understand your concern regarding method comparisons and would like to clarify that we have already included comparative validation against fixed-strategy opponents in our extensibility experiments.
>
> However, there exists a fundamental paradigm difference between GSCU's technical approach and our work. **GSCU relies on a dual pre-definition framework: it requires both a pre-defined opponent policy pool (handcrafted rule-based strategies) and a fixed ego policy during data collection. More critically, it adopts a strongly supervised learning paradigm, depending on accurate policy w labels for model training, which is impractical in real open environments.**
>
> In contrast, our method focuses on solving a more challenging problem: performing real-time inference of opponents' latent intentions and strategy structures through online interactions, without any pre-defined policy knowledge. Our technical advantage lies precisely in handling dynamically evolving, unknown types of intelligent opponents, which is exactly the limitation of methods like GSCU. In fact, in our extensibility experiments section, **we also evaluated our method's performance against various fixed-strategy opponents that did not encountered during the training and achieved a 100% win rate.** This result fully demonstrates our method's strong fundamental capabilities in handling such problems.

---

> > ### Author Response · Authors · 2025-11-20
> >
> > Whether using PBT-trained policies (as in OMIS) or handcrafted rules (as in GSCU), the essence of this approach is to retrieve and respond from a "known opponent library." This introduces two fundamental limitations:
> >
> >     - Unrealistic Assumption: In real-world multi-agent adversarial scenarios (e.g., UAV game with missiles), we cannot predefine or access the private strategy information of opponents.
> >
> >     - Limited Adaptability: Such methods fail against novel opponents whose strategies lie outside the training distribution or can evolve autonomously, which is the core challenge in real adversarial settings.
> >
> > Overall, OMIS, GSCU, and CSP utilize the target agent’s behavior data to learn static representation, which is impractical in real-world scenarios

---

### Author Response · Authors · 2025-11-19
**Some clarifications**

# Response to Model Complexity

Model Complexity Analysis
## 1. Hypernetwork-based Adaptive Architecture
In multi-agent systems, **conventional approaches either share networks (policy homogenization) or maintain separate networks (causing parameter growth with agent count)**. Our framework addresses both limitations through hypernetworks in the hierarchical representation model, transition model, and policy network (HEA). This design achieves agent **adaptation without parameter inflation**:

- Hierarchical Model: The hypernetwork generates distinct parameters for each agent, enabling non-shared, individualized reasoning and **adaptation to varying opponents**

- Transition Model: As specified in Lines 102-103 and 1070-1072 and Appendix A4, our efficient transition modeling **compresses historical information** into latent neural weights using a hyper-network, **eliminating the need for explicit historical storage** like TSSM while supporting **adaptive mental state estimation for all agents without parameter count increase**
- **hypernetwork-based coordination**

## 2. Implicit Model Ensemble through Multi-step Updates
As discussed in Appendix A7, multiple arbitrary-step updates within **a single model are equivalent to implicit averaging over an ensemble of models with different horizons (k=1,2,...,H, where H ∼ random(maximum horizon)), thereby further reducing model complexity while maintaining performance.**

## 3. Ablation Study Validation

- Figure 4(a): When only the intention reasoning module is retained, the model **cannot deduce which latent strategies employed by the opponents triggered teammates' specific reaction patterns, nor predict their impact on our team's behavior and future trajectories. This results in significant learning oscillations and slower convergence during training**

- Figure 4(f): As mentioned earlier, using shared Transformers across multiple agents leads to reasoning homogenization. The results demonstrate that hypernetworks enable adaptive reasoning for different agents **while avoiding homogenization, without compromising reasoning performance. This approach achieves both faster learning speed and superior final performance**

## 4. Challenges in Multi-UAV Game Environments
The multi-uav domain in gym-jsbsim presents unique challenges that existing methods cannot adequately address:

- Each aircraft is equipped with four autonomously guided missiles with 20-second tracking time

- Extreme velocity differentials: aircraft at 2 Mach (twice the speed of sound) vs missiles at 4 Mach

- High-dimensional state space with complex spatial relationships

- Critical importance of launch timing and resource management

- Highly dynamic and rapidly evolving tactical situations

This demonstrates the critical need for reasoning about both opponents and their launched missiles. Given that missiles travel significantly faster than aircraft, agents must make early and continuous predictions to execute timely attitude adjustments or rapid maneuvers, avoiding disadvantageous positions and evading multiple incoming threats. Effective agents must learn to secure angular and altitude advantages while achieving maximum tactical gains at minimal cost. In this scenario, agents must first learn fundamental flight control before advancing to strategic game play. Existing methods struggle in such intense adversarial environments characterized by high-speed dynamics and rapid attitude changes, particularly in evading missiles (each hit incurs a -100 reward) or gaining positional superiority, which explains why current opponent-based approaches fail to learn effective policies.


## 5. Advantages of Our Approach
Our method addresses these limitations through:

- Privacy-preserving modeling: Requires no access to opponent's private information (actions, policy parameters, or rewards)

- Hierarchical world model: Enables multi-level mental state **reasoning about both $m$ enemy agents and the maximum observable $m$ weapon systems**

- Temporal dynamics capture: Explicitly models the evolution of mental states over time

- Complex environment adaptation: Successfully handles the challenges of high-speed aerial combat with sophisticated weapon systems

This design **balances model simplicity with competitive performance**, leveraging hypernetworks and hierarchical reasoning to  reason about multi-level mental states of $m$ enemy agents and the maximum observable $m$ weapon systems in complex multi-agent adversarial settings.

---

### Author Response · Authors · 2025-11-19

# justification, mechanisms，and validations
Opponent modeling and intention reasoning lie at the heart of Theory of Mind (ToM), enabling agents to infer opponents' latent mental states, such as goals, beliefs, preferences, and strategies. The cognitive foundations for this capability run deep: research in developmental psychology demonstrates that even infants separate enduring goals from situational actions, recognizing that intentions remain stable while strategies adapt to contextual efficiency [1]. This intention-strategy dissociation is further supported by neuroscientific evidence showing distinct neural encodings for high-level goals in prefrontal regions versus action execution in inferior frontoparietal circuits [2]. Computational studies of human mentalizing reveal that this separation enables hierarchical causal reasoning: humans first infer others' task-relevant goals, then derive the specific action plans employed to achieve them [3].

**This provides direct theoretical justification for our model's hierarchical inference process, where intention recognition precedes strategy estimation.**

[1]  György Gergely and Gergely Csibra. Teleological reasoning in infancy: The infant’s naive theory of
rational action. Trends in Cognitive Sciences, 63(2):227–233, 1997

[2] Antonia F de C. Hamilton and Scott T Grafton. Action outcomes are represented in human inferior
frontoparietal cortex. Cerebral Cortex, 18(5):1160–1168, 2008

[3] Chris L Baker, Julian Jara-Ettinger, Rebecca Saxe, and Joshua B Tenenbaum. Rational quantitative
attribution of beliefs, desires and percepts in human mentalizing. Nature Human Behaviour, 1(4):
0064, 2017

The inputs of LDLRF are historical observations ${H_{c,t}} \in {\mathbb{R}^{N \times 512 \times D}}$ to cooperative agents, current observations ${O_c}$ to cooperative neighbors,  intent prediction ${z_I}$, and  latent strategies ${z_L}$, where the multi-dynamic latent strategies are initialized by the intentions feature ${z_I}$ generated by MITD: ${z_L} = MLP([Gate({z_{If}},MLP({z_I})),{z_{If}}]) \in {\mathbb{R}^{T \times 2m \times N \times C}}$ as defined in Figure 6. The subsequent latent strategies embeddings are derived from the latent strategies inferred in the previous layer.

Intention inference layer estimates by three core mechanisms: First, the observation-based encoding mechanism,  where H2TE exclusively uses our observations for opponents to construct feature representations, thoroughly avoiding interference from teammate response patterns and ensuring the purity of intention features. Second, the temporal consistency modeling mechanism, where by analyzing behavioral sequences over 512 time steps, extracting macro-behavioral trends that characterize persistent intentions. Third, the threat-centric interpretation mechanism, where MITD directly constructs intention queries around ``which ally faces the greatest threat." Thus, each agent enhances its intent prediction for a given opponent through the team's collective threat consensus. This integrated computational process is mathematically formalized through the Bayesian framework: $P(\text{Intent} \mid H_{opp}, O_{opp}) \propto
P(O_{opp} \mid \text{Intent}) \cdot
P(H_{opp} \mid O_{opp}, \text{Intent}) \cdot
P(\text{Intent})$

The strategy inference layer estimates by building an inverse reasoning framework based on the behavioral mirror principle: LHTE forms a “behavioral mirror" by encoding historical team states, comprehensively recording the characteristic response patterns of the team under various strategic pressures. On this foundation, MLTD implements Bayesian inverse reasoning to establish a complete causal chain from observed effects back to potential strategies. The core of this process lies in the concrete computation of the probability formula $P(\\text{Strategy} \\mid \\text{Response}, \\text{Intent}, O_c) \\propto
P(O_c \\mid \\text{Intent}) \\cdot
P(\\text{Response} \\mid \\text{Strategy}, O_c, \\text{Intent}) \\cdot
P(\\text{Strategy} \\mid \\text{Intent})$: The latent strategy prior is embedded through query initialization, incorporating assumptions about strategy distributions given specific intentions. The intention self-cross attention module computes the observation conditioning term $P(O_c \mid \text{Intent})$, evaluating how current situational evidence aligns with inferred intentions. The likelihood term $P(\text{Response} \mid \text{Strategy}, O_c, \text{Intent})$ is calculated through the latent strategy cross-attention module, assessing how well latent strategies explain both historical patterns and current team reactions.


This dual-layer architecture preserves the advantages of direct observation in intention recognition while ensuring the causal rationality of strategy inference, ultimately achieving precise threat assessment and multi-agent cooperative decision-making through the team consensus mechanism.

---

> ### Author Response · Authors · 2025-11-19
> **The rationale or intuitions**
>
> H2TE-MITD infers opponent goals from past observations, extracting macro-behavioral trends (``what they want to do"). \textcolor{red}{LHTE-MLTD} employs a cognitive logic that shifts from analyzing ``what the opponent has done" to examining ``what outcomes their behavior caused us." The rationale is that an opponent's intention determines its strategy choice, which in turn elicits distinctive team responses. These collective responses serve as a behavioral mirror, allowing inverse deduction of latent strategies by correlating reaction patterns with inferred intentions, thereby identifying which strategies the opponent employed to produce observed team reactions.

---

> ### Author Response · Authors · 2025-11-20
>
> ## Validation of Intention-Strategy Separation through Behavioral Visualization
> Our t-SNE visualizations over three episode segments ($\leq$500 steps, 1500–2500 steps, 5000–6000 steps) provide compelling empirical evidence supporting this architectural separation.
>
> The visualization reveals fundamentally different distribution patterns between intention and strategy representations. Intention representations form continuous strip-like distributions across tactical stages, reflecting their role as high-level, persistent tactical objectives that evolve smoothly over time. The observed 11, 7, and 3 intention transitions across three stages demonstrate their macro-level, phase-based nature. In contrast, latent strategy representations maintain separable clusters within each stage, confirming their role as contextualized execution methods that adapt to specific situations while remaining distinct.
>
> This structural divergence validates our core design principle. Intentions represent what opponents want to achieve, as evidenced by their continuous transitional patterns across tactical phases. Strategies represent how they achieve it, demonstrated by their separable context-specific clusters within phases.
>
> Furthermore, the alignment between predicted intention changes (3, 2, 1 per UAV across stages) and actual intention changes demonstrates that our hierarchical separation enables accurate tracking of both global tactical objectives and fine-grained strategic adaptations.
>
> 1. Evidence of Inter-Agent Discrimination
>
> - During inference, the inferred intentions of different opponents remain clearly separated in the latent space.
>
> - The latent strategies of each opponent maintain distinct semantic boundaries throughout the interaction, demonstrating stable feature independence .
>
> - The model effectively preserves discriminative characteristics among opponents solely based on their online behavioral observations.
>
> 2. Evidence of Intra-Agent Evolution
>
> - Real-Time Adaptability: Opponent intentions exhibit dynamic adjustment patterns during inference, accurately reflecting their decision-making logic in response to environmental changes.
>
> - Prediction Refinement: The expansion and stabilization of intention distribution, coupled with a significant reduction in state transition frequency, demonstrate increasingly precise and stable predictions of strategic objectives.
>
> - Behavioral Consistency: The latent strategies of each opponent evolve continuously within their specific semantic regions, maintaining core behavioral characteristics while demonstrating flexible tactical adjustments.
>
>
> These findings confirm that the separation is not merely architectural but reflects genuine differences in how intentions and strategies manifest in adversarial behavior, providing both empirical justification and interpretable reasoning capabilities.

---

### Author Response · Authors · 2025-11-19
**Definition of ${O_{opp}}$ and ${H_{opp}}$**

## ${O_{opp}}$ is observation relative to opponent rather than opponent's observation
Furthermore, as defined in Lines 181 and 674 of observation space and illustrated in Figure 1 of our manuscript, **${O_{opp}}$ refers to the observation relative to opponents from the ego agent's perspective**. Together with ${O_{c}}$, the observation relative to cooperative adjacent agents, they constitute the agent's local observation. **${O_{opp}}$ does not represent observations from the opponents' viewpoint.**
## ${H_{j,t}}$  are the  history observations of all agens relative to the  j-th opponent
Besides, as described in lines878, ${H_{opp,t}} \in {\mathbb{R}^{N \times 512 \times D}}$ is composed of ${O_{opp}}$， $D = m \times {d_m}$ is the observation dimensionalities relative to $m$ opponents within the observation scope of each agent. Therefore,  **${H_{j,t}}$  represents the observation history of all agents relative to the  j-th opponent.**

This approach shares conceptual similarities with the Centralized Training with Decentralized Execution (CTDE) paradigm but differs in several key aspects. During the decentralized execution phase, each agent relies solely on: (1) local observations relative to cooperative adjacent agents ( ${O_{c}}$ ), (2) local observations relative to opponents ( ${O_{opp}}$), and (3) mental states inferred via the representation model q. Furthermore, the value function takes as input only the observations (their ${O_{c}}$, ${O_{opp}}$) and inferred mental states of the agent **itself and its allied neighbors, without incorporating observations from opponents**, unlike methods such as MAPPO, which often use global observations including those of adversaries during training. Consequently, neither the policy nor the value function  utilizes opponents' observations.

## Noise observation input from gym-jsbsim
JSBSim employs a real-time physics engine that integrates atmospheric models and sensor modeling to deliver a simulation environment closely approximating real-world flight conditions. In the Gym-JSBSim environment, sensors are indeed subject to noise influences, which are primarily manifested in the following aspects:

**Sensor Noise Characteristics**

- Aerodynamic Sensors: Directly measured atmospheric data (airspeed, barometric altitude) contains Gaussian white noise

- Inertial Measurement Units (IMU): Accelerometers and gyroscopes exhibit drift errors and random walk noise

- GPS Receivers: Position and velocity measurements contain colored noise and multipath effects

- Attitude Sensors: Euler angle and quaternion estimations are affected by vibrational environmental factors

 By incorporating Kalman filters or complementary filters, the system effectively combines the high-frequency response of IMU with the low-frequency stability of GPS, thereby significantly enhancing the accuracy of attitude estimation.

---

### Author Response · Authors · 2025-11-19
**Reasons for the failure of other opponent modeling methods**

# Reasons for the failure of other opponent modeling methods
We have conducted extensive parameter tuning and five random seed experiments for all algorithms. The reasons for the failure of these methods are as follows:

## 1. Inadequate Temporal Reasoning
Existing opponent modeling methods are unable to continuously reason about the temporal evolution of intentions and strategies. They typically operate on static or short-term representations, failing to capture the dynamic nature of adversarial interactions in complex environments.

## 2. Limited Environmental Validation
These methods have been primarily designed and validated in simplified settings:

- PR2 and ROMMEO were only tested in matrix games and differential games

- TDOM was evaluated solely in differential games, MPE, and predator-prey scenarios

- AORPO was trained and tested only on climb tasks and MPE environments

Besides, the methods [1,2, 3] recommended by the second reviewer were also been evaluated in low-dimensional or simplified environments, such as two-player Kuhn Poker, 2D grid-world Predator-Prey, Multi-Agent Particle Environments (MPE), Level-Based Foraging, and Overcooked. Only the CSP method has been tested on the Google Research Football platform.

In terms of experimental validation, we have compared different baselines including model-free MARL, opponent modelling methods, and world-model based MARL in large-scale, high-dimensional multi-agent cooperative–competitive experiments, covering scenarios with varying numbers of UAVs in adversarial games, diverse settings in SMAC, and complex simulations in Google Research Football. Notably, in the Google Research Football benchmark, **our method significantly outperforms CSP across all tested scenarios**: achieving a win rate of 92.54% in 3v1, 89.94% in RPS, and 93.09% in CA. These results robustly demonstrate our model's superior generalization capability and decision-making performance in complex, high-dimensional spaces.

## 3. Challenges in Multi-UAV Game Environments
The multi-uav domain in gym-jsbsim presents unique challenges that existing methods cannot adequately address:

- Each aircraft is equipped with four **autonomously guided missiles with 20-second engagement capability**

- Extreme velocity differentials: **aircraft at 2 Mach (twice the speed of sound) vs missiles at 4 Mach**

- High-dimensional state space with complex spatial relationships

- Critical importance of launch timing and resource management

- Highly dynamic and rapidly evolving tactical situations

## 4. Advantages of Our Approach
Our method addresses these limitations through:

- Privacy-preserving modeling: Requires no access to opponent's private information (actions, policy parameters, or rewards)

- Hierarchical world model: Enables multi-level mental state **reasoning about both $m$ enemy agents and the maximum observable $m$ weapon systems**

- Temporal dynamics capture: Explicitly models the evolution of mental states over time

- Complex environment adaptation: Successfully handles the challenges of high-speed aerial combat with sophisticated weapon systems

This demonstrates the critical need for reasoning about both opponents and their launched missiles. Given that missiles travel significantly faster than aircraft, agents must make early and continuous predictions to execute timely attitude adjustments or rapid maneuvers, avoiding disadvantageous positions and evading multiple incoming threats. Effective agents must learn to secure angular and altitude advantages while achieving maximum tactical gains at minimal cost. In this scenario, agents must first learn fundamental flight control before advancing to strategic game play. Existing methods struggle in such intense adversarial environments characterized by high-speed dynamics and rapid attitude changes, particularly in evading missiles (each hit incurs a -100 reward) or gaining positional superiority, which explains why current  opponent-based approaches fail to learn effective policies.

This represents the first method for opponent modeling in intense adversarial environments, as well as the first approach to employ world models for opponent modeling. It not only advances the development of world models but also promotes the progress of opponent modeling techniques and contributes to the field of multi-agent adversarial decision-making.

---

### Author Response · Authors · 2025-11-19
**Observable States**

# Observable States
We appreciate the opportunity to clarify the important distinction between observable kinematic states and unobservable control actions in our problem formulation.

That various motion states can be measured by sensors. Indeed, in our environment:

## Observable States (Sensor Measurements):

- Relative position, velocity,

 - Angles and distance relationships

- Altitude and orientation information

These measurements constitute the legitimate observations in our reinforcement learning framework and are consistent with real-world aerial surveillance capabilities. However, the critical distinction lies in what remains unobservable:

## Unobservable Elements (opponents' action spaces described in A3 and opponents' observations relative to cooperative agents):
**For unmanned aerial vehicles**:

Direct aircraft control signals:

- Aileron deflection angles

- Elevator deflection angles

- Rudder deflection angles

- Thrust magnitude settings

- Missile launch activation signals

**For manned aircraft scenarios:**

Pilot's physical control inputs:

- Control stick movements (forward/backward for pitch via elevator, left/right for roll via ailerons)

- Rudder pedal inputs for yaw control

- Throttle lever positions for thrust adjustment

- Weapon release button/trigger activation

We would like to clarify that the relative distance, velocity, and attitude measurements in our model are indeed grounded in realistic aircraft sensing capabilities:
1. **Relative Distance Measurement:**
- **Active electromagnetic ranging using onboard radar systems (primary radar, secondary radar, millimeter-wave radar, lidar)**

- The radar transmits directed electromagnetic pulses toward targets and measures the round-trip time Δt

- Distance is calculated using the fundamental principle: R = c·Δt/2, where c is the speed of light

**This provides accurate relative distance measurements to other aircraft**

2. **Relative Velocity Measurement:**

Inertial/multi-sensor fusion approaches combining:

- **IMU-based measurements** from MEMS accelerometers and gyroscopes

- Continuous integration of velocity changes

- **Fusion with GPS, barometric sensors, and visual odometry data**

**This sensor fusion yields precise 3D velocity vectors relative to ground or surrounding targets**

3. **Relative Attitude/Orientation Measurement:**

- **Phased array radar beam pointing** using active electronically scanned array (AESA) radar

- Electronic scanning rapidly locks onto multiple targets

- Beam pointing angles and target echo phase differences directly provide: Target azimuth relative to aircraft axis,  Target elevation angles, Complete relative orientation information

These sensing methodologies are well-established in aerospace systems and provide the technical foundation for the observable kinematic states in our model. The measurements of relative position, velocity, and attitude **represent realistic sensor capabilities rather than idealized assumptions.**

We will add this technical explanation to the manuscript to better justify our observation space definition and its correspondence to real-world aerial combat sensing capabilities.

Our end-to-end RL controller generates these low-level control signals directly from processed observations. The key insight is that while we can observe the results of opponent actions (kinematic states), we **cannot access their decision-making process, control inputs, opponents' observations relative to cooperative agents, nor the strategic intent behind them.**

---

### Author Response · Authors · 2025-11-26

**The recommended baselines require us to design manual policies specifically for this environment, yet crafting high-quality strategies in such a complex situational setting is extremely challenging. This also implies that these methods necessitate either pre-training or manual policy design for different tasks, highlighting their weak adaptability.** Moreover, they learn static opponent representation using opponents' private information.

---

### Author Response · Authors · 2025-11-26
**Request for Re-evaluation of ICLR Submission 12092**

Dear ICLR Reviewers,

Hope this message finds you well.

I am writing to kindly follow up on our submission  and the rebuttal we submitted. We have not yet received further feedback on our revisions and wanted to ensure you have had the opportunity to review our updated materials.

In response to the reviewers' valuable comments, we have made significant enhancements to the paper. We believe these substantial improvements effectively address the reviewers' concerns and further strengthen our contributions. We would be deeply grateful if you could re-evaluate our work and consider these enhancements in your final assessment. Thank you for your time and consideration.

Best regards,

Authors

---

### Author Response · Authors · 2025-11-27
**The main modifications**

Dear reviewers and chairs,

We hope to clarify the content we have revised. The main modifications are as follows:

1. We have enriched the Introduction by incorporating relevant literature on human hierarchical reasoning and mixed-motive games.

2. As suggested, we have merged the first two paragraphs on hierarchical variational inference into the revised Section 3.1. More importantly, we have **supplemented the design intuition** behind our two-level inference architecture: **H2TE-MITD infers opponent goals from past observations, extracting macro-behavioral trends (“what they want to do”). LHTE-MLTD employs a cognitive logic that shifts from analyzing “what the opponent has done” to examining “what outcomes their behavior caused us.” The rationale is that an opponent's intention determines its strategy choice, which in turn elicits distinctive team responses. These collective responses serve as a behavioral mirror, allowing inverse deduction of latent strategies by correlating reaction patterns with inferred intentions, thereby identifying which strategies the opponent employed to produce observed team reactions.**

3. **In the previous Neurips review round, we received explicit recommendations to "move technical details to the appendix and focus the main text on the core ideas and general introduction of the method, to ensure readers first grasp the overall framework and innovative concepts without getting lost in technical details." In direct response to that feedback, we initially placed the detailed model architecture in the appendix.**

 **However, in light of the current reviewer's valuable perspective that these details are essential for understanding in the main text, we have re-evaluated this balance.**

We have moved the model design details from the appendix to the main body and further elaborated on the computational mechanisms: "Intention inference layer estimates by three core mechanisms: First, the observation-based encoding mechanism,  where H2TE exclusively uses our observations for opponents to construct feature representations, thoroughly avoiding interference from teammate response patterns and ensuring the purity of intention features. Second, the temporal consistency modeling mechanism, where by analyzing behavioral sequences over 512 time steps, extracting macro-behavioral trends that characterize persistent intentions. Third, the threat-centric interpretation mechanism, where MITD directly constructs intention queries around ``which ally faces the greatest threat." Thus, each agent enhances its intent prediction for a given opponent through the team's collective threat consensus. This integrated computational process is mathematically formalized through the Bayesian framework: $P(\text{Intent} \mid H_{opp}, O_{opp}) \propto
P(O_{opp} \mid \text{Intent}) \cdot
P(H_{opp} \mid O_{opp}, \text{Intent}) \cdot
P(\text{Intent})$ The strategy inference layer estimates by building an inverse reasoning framework based on the behavioral mirror principle: LHTE forms a “behavioral mirror" by encoding historical team states, comprehensively recording the characteristic response patterns of the team under various strategic pressures. On this foundation, MLTD implements Bayesian inverse reasoning to establish a complete causal chain from observed effects back to potential strategies. The core of this process lies in the concrete computation of the probability formula $P(\\text{Strategy} \\mid \\text{Response}, \\text{Intent}, O_c) \\propto
P(O_c \\mid \\text{Intent}) \\cdot
P(\\text{Response} \\mid \\text{Strategy}, O_c, \\text{Intent}) \\cdot
P(\\text{Strategy} \\mid \\text{Intent})$: The latent strategy prior is embedded through query initialization, incorporating assumptions about strategy distributions given specific intentions. The intention self-cross attention module computes the observation conditioning term $P(O_c \mid \text{Intent})$, evaluating how current situational evidence aligns with inferred intentions. The likelihood term $P(\text{Response} \mid \text{Strategy}, O_c, \text{Intent})$ is calculated through the latent strategy cross-attention module, assessing how well latent strategies explain both historical patterns and current team reactions."

---

### Note · Authors · 2025-11-29

I have read and agree with the venue's withdrawal policy on behalf of myself and my co-authors.